# Clifford Group Equivariant Neural Networks

**David Ruhe**
AI4Science Lab, AMLab, API
University of Amsterdam
`david.ruhe@gmail.com`

**Johannes Brandstetter**
Microsoft Research
AI4Science
`brandstetter@ml.jku.at`

**Patrick Forré**
AI4Science Lab, AMLab
University of Amsterdam
`p.d.forre@uva.nl`

## Abstract

We introduce Clifford Group Equivariant Neural Networks: a novel approach for constructing $O(n)$- and $E(n)$-equivariant models. We identify and study the *Clifford group*: a subgroup inside the Clifford algebra tailored to achieve several favorable properties. Primarily, the group's action forms an orthogonal automorphism that extends beyond the typical vector space to the entire Clifford algebra while respecting the multivector grading. This leads to several non-equivalent subrepresentations corresponding to the multivector decomposition. Furthermore, we prove that the action respects not just the vector space structure of the Clifford algebra but also its multiplicative structure, i.e., the geometric product. These findings imply that every polynomial in multivectors, including their grade projections, constitutes an equivariant map with respect to the Clifford group, allowing us to parameterize equivariant neural network layers. An advantage worth mentioning is that we obtain expressive layers that can elegantly generalize to inner-product spaces of any dimension. We demonstrate, notably from a single core implementation, state-of-the-art performance on several distinct tasks, including a three-dimensional $n$-body experiment, a four-dimensional Lorentz-equivariant high-energy physics experiment, and a five-dimensional convex hull experiment. ♋

## 1 Introduction

Incorporating *group equivariance* to ensure symmetry constraints in neural networks has been a highly fruitful line of research [CW16, WGTB17, CGKW18, KT18, WC19, WGW$^+$18, Est20, BBCV21, WFVW21, CLW22]. Besides translation and permutation equivariance [ZKR$^+$17, SGT$^+$08], rotation equivariance proved to be vitally important for many graph-structured problems as encountered in, e.g., the natural sciences. Applications of such methods include modeling the dynamics of complex physical systems or motion trajectories [KFW$^+$18, BHP$^+$22]; studying or generating molecules, proteins, and crystals [RDRL14, GLB$^+$16, CTS$^+$17, SSK$^+$18, ZCD$^+$20, TVS$^+$21, AGB22]; and point cloud analysis [WSK$^+$15, UPH$^+$19]. Note that many of these focus on three-dimensional problems involving rotation, reflection, or translation equivariance by considering the groups $O(3)$, $SO(3)$, $E(3)$, or $SE(3)$.

Such equivariant neural networks can be broadly divided into three categories: approaches that scalarize geometric quantities, methods employing regular group representations, and those utilizing irreducible representations, often of $O(3)$ [HRXH22]. Scalarization methods operate exclusively on

---

♋Code is available at `https://github.com/DavidRuhe/clifford-group-equivariant-neural-networks`

37th Conference on Neural Information Processing Systems (NeurIPS 2023).

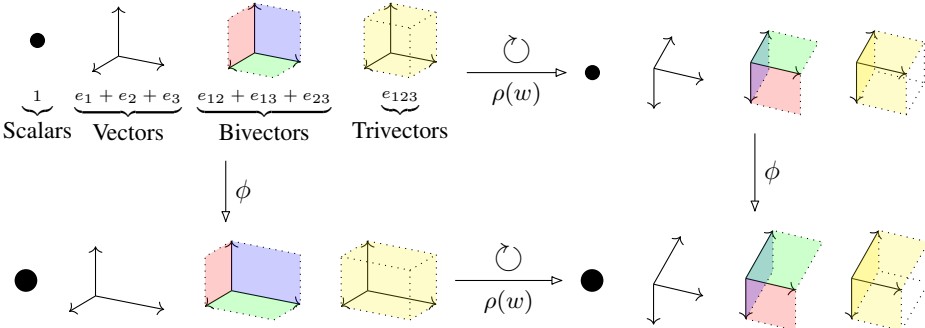

Figure 1: CGENNs (represented with $\phi$) are able to operate on multivectors (elements of the Clifford algebra) in an O($n$)- or E($n$)-equivariant way. Specifically, when an action $\rho(w)$ of the Clifford group, representing an orthogonal transformation such as a rotation, is applied to the data, the model's representations *corotate*. Multivectors can be decomposed into scalar, vector, bivector, trivector, and even higher-order components. These elements can represent geometric quantities such as (oriented) areas or volumes. The action $\rho(w)$ is designed to respect these structures when acting on them.

scalar features or manipulate higher-order geometric quantities such as vectors via scalar multiplication [SSK+18, CCG18, KGG20, KKN20, SHW21, JES+21, GBG21, SUG21, DLD+21, HHR+22, TF22]. They can be limited by the fact that they do not extract all directional information. Regular representation methods construct equivariant maps through an integral over the respective group [CW16, KT18]. For continuous Lie groups, however, this integral is intractable and requires coarse approximation [FSIW20, Bek19]. Methods of the third category employ the irreducible representations of O(3) (the Wigner-D matrices) and operate in a *steerable* spherical harmonics basis [TSK+18, AHK19, FWFW20, BHP+22, BMS+22]. This basis allows a decomposition into type-$l$ vector subspaces that transform under $D^l$: the type-$l$ matrix representation of O(3) [Len90, FA+91]. Through tensor products decomposed using Clebsch-Gordan coefficients (Clebsch-Gordan tensor product), vectors (of different types) interact equivariantly. These tensor products can be parameterized using learnable weights. Key limitations of such methods include the necessity for an alternative basis, along with acquiring the Clebsch-Gordan coefficients, which, although they are known for unitary groups of any dimension [KR67], are not trivial to obtain [AKHvD11].

We propose *Clifford Group Equivariant Neural Networks* (CGENNs): an equivariant parameterization of neural networks based on *Clifford algebras*. Inside the algebra, we identify the *Clifford group* and its action, termed the (adjusted) *twisted conjugation*, which has several advantageous properties. Unlike classical approaches that represent these groups on their corresponding vector spaces, we carefully extend the action to the entire Clifford algebra. There, it automatically acts as an *orthogonal automorphism* that respects the multivector grading, enabling nontrivial subrepresentations that operate on the algebra subspaces. Furthermore, the twisted conjugation respects the Clifford algebra's multiplicative structure, i.e. the *geometric product*, allowing us to bypass the need for explicit tensor product representations. As a result, we obtain two remarkable properties. First, all polynomials in multivectors generate Clifford group equivariant maps from the Clifford algebra to itself. Additionally, *grade projections* are equivariant, allowing for a denser parameterization of such polynomials. We then demonstrate how to construct parameterizable neural network layers using these properties.

Our method comes with several advantages. First, instead of operating on alternative basis representations such as the spherical harmonics, CGENNs (similarly to scalarization methods) directly transform data in a vector basis. Second, multivector representations allow a (geometrically meaningful) product structure while maintaining a finite dimensionality as opposed to tensor product representations. Through geometric products, we can transform vector-valued information, resulting in a more accurate and nuanced interpretation of the underlying structures compared to scalarization methods. Further, we can represent exotic geometric objects such as pseudovectors, encountered in certain physics problems, which transform in a nonstandard manner. Third, our method readily generalizes to orthogonal groups *regardless of the dimension or metric signature of the space*, thereby attaining O($n$)- or E($n$)-equivariance. These advantages are demonstrated on equivariance bench-

marks of different dimensionality. Note that specialized tools were developed for several of these tasks, while CGENNs can be applied more generally.

## 2 The Clifford Algebra

Clifford algebras (also known as *geometric algebras*) are powerful mathematical objects with applications in various areas of science and engineering. For a complete formal construction, we refer the reader to Appendix D. Let $V$ be a finite-dimensional vector space over a field $\mathbb{F}$ equipped with a *quadratic form* $\mathfrak{q} : V \to \mathbb{F}$. The *Clifford algebra* $\mathrm{Cl}(V, \mathfrak{q})$ is the unitary, associative, non-commutative algebra generated by $V$ such that for every $v \in V$ the relation $v^2 = \mathfrak{q}(v)$ holds, i.e., *vectors square to scalars*. This simple property solely generates a unique mathematical theory that underpins many applications. Note that every element $x$ of the Clifford algebra $\mathrm{Cl}(V, \mathfrak{q})$ is a linear combination of (formal, non-commutative) products of vectors modulo the condition that every appearing square $v^2$ gets identified with the scalar square $\mathfrak{q}(v)$: $x = \sum_{i \in I} c_i \cdot v_{i,1} \cdots v_{i,k_i}$.[2] Here, the index set $I$ is finite, $c_i \in \mathbb{F}$, $k \in \mathbb{N}_0$, $v_{i,j} \in V$. The Clifford algebra's associated *bilinear form* $\mathfrak{b}(v_1, v_2) := \frac{1}{2} \left( \mathfrak{q}(v_1 + v_2) - \mathfrak{q}(v_1) - \mathfrak{q}(v_2) \right)$ yields the *fundamental Clifford identity*: $v_1 v_2 + v_2 v_1 = 2\mathfrak{b}(v_1, v_2)$ for $v_1, v_2 \in V$ (Lemma D.3). In this context, the quantity $v_1 v_2$ represents the *geometric product*, which is aptly named for its ability to compute geometric properties and facilitate various transformations. Note that when $v_1, v_2$ are orthogonal (e.g., for orthogonal basis vectors), $\mathfrak{b}(v_1, v_2) = 0$, in which case $v_1 v_2 = -v_2 v_1$. The dimensionality of the algebra is $2^n$, where $n := \dim V$ (Theorem D.15). Let $e_1, \ldots, e_n$ be an orthogonal basis of $V$. The tuple $(e_A)_{A \subseteq [n]}$, $[n] := \{1, \ldots, n\}$, is an orthogonal basis for $\mathrm{Cl}(V, \mathfrak{q})$, where for all such $A$ the product $e_A := \prod_{i \in A}^{<} e_i$ is taken in increasing order (Theorem D.26). We will see below that we can decompose the algebra into vector subspaces $\mathrm{Cl}^{(m)}(V, \mathfrak{q})$, $m = 0, \ldots, n$, called *grades*, where $\dim \mathrm{Cl}^{(m)}(V, \mathfrak{q}) = \binom{n}{m}$. Elements of grade $m = 0$ and $m = 1$ are scalars ($\mathrm{Cl}^{(0)}(V, \mathfrak{q}) = \mathbb{F}$) and vectors ($\mathrm{Cl}^{(1)}(V, \mathfrak{q}) = V$), respectively, while elements of grade $m = 2$ and $m = 3$ are referred to as *bivectors* and *trivectors*. Similar terms are used for elements of even higher grade. These higher-order grades can represent (oriented) points, areas, and volumes, as depicted in Figure 1.

Clifford algebras provide tools that allow for meaningful algebraic representation and manipulation of geometric quantities, including areas and volumes [RDK21, DM02, DFM09]. In addition, they offer generalizations such as extensions of the exterior and Grassmannian algebras, along with the natural inclusion of complex numbers and Hamilton's quaternions [Gra62, Ham66]. Applications of Clifford algebras can be found in robotics [BCRLZE06, HZBC08], computer graphics [WCL05, BTH22], signal processing [Hit12, BMQQS$^+$21], and animation [HP10, CC12]. In the context of machine learning, Clifford algebras and hypercomplex numbers have been employed to improve the performance of algorithms in various tasks. For example, [MFW21] learn an equivariant embedding using *geometric neurons* used for classification tasks. Further, [Spe21] introduce geometric algebra attention networks for point cloud problems in physics, chemistry, and biology. More recently, [BBWG22] introduce Clifford neural layers and Clifford Fourier transforms for accelerating solving partial differential equations. [RGdK$^+$23] continue this direction, strengthening the geometric inductive bias by the introduction of geometric templates. Concurrently with this work, [BDHBC23] develop the geometric algebra transformer. Further, [LK14, PRM$^+$18, ZXXC18] introduce complex-valued and quaternion-valued networks. Finally, [KIdHN23] define normalizing flows on the group of unit quaternions for sampling molecular crystal structures.

## 3 Theoretical Results

In order to construct equivariant multivector-valued neural networks, we outline our theoretical results. We first introduce the following theorem regarding the multivector grading of the Clifford algebra, which is well-known in case the algebra's metric is non-degenerate. Although the general case, including a potentially degenerate metric, appears to be accepted, we were unable to find a self-contained proof during our studies. Hence, we include it here for completeness.

**Theorem 3.1** (The multivector grading of the Clifford algebra). *Let $e_1, \ldots, e_n$ and $b_1, \ldots, b_n$ be two orthogonal bases of $(V, \mathfrak{q})$. Then the following sub-vector spaces $\mathrm{Cl}^{(m)}(V, \mathfrak{q})$ of $\mathrm{Cl}(V, \mathfrak{q})$,*

---

[2]If $k_i = 0$ the product of vectors $v_{i,1} \cdots v_{i,k_i}$ is empty, and, in this case, we mean the unit 1 in $\mathrm{Cl}(V, \mathfrak{q})$.

$m = 0, \ldots, n$, *are independent of the choice of the orthogonal basis, i.e.,*

$$\mathrm{Cl}^{(m)}(V, \mathfrak{q}) := \mathrm{span}\left\{ e_A \mid A \subseteq [n], |A| = m \right\} \overset{!}{=} \mathrm{span}\left\{ b_A \mid A \subseteq [n], |A| = m \right\}. \tag{1}$$

The proof can be found in Theorem D.27. Intuitively, this means that the claims made in the following (and the supplementary material) are not dependent on the chosen frame of reference, even in the degenerate setting. We now declare that the Clifford algebra $\mathrm{Cl}(V, \mathfrak{q})$ decomposes into an orthogonal direct sum of the vector subspaces $\mathrm{Cl}^{(m)}(V, \mathfrak{q})$. To this end, we need to extend the bilinear form $\mathfrak{b}$ from $V$ to $\mathrm{Cl}(V, \mathfrak{q})$. For elements $x_1, x_2, x \in \mathrm{Cl}(V, \mathfrak{q})$, the *extended bilinear form* $\bar{\mathfrak{b}}$ and the *extended quadratic form* $\bar{\mathfrak{q}}$ are given via the projection onto the zero-component (explained below), where $\beta : \mathrm{Cl}(V, \mathfrak{q}) \to \mathrm{Cl}(V, \mathfrak{q})$ denotes the *main anti-involution* of $\mathrm{Cl}(V, \mathfrak{q})^3$ :

$$\bar{\mathfrak{b}} : \mathrm{Cl}(V, \mathfrak{q}) \times \mathrm{Cl}(V, \mathfrak{q}) \to \mathbb{F}, \qquad \bar{\mathfrak{b}}(x_1, x_2) := (\beta(x_1)x_2)^{(0)}, \qquad \bar{\mathfrak{q}}(x) := \bar{\mathfrak{b}}(x, x). \tag{2}$$

Note that by construction, both $\bar{\mathfrak{b}}$ and $\bar{\mathfrak{q}}$ reduce to their original versions when restricted to $V$. Using the extended quadratic form, the tuple $(\mathrm{Cl}(V, \mathfrak{q}), \bar{\mathfrak{q}})$ turns into a quadratic vector space in itself. As a corollary (see Corollary D.30), the Clifford algebra has an orthogonal-sum decomposition w.r.t. the extended bilinear form $\bar{\mathfrak{b}}$:

$$\mathrm{Cl}(V, \mathfrak{q}) = \bigoplus_{m=0}^{n} \mathrm{Cl}^{(m)}(V, \mathfrak{q}), \qquad\qquad \dim \mathrm{Cl}^{(m)}(V, \mathfrak{q}) = \binom{n}{m}. \tag{3}$$

This result implies that we can always write $x \in \mathrm{Cl}(V, \mathfrak{q})$ as $x = x^{(0)} + x^{(1)} + \cdots + x^{(n)}$, where $x^{(m)} \in \mathrm{Cl}^{(m)}(V, \mathfrak{q})$ denotes the grade-$m$ part of $x$. Selecting a grade defines an orthogonal projection:

$$(\_)^{(m)} : \mathrm{Cl}(V, \mathfrak{q}) \to \mathrm{Cl}^{(m)}(V, \mathfrak{q}), \qquad x \mapsto x^{(m)}, \qquad m = 0, \ldots, n. \tag{4}$$

Let us further introduce the notation $\mathrm{Cl}^{[0]}(V, \mathfrak{q}) := \bigoplus_{m \text{ even}}^{n} \mathrm{Cl}^{(m)}(V, \mathfrak{q})$, whose elements are of *even parity*, and $\mathrm{Cl}^{[1]}(V, \mathfrak{q}) := \bigoplus_{m \text{ odd}}^{n} \mathrm{Cl}^{(m)}(V, \mathfrak{q})$ for those of *odd parity*. We use $x = x^{[0]} + x^{[1]}$ to denote the parity decomposition of a multivector.

## 3.1 The Clifford Group and its Clifford Algebra Representations

Let $\mathrm{Cl}^{\times}(V, \mathfrak{q})$ denote the group of invertible elements of the Clifford algebra, i.e., the set of those elements $w \in \mathrm{Cl}(V, \mathfrak{q})$ that have an inverse $w^{-1} \in \mathrm{Cl}(V, \mathfrak{q})$: $w^{-1}w = ww^{-1} = 1$. For $w \in \mathrm{Cl}^{\times}(V, \mathfrak{q})$, we then define the (adjusted) *twisted conjugation* as follows:

$$\rho(w) : \mathrm{Cl}(V, \mathfrak{q}) \to \mathrm{Cl}(V, \mathfrak{q}), \qquad \rho(w)(x) := wx^{[0]}w^{-1} + \alpha(w)x^{[1]}w^{-1}, \tag{5}$$

where $\alpha$ is the *main involution* of $\mathrm{Cl}(V, \mathfrak{q})$, which is given by $\alpha(w) := w^{[0]} - w^{[1]}$. This map $\rho(w) : \mathrm{Cl}(V, \mathfrak{q}) \to \mathrm{Cl}(V, \mathfrak{q})$, notably not just from $V \to V$, will be essential for constructing equivariant neural networks operating on the Clifford algebra. In general, $\rho$ and similar maps defined in the literature do not always posses the required properties (see Motivation E.1). However, when our $\rho$ is restricted to a carefully chosen subgroup of $\mathrm{Cl}^{\times}(V, \mathfrak{q})$, many desirable characteristics emerge. This subgroup will be called the *Clifford group*[4] of $\mathrm{Cl}(V, \mathfrak{q})$ and we define it as:

$$\Gamma(V, \mathfrak{q}) := \left\{ w \in \mathrm{Cl}^{\times}(V, \mathfrak{q}) \cap \left( \mathrm{Cl}^{[0]}(V, \mathfrak{q}) \cup \mathrm{Cl}^{[1]}(V, \mathfrak{q}) \right) \,\middle|\, \forall v \in V, \rho(w)(v) \in V \right\}. \tag{6}$$

In words, $\Gamma(V, \mathfrak{q})$ contains all invertible (parity) homogeneous elements that preserve vectors ($m = 1$ elements) via $\rho$. The *special Clifford group* is defined as $\Gamma^{[0]}(V, \mathfrak{q}) := \Gamma(V, \mathfrak{q}) \cap \mathrm{Cl}^{[0]}(V, \mathfrak{q})$.

Regarding the twisted conjugation, $\rho(w)$ was ensured to reduce to a *reflection* when restricted to $V$ – a property that we will conveniently use in the upcoming section. Specifically, when $w, x \in \mathrm{Cl}^{(1)}(V, \mathfrak{q}) = V$, $w \in \mathrm{Cl}^{\times}(V, \mathfrak{q})$, then $\rho(w)$ reflects $x$ in the hyperplane normal to $w$:

$$\rho(w)(x) = -wxw^{-1} \overset{!}{=} x - 2\frac{\mathfrak{b}(w, x)}{\mathfrak{b}(w, w)}w. \tag{7}$$

Next, we collect several advantageous identities of $\rho$ in the following theorem, which we elaborate on afterwards. For proofs, consider Lemma E.8, Theorem E.10, and Theorem E.29.

---

[3] Recall that any $x \in \mathrm{Cl}(V, \mathfrak{q})$ can be written as a linear combination of vector products. $\beta$ is the map that reverses the order: $\beta\left( \sum_{i \in I} c_i \cdot v_{i,1} \cdots v_{i,k_i} \right) := \sum_{i \in I} c_i \cdot v_{i,k_i} \cdots v_{i,1}$. For details, see Definition D.18.

[4] We elaborate shortly on the term *Clifford group* in contrast to similar definitions in Remark E.14.

$$\overbrace{\mathrm{Cl}(V,\mathfrak{q}) \times \cdots \times \mathrm{Cl}(V,\mathfrak{q})}^{\ell \text{ times}} \xrightarrow{\ F\ } \mathrm{Cl}(V,\mathfrak{q}) \qquad\qquad \mathrm{Cl}(V,\mathfrak{q}) \xrightarrow{\ (\_)^{(m)}\ } \mathrm{Cl}^{(m)}(V,\mathfrak{q})$$

$$\Big\downarrow{\scriptstyle\rho(w)} \quad \Big\downarrow{\scriptstyle\rho(w)} \quad \Big\downarrow{\scriptstyle\rho(w)} \qquad \Big\downarrow{\scriptstyle\rho(w)} \qquad\qquad \Big\downarrow{\scriptstyle\rho(w)} \qquad\qquad\qquad \Big\downarrow{\scriptstyle\rho(w)}$$

$$\mathrm{Cl}(V,\mathfrak{q}) \times \cdots \times \mathrm{Cl}(V,\mathfrak{q}) \xrightarrow{\ F\ } \mathrm{Cl}(V,\mathfrak{q}) \qquad\qquad \mathrm{Cl}(V,\mathfrak{q}) \xrightarrow{\ (\_)^{(m)}\ } \mathrm{Cl}^{(m)}(V,\mathfrak{q})$$

Figure 2: Commutative diagrams expressing Clifford group equivariance with respect to the main operations: polynomials $F$ (left) and grade projections $(\_)^{(m)}$ (right).

**Theorem 3.2.** *Let $w_1, w_2, w \in \Gamma(V,\mathfrak{q})$, $x_1, x_2, x \in \mathrm{Cl}(V,\mathfrak{q})$, $c \in \mathbb{F}$, $m \in \{0, \ldots, n\}$. $\rho$ then satisfies:*

1. *Additivity: $\rho(w)(x_1 + x_2) = \rho(w)(x_1) + \rho(w)(x_2)$,*

2. *Multiplicativity: $\rho(w)(x_1 x_2) = \rho(w)(x_1)\rho(w)(x_2)$, and: $\rho(w)(c) = c$,*

3. *Invertibility: $\rho(w^{-1})(x) = \rho(w)^{-1}(x)$,*

4. *Composition: $\rho(w_2)\,(\rho(w_1)(x)) = \rho(w_2 w_1)(x)$, and: $\rho(c)(x) = x$ for $c \neq 0$,*

5. *Orthogonality: $\bar{\mathfrak{b}}\,(\rho(w)(x_1), \rho(w)(x_2)) = \bar{\mathfrak{b}}\,(x_1, x_2)$.*

The first two properties state that $\rho(w)$ is not only *additive*, but even *multiplicative* regarding the geometric product. The third states that $\rho(w)$ is invertible, making it an *algebra automorphism* of $\mathrm{Cl}(V,\mathfrak{q})$. The fourth property then states that $\rho : \Gamma(V,\mathfrak{q}) \to \mathrm{Aut}_{\mathbf{Alg}}(\mathrm{Cl}(V,\mathfrak{q}))$ is also a *group homomorphism* to the group of all algebra automorphisms. In other words, $\mathrm{Cl}(V,\mathfrak{q})$ is a linear representation of $\Gamma(V,\mathfrak{q})$ and, moreover, it is also an algebra representation of $\Gamma(V,\mathfrak{q})$. Finally, the last point shows that each $\rho(w)$ generates an orthogonal map with respect to the extended bilinear form $\bar{\mathfrak{b}}$. These properties yield the following results (see also Theorem E.16, Corollary E.18, Figure 2).

**Corollary 3.3** (All grade projections are Clifford group equivariant)**.** *For $w \in \Gamma(V,\mathfrak{q})$, $x \in \mathrm{Cl}(V,\mathfrak{q})$ and $m = 0, \ldots, n$ we have the following equivariance property:*

$$\rho(w)(x^{(m)}) = (\rho(w)(x))^{(m)}. \tag{8}$$

*In particular, for $x \in \mathrm{Cl}^{(m)}(V,\mathfrak{q})$ we also have $\rho(w)(x) \in \mathrm{Cl}^{(m)}(V,\mathfrak{q})$.*

This implies that the grade projections: $\mathrm{Cl}(V,\mathfrak{q}) \to \mathrm{Cl}^{(m)}(V,\mathfrak{q})$ are $\Gamma(V,\mathfrak{q})$-equivariant maps, and, that each $\mathrm{Cl}^{(m)}(V,\mathfrak{q})$ constitutes an orthogonal representation of $\Gamma(V,\mathfrak{q})$. The latter means that $\rho$ induces a group homomorphisms from the Clifford group to the group of all orthogonal invertible linear transformations of $\mathrm{Cl}^{(m)}(V,\mathfrak{q})$:

$$\rho^{(m)} : \Gamma(V,\mathfrak{q}) \to \mathrm{O}\left(\mathrm{Cl}^{(m)}(V,\mathfrak{q}), \bar{\mathfrak{q}}\right), \qquad\qquad \rho^{(m)}(w) := \rho(w)|_{\mathrm{Cl}^{(m)}(V,\mathfrak{q})}. \tag{9}$$

**Corollary 3.4** (All polynomials are Clifford group equivariant)**.** *Let $F \in \mathbb{F}[T_1, \ldots, T_\ell]$ be a polynomial in $\ell$ variables with coefficients in $\mathbb{F}$, $w \in \Gamma(V,\mathfrak{q})$. Further, consider $\ell$ elements $x_1, \ldots, x_\ell \in \mathrm{Cl}(V,\mathfrak{q})$. Then we have the following equivariance property:*

$$\rho(w)\,(F(x_1, \ldots, x_\ell)) = F(\rho(w)(x_1), \ldots, \rho(w)(x_\ell)). \tag{10}$$

To prove that $\rho(w)$ distributes over any polynomial, we use both its additivity and multiplicativity regarding the geometric product.

By noting that one can learn the coefficients of such polynomials, we can build flexible parameterizations of Clifford group equivariant layers for quadratic vector spaces of any dimension or metric. We involve grade projections to achieve denser parameterizations. More details regarding neural network constructions follow in Section 4.

## 3.2 Orthogonal Representations

To relate to the *orthogonal group* $\mathrm{O}(V, \mathfrak{q})$, the set of invertible linear maps $\Phi : V \to V$ preserving $\mathfrak{q}$, first note that Equation (5) shows that for every $w \in \mathbb{F}^\times$ we always have: $\rho(w) = \mathrm{id}_{\mathrm{Cl}(V,\mathfrak{q})}$. So the action $\rho$ on $\mathrm{Cl}(V, \mathfrak{q})$ can already be defined on the quotient group $\Gamma(V, \mathfrak{q})/\mathbb{F}^\times$. Moreover, we have:

**Theorem 3.5** (See also Remark E.31 and Corollary E.32 ). *If $(V, \mathfrak{q})$ is* non-degenerate[5] *then $\rho$ induces a well-defined group isomorphism:*

$$\bar{\rho}^{(1)} : \Gamma(V, \mathfrak{q})/\mathbb{F}^\times \xrightarrow{\sim} \mathrm{O}(V, \mathfrak{q}), \qquad\qquad [w] \mapsto \rho(w)|_V. \tag{11}$$

The above implies that $\mathrm{O}(V, \mathfrak{q})$ acts on whole $\mathrm{Cl}(V, \mathfrak{q})$ in a well-defined way. Concretely, if $x \in \mathrm{Cl}(V, \mathfrak{q})$ is of the form $x = \sum_{i \in I} c_i \cdot v_{i,1} \cdots v_{i,k_i}$ with $v_{i,j} \in V$, $c_i \in \mathbb{F}$ and $\Phi \in \mathrm{O}(V, \mathfrak{q})$ is given by $\bar{\rho}^{(1)}([w]) = \Phi$ with $w \in \Gamma(V, \mathfrak{q})$, then we have:

$$\rho(w)(x) = \sum_{i \in I} c_i \cdot \rho(w)(v_{i,1}) \cdots \rho(w)(v_{i,k_i}) = \sum_{i \in I} c_i \cdot \Phi(v_{i,1}) \cdots \Phi(v_{i,k_i}). \tag{12}$$

This means that $\Phi$ acts on a multivector $x$ by applying an orthogonal transformation to all its vector components $v_{i,j}$, as illustrated in Figure 1. Theorem 3.5 implies that that a map $f : \mathrm{Cl}(V, \mathfrak{q})^{\ell_{\mathrm{in}}} \to \mathrm{Cl}(V, \mathfrak{q})^{\ell_{\mathrm{out}}}$ is **equivariant to the Clifford group** $\Gamma(V, \mathfrak{q})$ **if and only if it is equivariant to the orthogonal group** $\mathrm{O}(V, \mathfrak{q})$ when both use the actions on $\mathrm{Cl}(V, \mathfrak{q})$ described above (for each component). To leverage these results for constructing $\mathrm{O}(V, \mathfrak{q})$-equivariant neural networks acting on vector features, i.e., $x \in V^{\ell_{\mathrm{in}}}$, we refer to Section 4.

To prove Theorem 3.5, we invoke *Cartan-Dieudonné* (Theorem C.13), stating that every orthogonal transformation decomposes into compositions of reflections, and recall that $\rho$ reduces to a reflection when restricted to $V$, see Theorem E.25. Theorem 3.5 also allows one to provide explicit descriptions of the element of the Clifford group, see Corollary E.27.

Finally, it is worth noting that our method is also $\mathrm{Pin}$ and $\mathrm{Spin}$ group-equivariant[6]. These groups, sharing properties with the Clifford group, are also often studied in relation to the orthogonal group.

## 4 Methodology

We restrict our layers to use $\mathbb{F} := \mathbb{R}$. Our method is most similar to steerable methods such as [BHP+22]. However, unlike these works, we do not require an alternative basis representation based on spherical harmonics, nor do we need to worry about Clebsch-Gordan coefficients. Instead, we consider simply a steerable vector basis for $V$, which then automatically induces a *steerable multivector basis* for $\mathrm{Cl}(V, \mathfrak{q})$ and its transformation kernels. By steerability, we mean that this basis can be transformed in a predictable way under an action from the Clifford group, which acts orthogonally on both $V$ and $\mathrm{Cl}(V, \mathfrak{q})$ (see Figure 1).

We present layers yielding Clifford group-equivariant optimizable transformations. All the main ideas are based on Corollary 3.3 and Corollary 3.4. It is worth mentioning that the methods presented here form a first exploration of applying our theoretical results, making future optimizations rather likely.

**Linear Layers** Let $x_1, \ldots, x_\ell \in \mathrm{Cl}(V, \mathfrak{q})$ be a tuple of multivectors expressed in a steerable basis, where $\ell$ represents the number of input channels. Using the fact that a polynomial restricted to the first order constitutes a linear map, we can construct a linear layer by setting

$$y_{c_{\mathrm{out}}}^{(k)} := T_{\phi_{c_{\mathrm{out}}}}^{\mathrm{lin}}(x_1, \ldots, x_\ell)^{(k)} := \sum_{c_{\mathrm{in}}=1}^{\ell} \phi_{c_{\mathrm{out}} c_{\mathrm{in}} k} \, x_{c_{\mathrm{in}}}^{(k)}, \tag{13}$$

where $\phi_{c_{\mathrm{out}} c_{\mathrm{in}} k} \in \mathbb{R}$ are optimizable coefficients and $c_{\mathrm{in}}$, $c_{\mathrm{out}}$ denote the input and output channel, respectively. As such, $T_\phi : \mathrm{Cl}(V, \mathfrak{q})^\ell \to \mathrm{Cl}(V, \mathfrak{q})$ is a linear transformation in each algebra subspace

---

[5]Here we restrict $(V, \mathfrak{q})$ to be *non-degenerate* as we never consider degenerate quadratic forms in our experiments. However, in the supplementary material we generalize this also to the *degenerate* case by carefully taking the radical subspace $\mathcal{R}$ of $(V, \mathfrak{q})$ into account on both sides of the isomorphism.

[6]We discuss in Appendix E.6 a general definition of these groups.

$k$. Recall that this is possible due to the result that $\rho(w)$ respects the multivector subspaces. This computes a transformation for the output channel $c_{\text{out}}$; the map can be repeated (using different sets of parameters) for other output channels, similar to classical neural network linear layers. For $y_{c_{\text{out}}}^{(0)} \in \mathbb{R}$ (the scalar part of the Clifford algebra), we can additionally learn an invariant bias parameter.

**Geometric Product Layers**  A core strength of our method comes from the fact that we can also parameterize interaction terms. In this work, we only consider layers up to second order. Higher-order interactions are indirectly modeled via multiple successive layers. As an example, we take the pair $x_1, x_2$. Their interaction terms take the form $\left( x_1^{(i)} x_2^{(j)} \right)^{(k)}$, $i, j, k = 0, \ldots, n$; where we again make use of the fact that $\rho(w)$ respects grade projections. As such, all the grade-$k$ terms resulting from the interaction of $x_1$ and $x_2$ are parameterized with

$$P_\phi(x_1, x_2)^{(k)} := \sum_{i=0}^{n} \sum_{j=0}^{n} \phi_{ijk} \left( x_1^{(i)} x_2^{(j)} \right)^{(k)}, \tag{14}$$

where $P_\phi : \mathrm{Cl}(V, \mathfrak{q}) \times \mathrm{Cl}(V, \mathfrak{q}) \to \mathrm{Cl}(V, \mathfrak{q})$. This means that we get $(n+1)^3$ parameters for every geometric product between a pair of multivectors[7]. Parameterizing and computing all second-order terms amounts to $\ell^2$ such operations, which can be computationally expensive given a reasonable number of channels $\ell$. Instead, we first apply a linear map to obtain $y_1, \ldots, y_\ell \in \mathrm{Cl}(V, \mathfrak{q})$. Through this map, the mixing (i.e., the terms that will get multiplied) gets learned. That is, we only get $\ell$ pairs $(x_1, y_1), \ldots, (x_\ell, y_\ell)$ from which we then compute $z_{c_{\text{out}}}^{(k)} := P_{\phi_{c_{\text{out}}}}(x_{c_{\text{in}}}, y_{c_{\text{in}}})^{(k)}$. Note that here we have $c_{\text{in}} = c_{\text{out}}$, i.e., the number of channels does not change. Hence, we refer to this layer as the *element-wise* geometric product layer. We can obtain a more expressive (yet more expensive) parameterization by linearly combining such products by computing

$$z_{c_{\text{out}}}^{(k)} := T_{\phi_{c_{\text{out}}}}^{\text{prod}}(x_1, \ldots, x_\ell, y_1, \ldots, y_\ell)^{(k)} := \sum_{c_{\text{in}}=1}^{\ell} P_{\phi_{c_{\text{out}} c_{\text{in}}}}(x_{c_{\text{in}}}, y_{c_{\text{in}}})^{(k)}, \tag{15}$$

which we call the *fully-connected* geometric product layer. Computational feasibility and experimental verification should determine which parameterization is preferred.

**Normalization and Nonlinearities**  Since our layers involve quadratic and potentially higher-order interaction terms, we need to ensure numerical stability. In order to do so, we use a normalization operating on each multivector subspace before computing geometric products by putting

$$x^{(m)} \mapsto \frac{x^{(m)}}{\sigma(a_m) \left( \bar{\mathfrak{q}}(x^{(m)}) - 1 \right) + 1}, \tag{16}$$

where $x^{(m)} \in \mathrm{Cl}^{(m)}(V, \mathfrak{q})$. Here, $\sigma$ denotes the logistic sigmoid function, and $a_m \in \mathbb{R}$ is a learned scalar. The denominator interpolates between 1 and the quadratic form $\bar{\mathfrak{q}}\left(x^{(m)}\right)$, normalizing the magnitude of $x^{(m)}$. This ensures that the geometric products do not cause numerical instabilities without losing information about the magnitude of $x^{(m)}$, where a learned scalar interpolates between both regimes. Note that by Theorem 3.2, $\bar{\mathfrak{q}}(x^{(m)})$ is invariant under the action of the Clifford group, rendering Equation (16) an equivariant map.

Next, we use the layer-wide normalization scheme proposed by [RGdK$^+$23], which, since it is also based on the extended quadratic form, is also equivariant with respect to the twisted conjugation.

Regarding nonlinearities, we use a slightly adjusted version of the units proposed by [RGdK$^+$23]. Since the scalar subspace $\mathrm{Cl}^{(0)}(V, \mathfrak{q})$ is always invariant with respect to the twisted conjugation, we can apply $x^{(m)} \mapsto \mathrm{ReLU}\left(x^{(m)}\right)$ when $m = 0$ and $x^{(m)} \mapsto \sigma_\phi\left(\bar{\mathfrak{q}}\left(x^{(m)}\right)\right) x^{(m)}$ otherwise. We can replace ReLU with any common scalar activation function. Here, $\sigma_\phi$ represents a potentially parameterized nonlinear function. Usually, however, we restrict it to be the sigmoid function. Since we modify $x^{(m)}$ with an invariant scalar quantity, we retain equivariance. Such gating activations are commonly used in the equivariance literature [WGW$^+$18, GS22].

---

[7]In practice, we use fewer parameters due to the sparsity of the geometric product, implying that many interactions will invariably be zero, thereby making their parameterization redundant.

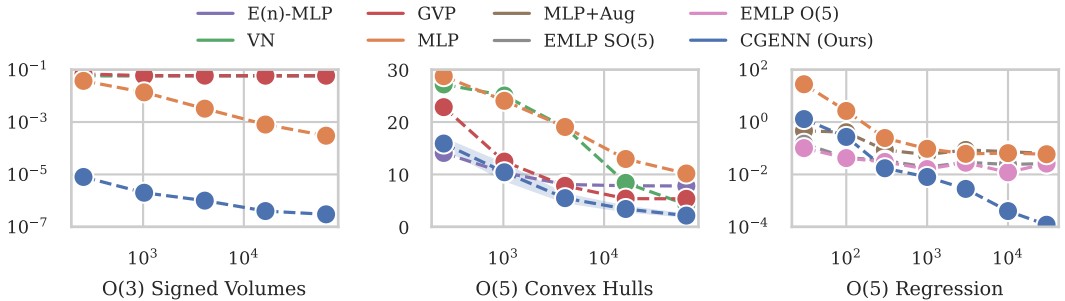

Figure 3: Left: Test mean-squared errors on the O(3) signed volume task as functions of the number of training data. Note that due to identical performance, some baselines are not clearly visible. Right: same, but for the O(5) convex hull task.

Figure 4: Test mean-squared-errors on the O(5) regression task.

**Embedding Data in the Clifford Algebra**  In this work, we consider data only from the vector space $V$ or the scalars $\mathbb{R}$, although generally one might also encounter, e.g., *bivector* data. That is, we have some scalar features $h_1, \ldots, h_k \in \mathbb{R}$ and some vector features $h_{k+1}, \ldots, h_\ell \in V$. Typical examples of scalar features include properties like mass, charge, temperature, and so on. Additionally, one-hot encoded categorical features are also included because $\{0, 1\} \subset \mathbb{R}$ and they also transform trivially. Vector features include positions, velocities, and the like. Then, using the identifications $\mathrm{Cl}^{(0)}(V, \mathfrak{q}) \cong \mathbb{R}$ and $\mathrm{Cl}^{(1)}(V, \mathfrak{q}) \cong V$, we can embed the data into the scalar and vector subspaces of the Clifford algebra to obtain Clifford features $x_1, \ldots, x_\ell \in \mathrm{Cl}(V, \mathfrak{q})$.

Similarly, we can predict scalar- or vector-valued data as output of our model by grade-projecting onto the scalar or vector parts, respectively. We can then directly compare these quantities with ground-truth vector- or scalar-valued data through a loss function and use standard automatic differentiation techniques to optimize the model. Note that *invariant* predictions are obtained by predicting scalar quantities.

## 5  Experiments

Here, we show that CGENNs excel across tasks, attaining top performance in several unique contexts. Parameter budgets as well as training setups are kept as similar as possible to the baseline references. All further experimental details can be found in the public code release.

### 5.1  Estimating Volumetric Quantities

O(3) **Experiment: Signed Volumes**  This task highlights the fact that equivariant architectures based on scalarization are not able to extract some essential geometric properties from input data. In a synthetic setting, we simulate a dataset consisting of random three-dimensional tetrahedra. A main advantage of our method is that it can extract covariant quantities including (among others) *signed volumes*, which we demonstrate in this task. Signed volumes are geometrically significant because they capture the orientation of geometric objects in multidimensional spaces. For instance, in computer graphics, they can determine whether a 3D object is facing towards or away from the camera, enabling proper rendering. The input to the network is the point cloud and the loss function is the mean-squared error between the signed volume and its true value. Note that we are predicting a *covariant* (as opposed to *invariant*) scalar quantity (also known as a *pseudoscalar*) under O(3) transformations using a positive-definite (Euclidean) metric. The results are displayed in the left part of Figure 3. We compare against a standard multilayer perceptron (MLP), an MLP version of the E($n$)-GNN [SHW21] which uses neural networks to update positions with scalar multiplication, *Vector Neurons* (VN) [DLD$^+$21], and *Geometric Vector Perceptrons* (GVP) [JES$^+$21]. We see that the scalarization methods fail to access the features necessary for this task, as evidenced by their test loss not improving even with more available data. The multilayer perceptron, although a universal approximator, lacks the correct inductive biases. Our model, however, has the correct inductive biases (e.g., the equivariance property) and can also access the signed volume. Note that we do not take

the permutation invariance of this task into account, as we are interested in comparing our standard feed-forward architectures against similar baselines.

O(5) **Experiment: Convex Hulls** We go a step further and consider a *five-dimensional* Euclidean space, showcasing our model's ability to generalize to high dimensions. We also make the experiment more challenging by including more points and estimating the volume of the convex hull generated by these points – a task that requires sophisticated algorithms in the classical case. Note that some points may live inside the hull and do not contribute to the volume. We use the same network architectures as before (but now embedded in a five-dimensional space) and present the results in Figure 3. We report the error bars for CGENNs, representing three times the standard deviation of the results of eight runs with varying seeds. Volume (unsigned) is an invariant quantity, enabling the baseline methods to approximate its value. However, we still see that CGENNs outperform the other methods, the only exception being the low-data regime of only 256 available data points. We attribute this to our method being slightly more flexible, making it slightly more prone to overfitting. To mitigate this issue, future work could explore regularization techniques or other methods to reduce overfitting in low-data scenarios.

## 5.2 O(5) **Experiment: Regression**

We compare against the methods presented by [FWW21] who propose an O(5)-invariant regression problem. The task is to estimate the function $f(x_1, x_2) := \sin(\|x_1\|) - \|x_2\|^3/2 + \frac{x_1^\top x_2}{\|x_1\|\|x_2\|}$, where the five-dimensional vectors $x_1, x_2$ are sampled from a standard Gaussian distribution in order to simulate train, test, and validation datasets. The results are shown in Figure 4. We used baselines from [FWW21] including an MLP, MLP with augmentation (MLP+Aug), and the O(5)- & SO(5)-MLP architectures. We maintain the same number of parameters for all data regimes. For extremely small datasets (30 and 100 samples), we observe some overfitting tendencies that can be countered with regularization (e.g., weight decay) or using a smaller model. For higher data regimes (300 samples and onward), CGENNs start to significantly outperform the baselines.

## 5.3 E(3) **Experiment: $n$-Body System**

The $n$-body experiment [KFW$^+$18] serves as a benchmark for assessing the performance of equivariant (graph) neural networks in simulating physical systems [HRXH22]. In this experiment, the dynamics of $n = 5$ charged particles in a three-dimensional space are simulated. Given the initial positions and velocities of these particles, the task is to accurately estimate their positions after 1 000 timesteps. To address this challenge, we construct a graph neural network (GNN) using the Clifford equivariant layers introduced in the previous section. We use a standard message-passing algorithm [GSR$^+$17] where the message and update networks are CGENNs. So long as the message aggregator is equivariant, the end-to-end model also maintains equivariance. The input to the network

| Method | MSE ($\downarrow$) |
|---|---|
| SE(3)-Tr. | 0.0244 |
| TFN | 0.0155 |
| NMP | 0.0107 |
| Radial Field | 0.0104 |
| EGNN | 0.0070 |
| SEGNN | 0.0043 |
| **CGENN** | **0.0039 ± 0.0001** |

Table 1: Mean-squared error (MSE) on the $n$-body system experiment.

consists of the mean-subtracted positions of the particles (to achieve translation invariance) and their velocities. The model's output is the estimated displacement, which is to the input to achieve translation-equivariant estimated target positions. We include the invariant charges as part of the input and their products as edge attributes. We compare against the steerable SE(3)-Transformers [FWFW20], Tensor Field Networks [TSK$^+$18], and SEGNN [BHP$^+$22]. Scalarization baselines include Radial Field [KKN20] and EGNN [SHW21]. Finally, NMP [GSR$^+$17] is not an E(3)-equivariant method. The number of parameters in our model is maintained similar to the EGNN and SEGNN baselines to ensure a fair comparison.

Results of our experiment are presented in Table 1, where we also present for CGENN three times the standard deviation of three identical runs with different seeds. Our approach clearly outperforms earlier methods and is significantly better than [BHP$^+$22], thereby surpassing the baselines. This experiment again demonstrates the advantage of leveraging covariant information in addition to scalar quantities, as it allows for a more accurate representation of the underlying physics and leads to better predictions.

| Model | Accuracy ($\uparrow$) | AUC ($\uparrow$) | $1/\epsilon_B$ ($\uparrow$) ($\epsilon_S = 0.5$) | $1/\epsilon_B$ ($\uparrow$) ($\epsilon_S = 0.3$) |
|---|---|---|---|---|
| ResNeXt [XGD$^+$17] | 0.936 | 0.9837 | 302 | 1147 |
| P-CNN [CMS17] | 0.930 | 0.9803 | 201 | 759 |
| PFN [KMT19] | 0.932 | 0.9819 | 247 | 888 |
| ParticleNet [QG20] | 0.940 | 0.9858 | 397 | 1615 |
| EGNN [SHW21] | 0.922 | 0.9760 | 148 | 540 |
| LGN [BAO$^+$20] | 0.929 | 0.9640 | 124 | 435 |
| LorentzNet [GMZ$^+$22] | **0.942** | **0.9868** | **498** | **2195** |
| CGENN | **0.942** | **0.9869** | **500** | **2172** |

Table 2: Performance comparison between our proposed method and alternative algorithms on the top tagging experiment. We present the accuracy, Area Under the Receiver Operating Characteristic Curve (AUC), and background rejection $1/\epsilon_B$ and at signal efficiencies of $\epsilon_S = 0.3$ and $\epsilon_S = 0.5$.

### 5.4 $\mathrm{O}(1,3)$ **Experiment: Top Tagging**

Jet tagging in collider physics is a technique used to identify and categorize high-energy jets produced in particle collisions, as measured by, e.g., CERN's ATLAS detector [BTB$^+$08]. By combining information from various parts of the detector, it is possible to trace back these jets' origins [KNS$^+$19, ATL17]. The current experiment seeks to tag jets arising from the heaviest particles of the standard model: the 'top quarks' [IQWW09]. A jet tag should be invariant with respect to the global reference frame, which can transform under *Lorentz boosts* due to the relativistic nature of the particles. A Lorentz boost is a transformation that relates the space and time coordinates of an event as seen from two inertial reference frames. The defining characteristic of these transformations is that they preserve the *Minkowski metric*, which is given by $\gamma(ct, x, y, z) := (ct)^2 - x^2 - y^2 - z^2$. Note the difference with the standard positive definite Euclidean metric, as used in the previous experiments. The set of all such transformations is captured by the orthogonal group $\mathrm{O}(1,3)$; therefore, our method is fully compatible with modeling this problem.

We evaluate our model on a top tagging benchmark published by [KPTR19]. It contains 1.2M training entries, 400k validation entries, and 400k testing entries. For each jet, the energy-momentum 4-vectors are available for up to 200 constituent particles, making this a much larger-scale experiment than the ones presented earlier. Again, we employ a standard message passing graph neural network [GSR$^+$17] using CGENNs as message and update networks. The baselines include ResNeXt [XGD$^+$17], P-CNN [CMS17], PFN [KMT19], ParticleNet [QG20], LGN [BAO$^+$20], EGNN [SHW21], and the more recent LorentzNet [GMZ$^+$22]. Among these, LGN is a steerable method, whereas EGNN and LorentzNet are scalarization methods. The other methods are not Lorentz-equivariant. Among the performance metrics, there are classification accuracy, Area Under the Receiver Operating Characteristic Curve (AUC), and the background rejection rate $1/\epsilon_B$ at signal efficiencies of $\epsilon_S = 0.3$ and $\epsilon_S = 0.5$, where $\epsilon_B$ and $\epsilon_S$ are the false positive and true positive rates, respectively. We observe that LorentzNet, a method that uses invariant quantities, is an extremely competitive baseline that was optimized for this task. Despite this, CGENNs are able to match its performance while maintaining the same core implementation.

## 6 Conclusion

We presented a novel approach for constructing $\mathrm{O}(n)$- and $\mathrm{E}(n)$-equivariant neural networks based on Clifford algebras. After establishing the required theoretical results, we proposed parameterizations of nonlinear multivector-valued maps that exhibit versatility and applicability across scenarios varying in dimension. This was achieved by the core insight that polynomials in multivectors are $\mathrm{O}(n)$-equivariant functions. Theoretical results were empirically substantiated in three distinct experiments, outperforming or matching baselines that were sometimes specifically designed for these tasks.

CGENNs induce a (non-prohibitive) degree of computational overhead similar to other steerable methods. On the plus side, we believe that improved code implementations such as custom GPU kernels or alternative parameterizations to the current ones can significantly alleviate this issue, potentially also resulting in improved performances on benchmark datasets. This work provides solid theoretical and experimental foundations for such developments.

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

## Acknowledgments

We thank Leo Dorst and Steven de Keninck for insightful discussions about Clifford (geometric) algebras and their applications as well as pointing us to the relevant literature. Further, we thank Robert-Jan Schlimbach and Jayesh Gupta for their help with computational infrastructure and scaling up the experiments. This work used the Dutch national e-infrastructure with the support of the SURF Cooperative using grant no. EINF-5757 as well as the DAS-5 and DAS-6 clusters [BEdL$^+$16].

## Broader Impact Statement

The advantages of more effective equivariant graph neural networks could be vast, particularly in the physical sciences. Improved equivariant neural networks could revolutionize fields like molecular biology, astrophysics, and materials science by accurately predicting system behaviors under various transformations. They could further enable more precise simulations of physical systems. Such new knowledge could drive forward scientific discovery, enabling innovations in healthcare, materials engineering, and our understanding of the universe. However, a significant drawback of relying heavily on advanced equivariant neural networks in physical sciences is the potential for unchecked errors or biases to propagate when these systems are not thoroughly cross-validated using various methods. This could potentially lead to flawed scientific theories or dangerous real-world applications.

## Reproducibility Statement

We have released the code to reproduce the experimental results, including hyperparameters, network architectures, and so on, at https://github.com/DavidRuhe/clifford-group-equivariant-neural-networks.

## Computational Resources

The volumetric quantities and regression experiments were carried out on $1 \times 11$ GB *NVIDIA GeForce GTX 1080 Ti* and $1 \times 11$ GB *NVIDIA GeForce GTX 2080 Ti* instances. The $n$-body experiment ran on $1 \times 24$ GB *NVIDIA GeForce RTX 3090* and $1 \times 24$ GB *NVIDIA RTX A5000* nodes. Finally, the top tagging experiment was conducted on $4 \times 40$ GB *NVIDIA Tesla A100 Ampere* instances.

## Licenses

We would like to thank the scientific software development community, without whom this work would not be possible.

This work made use of `Python` (PSF License Agreement), `NumPy` [VDWCV11] (BSD-3 Clause), `PyTorch` [PGM$^+$19] (BSD-3 Clause), `CUDA` (proprietary license), *Weights and Biases* [Bie20] (MIT), *Scikit-Learn* [PVG$^+$11] (BSD-3 Clause), *Seaborn* [Was21] (BSD-3 Clause), *Matplotlib* [Hun07] (PSF), [VGO$^+$20] (BSD-3).

The top tagging dataset is licensed under CC-By Attribution 4.0 International.

# Contents

# A  Glossary

| Notation | Meaning |
| --- | --- |
| O(n) | The orthogonal group acting on an $n$-dimensional vector space. |
| SO$(n)$ | The special orthogonal group acting on an $n$-dimensional vector space. |
| E$(n)$ | The Euclidean group acting on an $n$-dimensional vector space. |
| SE$(n)$ | The special Euclidean group acting on an $n$-dimensional vector space. |
| GL$(n)$ | The general linear group acting on an $n$-dimensional vector space. |
| O$(V, \mathfrak{q})$ | The orthogonal group of vector space $V$ with respect to a quadratic form $\mathfrak{q}$. |
| O$_\mathcal{R}(V, \mathfrak{q})$ | The orthogonal group of vector space $V$ with respect to a quadratic form $\mathfrak{q}$ that acts as the identity on $\mathcal{R}$. |
| $V$ | A vector space. |
| $\mathcal{R}$ | The radical vector subspace of $V$ such that for any $f \in \mathcal{R}$, $\mathfrak{b}(f, v) = 0$ for all $v \in V$. |
| $\mathbb{F}$ | A field. |
| $\mathbb{R}$ | The real numbers. |
| $\mathbb{N}$ | The natural numbers. |
| $\mathbb{Z}$ | The integers. |
| $[n]$ | The set of integers $\{1, \ldots, n\}$. |
| $\mathfrak{q}$ | A quadratic form, $\mathfrak{q} : V \to \mathbb{F}$. |
| $\mathfrak{b}$ | A bilinear form, $\mathfrak{b} : V \times V \to \mathbb{F}$. |
| Cl$(V, \mathfrak{q})$ | The Clifford algebra over a vector space $V$ with quadratic form $\mathfrak{q}$. |
| Cl$^\times(V, \mathfrak{q})$ | The group of invertible Clifford algebra elements. |
| Cl$^{[\times]}(V, \mathfrak{q})$ | The group of invertible parity-homogeneous Clifford algebra elements. Cl$^{[\times]}(V, \mathfrak{q}) := \{w \in \text{Cl}^\times(V, \mathfrak{q}) \mid \eta(w) \in \{\pm 1\}\}$. |
| Cl$^{[0]}(V, \mathfrak{q})$ | The even subalgebra of the Clifford algebra. Cl$^{[0]}(V, \mathfrak{q}) := \bigoplus_{m \text{ even}}^n \text{Cl}^{(m)}(V, \mathfrak{q})$. |
| Cl$^{[1]}(V, \mathfrak{q})$ | The odd part of the Clifford algebra. Cl$^{[1]}(V, \mathfrak{q}) := \bigoplus_{m \text{ odd}}^n \text{Cl}^{(m)}(V, \mathfrak{q})$. |
| Cl$^{(m)}(V, \mathfrak{q})$ | The grade-$m$ subspace of the Clifford algebra. |
| $(\_)^{(m)}$ | Grade projection, $(\_)^{(m)} : \text{Cl}(V, \mathfrak{q}) \to \text{Cl}^{(m)}(V, \mathfrak{q})$. |
| $\zeta$ | Projection onto the zero grade, $\zeta : \text{Cl}(V, \mathfrak{q}) \to \text{Cl}^{(0)}(V, \mathfrak{q})$. |
| $e_i$ | A basis vector, $e_i \in V$. |
| $f_i$ | A basis vector of $\mathcal{R}$. |
| $e_A$ | A Clifford basis element (product of basis vectors) with $e_A \in \text{Cl}(V, \mathfrak{q})$, $A \subseteq \{1, \ldots, n\}$. |
| $\bar{\mathfrak{b}}$ | The extended bilinear form on the Clifford algebra, $\bar{\mathfrak{b}} : \text{Cl}(V, \mathfrak{q}) \times \text{Cl}(V, \mathfrak{q}) \to \mathbb{F}$. |
| $\bar{\mathfrak{q}}$ | The extended quadratic form on the Clifford algebra, $\bar{\mathfrak{q}} : \text{Cl}(V, \mathfrak{q}) \to \mathbb{F}$. |
| $\alpha$ | Clifford main involution $\alpha : \text{Cl}(V, \mathfrak{q}) \to \text{Cl}(V, \mathfrak{q})$. |
| $\beta$ | Main Clifford anti-involution, also known as *reversion*, $\beta : \text{Cl}(V, \mathfrak{q}) \to \text{Cl}(V, \mathfrak{q})$. |
| $\gamma$ | Clifford conjugation $\gamma : \text{Cl}(V, \mathfrak{q}) \to \text{Cl}(V, \mathfrak{q})$. |
| $\eta$ | Coboundary of $\alpha$. For $w \in \text{Cl}^\times(V, \mathfrak{q})$, $\eta(w) \in \{\pm 1\}$ if and only if $w$ is parity-homogeneous. |
| $\rho(w)$ | The (adjusted) twisted conjugation, used as the action of the Clifford group. $\rho(w) : \text{Cl}(V, \mathfrak{q}) \to \text{Cl}(V, \mathfrak{q})$, $w \in \text{Cl}^\times(V, \mathfrak{q})$. |
| $\Gamma(V, \mathfrak{q})$ | The Clifford group of Cl$(V, \mathfrak{q})$. |

| Notation | Meaning |
| --- | --- |
| $\Gamma^{[0]}(V, \mathfrak{q})$ | The special Clifford group. It excludes orientation-reversing (odd) elements. $\Gamma^{[0]}(V, \mathfrak{q}) := \Gamma(V, \mathfrak{q}) \cap \mathrm{Cl}^{[0]}(V, \mathfrak{q})$. |
| $\bigwedge(\mathcal{R})$ | The radical subalgebra of $\mathrm{Cl}(V, \mathfrak{q})$. I.e., those elements that have zero $\bar{\mathfrak{q}}$. It is equal to the exterior algebra of $\mathcal{R}$. |
| $\bigwedge^{[i]}(\mathcal{R})$ | The even or odd ($i \in \{0, 1\}$) subalgebra of $\bigwedge(\mathcal{R})$. $\bigwedge^{[i]}(\mathcal{R}) := \bigwedge(\mathcal{R}) \cap \mathrm{Cl}^{[i]}(V, \mathfrak{q})$. |
| $\bigwedge^{(\geq 1)}(\mathcal{R})$ | The subalgebra of $\bigwedge(\mathcal{R})$ with grade greater than or equal to one. $\bigwedge^{(\geq 1)}(\mathcal{R}) := \mathrm{span}\{f_1 \ldots f_k \mid k \geq 1, f_l \in \mathcal{R}\}$. |
| $\bigwedge^{\times}(\mathcal{R})$ | The group of invertible elements of $\bigwedge(\mathcal{R})$. $\bigwedge^{\times}(\mathcal{R}) = \mathbb{F}^{\times} + \bigwedge^{(\geq 1)}(\mathcal{R})$. |
| $\bigwedge^{[\times]}(\mathcal{R})$ | The group of invertible elements of $\bigwedge(\mathcal{R})$ with even grades. $\bigwedge^{[\times]}(\mathcal{R}) := \mathbb{F}^{\times} + \mathrm{span}\{f_1 \ldots f_k \mid k \geq 2 \text{ even}, f_l \in \mathcal{R}\}$. |
| $\bigwedge^{*}(\mathcal{R})$ | Same as $\bigwedge^{\times}(\mathcal{R})$, but with $\mathbb{F}^{\times}$ set to 1. |
| $\bigwedge^{[*]}(\mathcal{R})$ | Same as $\bigwedge^{[\times]}(\mathcal{R})$, but with $\mathbb{F}^{\times}$ set to 1. |
| SN | Spinor norm, $\mathrm{SN} : \mathrm{Cl}(V, \mathfrak{q}) \to \mathbb{F}$. |
| CN | Clifford norm, $\mathrm{CN} : \mathrm{Cl}(V, \mathfrak{q}) \to \mathbb{F}$. |

# B Supplementary Material: Introduction

Existing literature on Clifford algebra uses varying notations, conventions, and focuses on several different applications. We gathered previous definitions and included independently derived results to achieve the desired outcomes for this work and to provide proofs for the most general cases, including, e.g., potential degeneracy of the metric. As such, this appendix acts as a comprehensive resource on quadratic spaces, orthogonal groups, Clifford algebras, their constructions, and specific groups represented within Clifford algebras.

We start in Appendix C with a primer to quadratic spaces and the orthogonal group. In particular, we specify how the orthogonal group is defined on a quadratic space, and investigate its action. We pay special attention to the case of spaces with degenerate quadratic forms, and what we call "radical-preserving" orthogonal automorphisms. The section concludes with the presentation of the Cartan-Dieudonné theorem.

In Appendix D, we introduce the definition of the Clifford algebra as a quotient of the tensor algebra. We investigate several key properties of the Clifford algebra. Then, we introduce its parity grading, which in turn is used to prove that the dimension of the algebra is $2^n$, where $n$ is the dimension of the vector space on which it is defined. This allows us to construct an algebra basis. Subsequently, we extend the quadratic and bilinear forms of the vector space to their Clifford algebra counterparts. This then leads to the construction of an orthogonal basis for the algebra. Furthermore, the multivector grading of the algebra is shown to be basis independent, leading to the orthogonal sum decomposition into the usual subspaces commonly referred to as scalars, vectors, bivectors, and so on. Finally, we investigate some additional properties of the algebra, such as its center, its radical subalgebra, and its twisted center.

In Appendix E we introduce, motivate, and adjust the twisted conjugation map. We show that it comprises an algebra automorphism, thereby respecting the algebra's vector space and also its multiplicative properties. Moreover, it can serve as a representation of the Clifford algebra's group of invertible elements. We then identify what we call the Clifford group, whose action under the twisted conjugation respects also the multivector decomposition. We investigate properties of the Clifford group and its action. Specifically, we show that its action yields a radical-preserving orthogonal automorphism. Moreover, it acts orthogonally on each individual subspace.

Finally, after introducing the Spinor and Clifford norm, the section concludes with the pursuit of a general definition for the Pin and Spin group, which are used in practice more often than the Clifford group. This, however, turns out to be somewhat problematic when generalizing to fields beyond $\mathbb{R}$. We motivate several choices, and outline their (dis)advantages. For $\mathbb{R}$, we finally settle on definitions that are compatible with existing literature.

# C Quadratic Vector Spaces and the Orthogonal Group

We provide a short introduction to quadratic spaces and define the orthogonal group of a (non-definite) quadratic space. We use these definitions in our analysis of how the Clifford group relates to the orthogonal group.

We will always denote with $\mathbb{F}$ a *field* of a characteristic different from 2, $\mathrm{char}(\mathbb{F}) \neq 2$. Sometimes we will specialize to the real numbers $\mathbb{F} = \mathbb{R}$. Let $V$ be a *vector space* over $\mathbb{F}$ of finite dimension $\dim_{\mathbb{F}} V = n$. We will follow [Ser12].

**Definition C.1** (Quadratic forms and quadratic vector spaces). *A map* $\mathfrak{q} : V \to \mathbb{F}$ *will be called a* quadratic form *of $V$ if for all $c \in \mathbb{F}$ and $v \in V$:*

$$\mathfrak{q}(c \cdot v) = c^2 \cdot \mathfrak{q}(v), \tag{17}$$

*and if:*

$$\mathfrak{b}(v_1, v_2) := \frac{1}{2} \left( \mathfrak{q}(v_1 + v_2) - \mathfrak{q}(v_1) - \mathfrak{q}(v_2) \right), \tag{18}$$

*is a bilinear form over $\mathbb{F}$ in $v_1, v_2 \in V$, i.e., it is separately $\mathbb{F}$-linear in each of the arguments $v_1$ and $v_2$ when the other one is fixed.*

*The tuple $(V, \mathfrak{q})$ will then be called a* quadratic (vector) space*.*

**Remark C.2.**     *1. Note that we explicitely do not make assumptions about the non-degeneracy of $\mathfrak{b}$. Even the extreme case with constant $\mathfrak{q} = 0$ is allowed and of interest.*

2. *Further note, that $\mathfrak{b}$ will automatically be a symmetric bilinear form.*

**Definition C.3** (The radical subspace). *Now consider the quadratic space $(V, \mathfrak{q})$. We then call the subspace:*

$$\mathcal{R} := \{ f \in V \,|\, \forall v \in V. \, \mathfrak{b}(f, v) = 0 \}, \tag{19}$$

*the* radical subspace *of $(V, \mathfrak{q})$.*

**Remark C.4.**     *1. The radical subspace of $(V, \mathfrak{q})$ is the biggest subspace of $V$ where $\mathfrak{q}$ is degenerate. Note that this space is orthogonal to all other subspaces of $V$.*

2. *If $W$ is any complementary subspace of $\mathcal{R}$ in $V$, so that $V = \mathcal{R} \oplus^{\perp} W$, then $\mathfrak{q}$ restricted to $W$ is non-degenerate.*

**Definition C.5** (Orthogonal basis). *A basis $e_1, \ldots, e_n$ of $V$ is called* orthogonal basis *of $V$ if for all $i \neq j$ we have:*

$$\mathfrak{b}(e_i, e_j) = 0. \tag{20}$$

*It is called an* orthonormal basis *if, in addition, $\mathfrak{q}(e_i) \in \{-1, 0, +1\}$ for all $i = 1, \ldots, n$.*

**Remark C.6.** *Note that every quadratic space $(V, \mathfrak{q})$ has an orthogonal basis by [Ser12] p. 30 Thm. 1, but not necessarily a orthonormal basis. However, if $\mathbb{F} = \mathbb{R}$ then $(V, \mathfrak{q})$ has an orthonormal basis by Sylvester's law of inertia, see [Syl52].*

**Definition C.7** (The orthogonal group). *For a quadratic space $(V, \mathfrak{q})$ we define the* orthogonal group *of $(V, \mathfrak{q})$ as follows:*

$$\mathrm{O}(\mathfrak{q}) := \mathrm{O}(V, \mathfrak{q}) := \left\{ \Phi : V \to V \,\middle|\, \Phi \; \mathbb{F}\text{-}linear\; automorphism^8, \; s.t. \; \forall v \in V. \, \mathfrak{q}(\Phi(v)) = \mathfrak{q}(v) \right\}. \tag{21}$$

*If $(V, \mathfrak{q}) = \mathbb{R}^{(p,q,r)}$, i.e. if $\mathbb{F} = \mathbb{R}$ and $V = \mathbb{R}^n$ and $\mathfrak{q}$ has the signature $(p, q, r)$ with $p + q + r = n$ then we define the* group of orthogonal matrices of signature $(p, q, r)$ *as follows:*

$$\mathrm{O}(p, q, r) := \left\{ O \in \mathrm{GL}(n) \,\middle|\, O^{\top} \Delta_{(p,q,r)} O = \Delta_{(p,q,r)} \right\}, \tag{22}$$

*where we used the $(n \times n)$-diagonal signature matrix:*

$$\Delta_{(p,q,r)} := \mathrm{diag}(\underbrace{+1, \ldots, +1}_{p\text{-}times}, \underbrace{-1, \ldots, -1}_{q\text{-}times}, \underbrace{0, \ldots, 0}_{r\text{-}times}). \tag{23}$$

**Theorem C.8** (See [YCo20]). *Let $(V, \mathfrak{q})$ be a finite dimensional quadratic space over a field $\mathbb{F}$ with $\mathrm{char}(\mathbb{F}) \neq 2$. Let $\mathcal{R} \subseteq V$ be the radical subspace, $r := \dim \mathcal{R}$, and, $W \subseteq V$ a complementary subspace, $m := \dim W$. Then we get an isomorphism:*

$$\mathrm{O}(V, \mathfrak{q}) \cong \begin{pmatrix} \mathrm{O}(W, \mathfrak{q}|_W) & 0_{m \times r} \\ \mathrm{M}(r, m) & \mathrm{GL}(r) \end{pmatrix} = \left\{ \begin{pmatrix} O & 0_{m \times r} \\ M & G \end{pmatrix} \middle| O \in \mathrm{O}(W, \mathfrak{q}|_W), M \in \mathrm{M}(r, m), G \in \mathrm{GL}(r) \right\}, \tag{24}$$

*where $\mathrm{M}(r, m) := \mathbb{F}^{r \times m}$ is the additive group of all $(r \times m)$-matrices with coefficients in $\mathbb{F}$ and where $\mathrm{GL}(r)$ is the multiplicative group of all invertible $(r \times r)$-matrices with coefficients in $\mathbb{F}$.*

*Proof.* Let $e_1, \dots, e_m$ be an orthogonal basis for $W$ and $f_1, \dots, f_r$ be a basis for $\mathcal{R}$, then the associated bilinear form $\mathfrak{b}$ of $\mathfrak{q}$ has the following matrix representation:

$$\begin{pmatrix} Q & 0 \\ 0 & 0 \end{pmatrix}, \tag{25}$$

with an invertible diagonal $(m \times m)$-matrix $Q$. For the matrix of any orthogonal automorphism $\Phi$ of $V$ we get the necessary and sufficient condition:

$$\begin{pmatrix} Q & 0 \\ 0 & 0 \end{pmatrix} \overset{!}{=} \begin{pmatrix} A^\top & C^\top \\ B^\top & D^\top \end{pmatrix} \begin{pmatrix} Q & 0 \\ 0 & 0 \end{pmatrix} \begin{pmatrix} A & B \\ C & D \end{pmatrix} \tag{26}$$

$$= \begin{pmatrix} A^\top Q A & A^\top Q B \\ B^\top Q A & B^\top Q B \end{pmatrix}. \tag{27}$$

This is equivalent to the conditions:

$$Q = A^\top Q A, \qquad\qquad 0 = A^\top Q B, \qquad\qquad 0 = B^\top Q B. \tag{28}$$

This shows that $A$ is the matrix of an orthogonal automorphism of $W$ (w.r.t. $\mathfrak{b}|_W$), which, since $Q$ is invertible, is also invertible. The second equation then shows that necessarily $B = 0$. Since the whole matrix needs to be invertible also $D$ must be invertible. Furthermore, there are no constraints on $C$.

If all those conditions are satisfied the whole matrix satisfies the orthogonality constraints from above. $\qquad\square$

**Corollary C.9.** *For $\mathbb{R}^{(p,q,r)}$ we get:*

$$\mathrm{O}(p, q, r) \cong \begin{pmatrix} \mathrm{O}(p, q) & 0_{(p+q) \times r} \\ \mathrm{M}(r, p+q) & \mathrm{GL}(r) \end{pmatrix}, \tag{29}$$

*where $\mathrm{O}(p, q) := \mathrm{O}(p, q, 0)$.*

**Remark C.10.** *Note that the composition $\Phi_1 \circ \Phi_2$ of orthogonal automorphisms of $(V, \mathfrak{q})$ corresponds to the matrix multiplication as follows:*

$$\begin{pmatrix} O_1 & 0 \\ M_1 & G_1 \end{pmatrix} \begin{pmatrix} O_2 & 0 \\ M_2 & G_2 \end{pmatrix} = \begin{pmatrix} O_1 O_2 & 0 \\ M_1 O_2 + G_1 M_2 & G_1 G_2 \end{pmatrix}. \tag{30}$$

**Definition C.11** (Radical preserving orthogonal automorphisms). *For a quadratic space $(V, \mathfrak{q})$ with radical subspace $\mathcal{R} \subseteq V$ we define the group of* radical preserving orthogonal automorphisms *as follows:*

$$\mathrm{O}_{\mathcal{R}}(V, \mathfrak{q}) := \{ \Phi \in \mathrm{O}(V, \mathfrak{q}) \mid \Phi|_{\mathcal{R}} = \mathrm{id}_{\mathcal{R}} \}. \tag{31}$$

*If $(V, \mathfrak{q}) = \mathbb{R}^{(p,q,r)}$, i.e. if $\mathbb{F} = \mathbb{R}$ and $V = \mathbb{R}^n$ and $\mathfrak{q}$ has the signature $(p, q, r)$ with $p + q + r = n$ then we define the* group of radical preserving orthogonal matrices of signature $(p, q, r)$ *as follows:*

$$\mathrm{O}_{\mathcal{R}}(p, q, r) := \{ O \in \mathrm{O}(p, q, r) \mid \textit{bottom right corner of } O = I_r \} = \begin{pmatrix} \mathrm{O}(p, q) & 0_{(p+q) \times r} \\ \mathrm{M}(r, p+q) & I_r \end{pmatrix}, \tag{32}$$

*where $I_r$ is the $(r \times r)$-identity matrix.*

**Remark C.12.** *Note that the matrix representation of $\Phi \in \mathrm{O}_{\mathcal{R}}(\mathfrak{q})$ w.r.t. an orthogonal basis like in the proof of theorem C.8 is of the form:*

$$\begin{pmatrix} O & 0_{m \times r} \\ M & I_r \end{pmatrix}, \tag{33}$$

*with $O \in \mathrm{O}(W, \mathfrak{q}|_W)$, $M \in \mathrm{M}(r, m)$ and where $I_r$ is the $(r \times r)$-identity matrix.*

*The composition $\Phi_1 \circ \Phi_2$ of $\Phi_1$ and $\Phi_2 \in \mathrm{O}_{\mathcal{R}}(\mathfrak{q})$ is then given by the corresponding matrix multiplication:*

$$\begin{pmatrix} O_1 & 0 \\ M_1 & I_r \end{pmatrix} \begin{pmatrix} O_2 & 0 \\ M_2 & I_r \end{pmatrix} = \begin{pmatrix} O_1 O_2 & 0 \\ M_1 O_2 + M_2 & I_r \end{pmatrix}. \tag{34}$$

*By observing the left column (the only part that does not transform trivially), we see that $\mathrm{O}(\mathfrak{q}|_W)$ acts on $\mathrm{M}(r, m)$ just by matrix multiplication from the right (in the corresponding basis):*

$$(O_1, M_1) \cdot (O_2, M_2) = (O_1 O_2, M_1 O_2 + M_2). \tag{35}$$

*This immediately shows that we can write $\mathrm{O}_{\mathcal{R}}(\mathfrak{q})$ as the semi-direct product:*

$$\mathrm{O}_{\mathcal{R}}(\mathfrak{q}) \cong \mathrm{O}(\mathfrak{q}|_W) \ltimes \mathrm{M}(r, m). \tag{36}$$

*In the special case of $\mathbb{R}^{(p,q,r)}$ we get:*

$$\mathrm{O}_{\mathcal{R}}(p, q, r) \cong \mathrm{O}(p, q) \ltimes \mathrm{M}(r, p + q). \tag{37}$$

We conclude this chapter by citing the Theorem of Cartan and Dieudonné about the structure of orthogonal groups in the non-degnerate case, but still for arbitrary fields $\mathbb{F}$ with $\mathrm{char}(\mathbb{F}) \neq 2$.

**Theorem C.13** (Theorem of Cartan-Dieudonné, see [Art57] Thm. 3.20)**.** *Let $(V, \mathfrak{q})$ be a non-degenerate quadratic space of finite dimension $\dim V = n < \infty$ over a field $\mathbb{F}$ of $\mathrm{char}(\mathbb{F}) \neq 2$. Then every element $g \in \mathrm{O}(V, \mathfrak{q})$ can be written as:*

$$g = r_1 \circ \cdots \circ r_k, \tag{38}$$

*with $1 \leq k \leq n$, where $r_i$ are reflections w.r.t. non-singular hyperplanes.*

# D  The Clifford Algebra and Typical Constructions

In this section, we provide the required definitions, constructions, and derivations leading to the results stated in the main paper. We start with a general introduction to the Clifford algebra.

## D.1  The Clifford Algebra and its Universal Property

We follow [LS09, Cru90].

Let $(V, \mathfrak{q})$ be a finite dimensional *quadratic space* over a field $\mathbb{F}$ (also denoted an $\mathbb{F}$-vector space) with $\mathrm{char}(\mathbb{F}) \neq 2$. We abbreviate the corresponding *bilinear form* $\mathfrak{b}$ on vectors $v_1, v_2 \in V$ as follows:

$$\mathfrak{b}(v_1, v_2) := \frac{1}{2} \left( \mathfrak{q}(v_1 + v_2) - \mathfrak{q}(v_1) - \mathfrak{q}(v_2) \right). \tag{39}$$

**Definition D.1.** *To define the* Clifford algebra $\mathrm{Cl}(V, \mathfrak{q})$ *we first consider the* tensor algebra *of $V$:*

$$\mathrm{T}(V) := \bigoplus_{m=0}^{\infty} V^{\otimes m} = \mathrm{span}\left\{ v_1 \otimes \cdots \otimes v_m \,\middle|\, m \geq 0, v_i \in V \right\}, \tag{40}$$

$$V^{\otimes m} := \underbrace{V \otimes \cdots \otimes V}_{m\text{-times}}, \qquad\qquad V^{\otimes 0} := \mathbb{F}, \tag{41}$$

*and the following two-sided ideal*[9]:

$$I(\mathfrak{q}) := \left\langle v \otimes v - \mathfrak{q}(v) \cdot 1_{\mathrm{T}(V)} \,\middle|\, v \in V \right\rangle. \tag{42}$$

*Then we define the* Clifford algebra $\mathrm{Cl}(V, \mathfrak{q})$ *as the following quotient:*

$$\mathrm{Cl}(V, \mathfrak{q}) := \mathrm{T}(V)/I(\mathfrak{q}). \tag{43}$$

*In words, we identify the square of a vector with its quadratic form. We also denote the canonical algebra quotient map as:*

$$\pi : \mathrm{T}(V) \to \mathrm{Cl}(V, \mathfrak{q}). \tag{44}$$

**Remark D.2.**  *1. It is not easy to see, but always true, that $\dim \mathrm{Cl}(V, \mathfrak{q}) = 2^n$, where $n := \dim V$, see Theorem D.15.*

  *2. If $e_1, \ldots, e_n$ is any basis of $(V, \mathfrak{q})$ then $(e_A)_{A \subseteq [n]}$ is a basis for $\mathrm{Cl}(V, \mathfrak{q})$, where we put for a subset $A \subseteq [n] := \{1, \ldots, n\}$:*

$$e_A := \prod_{i \in A}^{<} e_i, \qquad\qquad e_\emptyset := 1_{\mathrm{Cl}(V, \mathfrak{q})}. \tag{45}$$

  *where the product is taken in increasing order of the indices $i \in A$, see Corollary D.16.*

  *3. If $e_1, \ldots, e_n$ is any orthogonal basis of $(V, \mathfrak{q})$, then one can even show that $(e_A)_{A \subseteq [n]}$ is an orthogonal basis for $\mathrm{Cl}(V, \mathfrak{q})$ w.r.t. an extension of the bilinear form $\mathfrak{b}$ from $V$ to $\mathrm{Cl}(V, \mathfrak{q})$, see Theorem D.26.*

**Lemma D.3** (The fundamental identity). *Note that for $v_1, v_2 \in V$, we always have the fundamental identity in $\mathrm{Cl}(V, \mathfrak{q})$:*

$$v_1 v_2 + v_2 v_1 = 2\mathfrak{b}(v_1, v_2). \tag{46}$$

*Proof.* By definition of the Clifford algebra we have the identities:

$$\mathfrak{q}(v_1) + v_1 v_2 + v_2 v_1 + \mathfrak{q}(v_2) = v_1 v_1 + v_1 v_2 + v_2 v_1 + v_2 v_2 \tag{47}$$

$$= (v_1 + v_2)(v_1 + v_2) \tag{48}$$

$$= \mathfrak{q}(v_1 + v_2) \tag{49}$$

$$= \mathfrak{b}(v_1 + v_2, v_1 + v_2) \tag{50}$$

$$= \mathfrak{b}(v_1, v_1) + \mathfrak{b}(v_1, v_2) + \mathfrak{b}(v_2, v_1) + \mathfrak{b}(v_2, v_2) \tag{51}$$

$$= \mathfrak{q}(v_1) + 2\mathfrak{b}(v_1, v_2) + \mathfrak{q}(v_2). \tag{52}$$

Substracting $\mathfrak{q}(v_1) + \mathfrak{q}(v_2)$ on both sides gives the claim.  $\square$

---

[9]The ideal ensures that for all elements containing $v \otimes v$ (i.e., linear combinations, multiplications on the left, and multiplications on the right), $v \otimes v$ is identified with $\mathfrak{q}(v)$; e.g. $x \otimes v \otimes v \otimes y \sim x \otimes (\mathfrak{q}(v) \cdot 1_{\mathrm{T}(V)}) \otimes y$ for every $x, y \in \mathrm{T}(V)$.

Further, the Clifford algebra $\mathrm{Cl}(V, \mathfrak{q})$ is fully characterized by the following property.

**Theorem D.4** (The universal property of the Clifford algebra)**.** *For every* $\mathbb{F}$*-algebra*[10] *(an algebra over field* $\mathbb{F}$*)* $\mathcal{A}$ *and every* $\mathbb{F}$*-linear map* $f : V \to \mathcal{A}$ *such that for all* $v \in V$ *we have:*

$$f(v)^2 = \mathfrak{q}(v) \cdot 1_{\mathcal{A}}, \tag{53}$$

*there exists a unique* $\mathbb{F}$*-algebra homomorphism*[11] $\bar{f} : \mathrm{Cl}(V, \mathfrak{q}) \to \mathcal{A}$ *such that* $\bar{f}(v) = f(v)$ *for all* $v \in V$.

*More explicitely, if* $f$ *satisfies equation 53 and* $x \in \mathrm{Cl}(V, \mathfrak{q})$. *Then we can take* any *representation of* $x$ *of the following form:*

$$x = c_0 + \sum_{i \in I} c_i \cdot v_{i,1} \cdots v_{i,k_i}, \tag{54}$$

*with finite index sets* $I$ *and* $k_i \in \mathbb{N}$ *and coefficients* $c_0, c_i \in \mathbb{F}$ *and vectors* $v_{i,j} \in V$, $j = 1, \ldots, k_i$, $i \in I$, *and, then we can compute* $\bar{f}(x)$ *by the following formula (without ambiguity):*

$$\bar{f}(x) = c_0 \cdot 1_{\mathcal{A}} + \sum_{i \in I} c_i \cdot f(v_{i,1}) \cdots f(v_{i,k_i}). \tag{55}$$

*In the following, we will often denote* $\bar{f}$ *again with* $f$ *without further indication.*

## D.2 The Multivector Filtration and the Grade

Note that the Clifford algebra is a filtered algebra.

**Definition D.5.** *We define the* multivector filtration[12] *of* $\mathrm{Cl}(V, \mathfrak{q})$ *for grade* $m \in \mathbb{N}_0$ *as follows:*

$$\mathrm{Cl}^{(\leq m)}(V, \mathfrak{q}) := \pi \left( \mathrm{T}^{(\leq m)}(V) \right), \qquad \mathrm{T}^{(\leq m)}(V) := \bigoplus_{l=0}^{m} V^{\otimes l}. \tag{56}$$

**Remark D.6.** *Note that we really get a filtration on the space* $\mathrm{Cl}(V, \mathfrak{q})$:

$$\mathbb{F} = \mathrm{Cl}^{(\leq 0)}(V, \mathfrak{q}) \subseteq \mathrm{Cl}^{(\leq 1)}(V, \mathfrak{q}) \subseteq \mathrm{Cl}^{(\leq 2)}(V, \mathfrak{q}) \subseteq \ldots \subseteq \mathrm{Cl}^{(\leq n)}(V, \mathfrak{q}) = \mathrm{Cl}(V, \mathfrak{q}). \tag{57}$$

*Furthermore, note that this is compatible with the algebra structure of* $\mathrm{Cl}(V, \mathfrak{q})$, *i.e. for* $i, j \geq 0$ *we get:*

$$x \in \mathrm{Cl}^{(\leq i)}(V, \mathfrak{q}) \quad \wedge \quad y \in \mathrm{Cl}^{(\leq j)}(V, \mathfrak{q}) \qquad \Longrightarrow \qquad xy \in \mathrm{Cl}^{(\leq i+j)}(V, \mathfrak{q}). \tag{58}$$

*Together with the equality:* $\pi(\mathrm{T}^{(\leq m)}(V)) = \mathrm{Cl}^{(\leq m)}(V, \mathfrak{q})$ *for all* $m$, *we see that the natural map:*

$$\pi : \mathrm{T}(V) \to \mathrm{Cl}(V, \mathfrak{q}), \tag{59}$$

*is a surjective homomorphism of filtered algebras. Indeed, since* $\mathrm{Cl}(V, \mathfrak{q})$ *is a well-defined algebra, since we modded out a two-sided ideal,* $\pi$ *is clearly a homomorphism of filtered algebras. As a quotient map* $\pi$ *is automatically surjective.*

**Definition D.7** (The grade of an element)**.** *For* $x \in \mathrm{Cl}(V, \mathfrak{q}) \setminus \{0\}$ *we define its* grade *through the following condition:*

$$\mathrm{grd}\, x := k \qquad \text{such that} \qquad x \in \mathrm{Cl}^{(\leq k)}(V, \mathfrak{q}) \setminus \mathrm{Cl}^{(\leq k-1)}(V, \mathfrak{q}), \tag{60}$$

$$\mathrm{grd}\, 0 := -\infty. \tag{61}$$

---

[10]For the purpose of this text an algebra is always considered to be associative and unital (containing an identity element), but not necessarily commutative.

[11]An algebra homomorphism is both linear and multiplicative.

[12]A filtration $\mathcal{F}$ is an indexed family $(A_i)_{i \in I}$ ($I$ is an ordered index set) of subsets of an algebraic structure $A$ such that for $i \leq j : A_i \subseteq A_j$.

### D.3 The Parity Grading

In this section, we introduce the parity grading of the Clifford algebra. We will use the parity grading to later construct the (adjusted) twisted conjugation map, which will be used as a group action on the Clifford algebra.

**Definition D.8** (The main involution). *The linear map:*

$$\alpha : V \to \mathrm{Cl}(V, \mathfrak{q}), \qquad\qquad \alpha(v) := -v, \qquad\qquad (62)$$

*satisfies* $(-v)^2 = v^2 = \mathfrak{q}(v)$. *The universal property of* $\mathrm{Cl}(V, \mathfrak{q})$ *thus extends* $\alpha$ *to a unique algebra homomorphism:*

$$\alpha : \mathrm{Cl}(V, \mathfrak{q}) \to \mathrm{Cl}(V, \mathfrak{q}), \qquad \alpha \left( c_0 + \sum_{i \in I} c_i \cdot v_{i,1} \cdots v_{i,k_i} \right) \qquad (63)$$

$$= c_0 + \sum_{i \in I} c_i \cdot \alpha(v_{i,1}) \cdots \alpha(v_{i,k_i}) \qquad (64)$$

$$= c_0 + \sum_{i \in I} (-1)^{k_i} \cdot c_i \cdot v_{i,1} \cdots v_{i,k_i}, \qquad (65)$$

*for any finite sum representation with* $v_{i,j} \in V$ *and* $c_i \in \mathbb{F}$. *This extension* $\alpha$ *will be called the* main involution[13] *of* $\mathrm{Cl}(V, \mathfrak{q})$.

**Definition D.9** (Parity grading). *The main involution* $\alpha$ *of* $\mathrm{Cl}(V, \mathfrak{q})$ *now defines the* parity grading *of* $\mathrm{Cl}(V, \mathfrak{q})$ *via the following* homogeneous parts*:*

$$\mathrm{Cl}^{[0]}(V, \mathfrak{q}) := \{ x \in \mathrm{Cl}(V, \mathfrak{q}) \,|\, \alpha(x) = x \}, \qquad (66)$$

$$\mathrm{Cl}^{[1]}(V, \mathfrak{q}) := \{ x \in \mathrm{Cl}(V, \mathfrak{q}) \,|\, \alpha(x) = -x \}. \qquad (67)$$

*With this we get the direct sum decomposition:*

$$\mathrm{Cl}(V, \mathfrak{q}) = \mathrm{Cl}^{[0]}(V, \mathfrak{q}) \oplus \mathrm{Cl}^{[1]}(V, \mathfrak{q}), \qquad (68)$$

$$x = x^{[0]} + x^{[1]}, \qquad x^{[0]} := \frac{1}{2}(x + \alpha(x)), \quad x^{[1]} := \frac{1}{2}(x - \alpha(x)), \quad (69)$$

*with the homogeneous parts* $x^{[0]} \in \mathrm{Cl}^{[0]}(V, \mathfrak{q})$ *and* $x^{[1]} \in \mathrm{Cl}^{[1]}(V, \mathfrak{q})$.

*We define the* parity *of an (homogeneous) element* $x \in \mathrm{Cl}(V, \mathfrak{q})$ *as follows:*

$$\mathrm{prt}(x) := \begin{cases} 0 & \text{if } x \in \mathrm{Cl}^{[0]}(V, \mathfrak{q}), \\ 1 & \text{if } x \in \mathrm{Cl}^{[1]}(V, \mathfrak{q}). \end{cases} \qquad (70)$$

**Definition D.10** ($\mathbb{Z}/2\mathbb{Z}$-graded algebras). *An* $\mathbb{F}$*-algebra* $(\mathcal{A}, +, \cdot)$ *together with a direct sum decomposition of sub-vector spaces:*

$$\mathcal{A} = \mathcal{A}^{[0]} \oplus \mathcal{A}^{[1]}, \qquad (71)$$

*is called a* $\mathbb{Z}/2\mathbb{Z}$-graded algebra *if for every* $i, j \in \mathbb{Z}/2\mathbb{Z}$ *we always have:*

$$x \in \mathcal{A}^{[i]} \quad \wedge \quad y \in \mathcal{A}^{[j]} \qquad \Longrightarrow \qquad x \cdot y \in \mathcal{A}^{[i+j]}. \qquad (72)$$

*Note that* $[i + j]$ *is meant here to be computed modulo* 2. *The* $\mathbb{Z}/2\mathbb{Z}$-grade *of an (homogeneous) element* $x \in \mathcal{A}$ *will also be called the* parity *of* $x$:

$$\mathrm{prt}(x) := \begin{cases} 0 & \text{if } x \in \mathcal{A}^{[0]}, \\ 1 & \text{if } x \in \mathcal{A}^{[1]}. \end{cases} \qquad (73)$$

*The above requirement then implies for homogeneous elements* $x, y \in \mathcal{A}$ *the relation:*

$$\mathrm{prt}(x \cdot y) = \mathrm{prt}(x) + \mathrm{prt}(y) \mod 2. \qquad (74)$$

We can now summarize the results of this section as follows:

**Theorem D.11.** *The Clifford algebra* $\mathrm{Cl}(V, \mathfrak{q})$ *is a* $\mathbb{Z}/2\mathbb{Z}$-graded algebra *in its parity grading.*

---

[13] An involution is a map that is its own inverse.

## D.4 The Dimension of the Clifford Algebra

In this subsection we determine the dimension of the Clifford algebra, which allows us to construct bases for the Clifford algebra.

In the following, we again let $\mathbb{F}$ be any field of $\mathrm{char}(\mathbb{F}) \neq 2$.

**Definition/Lemma D.12** (The twisted tensor product of $\mathbb{Z}/2\mathbb{Z}$-graded $\mathbb{F}$-algebras). *Let $\mathcal{A}$ and $\mathcal{B}$ be two $\mathbb{Z}/2\mathbb{Z}$-graded algebras over $\mathbb{F}$. Then their twisted tensor product $\mathcal{A}\hat{\otimes}\mathcal{B}$ is defined via the usual tensor product $\mathcal{A} \otimes \mathcal{B}$ of $\mathbb{F}$-vector spaces, but where the product is defined on homogeneous elements $a_1, a_2 \in \mathcal{A}$, $b_1, b_2 \in \mathcal{B}$ via:*

$$(a_1\hat{\otimes}b_1) \cdot (a_2\hat{\otimes}b_2) := (-1)^{\mathrm{prt}(b_1)\cdot\mathrm{prt}(a_2)}(a_1 a_2)\hat{\otimes}(b_1 b_2). \tag{75}$$

*This turns $\mathcal{A}\hat{\otimes}\mathcal{B}$ also into a $\mathbb{Z}/2\mathbb{Z}$-graded algebra over $\mathbb{F}$ with the $\mathbb{Z}/2\mathbb{Z}$-grading:*

$$(\mathcal{A}\hat{\otimes}\mathcal{B})^{[0]} := \left(\mathcal{A}^{[0]} \otimes \mathcal{B}^{[0]}\right) \oplus \left(\mathcal{A}^{[1]} \otimes \mathcal{B}^{[1]}\right), \tag{76}$$

$$(\mathcal{A}\hat{\otimes}\mathcal{B})^{[1]} := \left(\mathcal{A}^{[0]} \otimes \mathcal{B}^{[1]}\right) \oplus \left(\mathcal{A}^{[1]} \otimes \mathcal{B}^{[0]}\right). \tag{77}$$

*In particular, if $a \in \mathcal{A}$, $b \in \mathcal{B}$ are homogeneous elements then we have:*

$$\mathrm{prt}_{\mathcal{A}\hat{\otimes}\mathcal{B}}(a\hat{\otimes}b) = \mathrm{prt}_{\mathcal{A}}(a) + \mathrm{prt}_{\mathcal{B}}(b) \mod 2. \tag{78}$$

*Proof.* By definition we already know that $\mathcal{A}\hat{\otimes}\mathcal{B}$ is an $\mathbb{F}$-vector space. So we only need to investigate the multiplication and $\mathbb{Z}/2\mathbb{Z}$-grading.

For homogenous elements $a_1, a_2, a_3 \in \mathcal{A}$, $b_1, b_2, b_3 \in \mathcal{B}$ we have:

$$\left((a_1\hat{\otimes}b_1) \cdot (a_2\hat{\otimes}b_2)\right) \cdot (a_3\hat{\otimes}b_3) \tag{79}$$

$$= (-1)^{\mathrm{prt}(b_1)\cdot\mathrm{prt}(a_2)} \left((a_1 a_2)\hat{\otimes}(b_1 b_2)\right) \cdot (a_3\hat{\otimes}b_3) \tag{80}$$

$$= (-1)^{\mathrm{prt}(b_1)\cdot\mathrm{prt}(a_2)} \cdot (-1)^{\mathrm{prt}(b_1 b_2)\cdot\mathrm{prt}(a_3)}(a_1 a_2 a_3)\hat{\otimes}(b_1 b_2 b_3) \tag{81}$$

$$= (-1)^{\mathrm{prt}(b_1)\cdot\mathrm{prt}(a_2)+\mathrm{prt}(b_1)\cdot\mathrm{prt}(a_3)+\mathrm{prt}(b_2)\cdot\mathrm{prt}(a_3)}(a_1 a_2 a_3)\hat{\otimes}(b_1 b_2 b_3) \tag{82}$$

$$= (-1)^{\mathrm{prt}(b_1)\cdot\mathrm{prt}(a_2 a_3)} \cdot (-1)^{\mathrm{prt}(b_2)\cdot\mathrm{prt}(a_3)}(a_1 a_2 a_3)\hat{\otimes}(b_1 b_2 b_3) \tag{83}$$

$$= (-1)^{\mathrm{prt}(b_2)\cdot\mathrm{prt}(a_3)}(a_1\hat{\otimes}b_1) \cdot \left((a_2 a_3)\hat{\otimes}(b_2 b_3)\right) \tag{84}$$

$$= (a_1\hat{\otimes}b_1) \cdot \left((a_2\hat{\otimes}b_2) \cdot (a_3\hat{\otimes}b_3)\right). \tag{85}$$

This shows associativity of multiplication on homogeneous elements, which extends by linearity to general elements. The distributive law is clear.

To check that we have a $\mathbb{Z}/2\mathbb{Z}$-grading, note that for homogeneous elements $a_1, a_2 \in \mathcal{A}$, $b_1, b_2 \in \mathcal{B}$ we have:

$$\mathrm{prt}_{\mathcal{A}\hat{\otimes}\mathcal{B}}\left((a_1\hat{\otimes}b_1) \cdot (a_2\hat{\otimes}b_2)\right) = \mathrm{prt}_{\mathcal{A}\hat{\otimes}\mathcal{B}}\left(((-1)^{\mathrm{prt}(a_2)\cdot\mathrm{prt}(b_1)}a_1 a_2)\hat{\otimes}(b_1 b_2)\right) \tag{86}$$

$$= \mathrm{prt}_{\mathcal{A}}(a_1 a_2) + \mathrm{prt}_{\mathcal{B}}(b_1 b_2) \tag{87}$$

$$= \mathrm{prt}_{\mathcal{A}}(a_1) + \mathrm{prt}_{\mathcal{A}}(a_2) + \mathrm{prt}_{\mathcal{B}}(b_1) + \mathrm{prt}_{\mathcal{B}}(b_2) \tag{88}$$

$$= \mathrm{prt}_{\mathcal{A}\hat{\otimes}\mathcal{B}}\left(a_1\hat{\otimes}b_1\right) + \mathrm{prt}_{\mathcal{A}\hat{\otimes}\mathcal{B}}\left(a_2\hat{\otimes}b_2\right) \mod 2. \tag{89}$$

The general case follows by linear combinations. $\square$

**Remark D.13** (The universal property of the twisted tensor product of $\mathbb{Z}/2\mathbb{Z}$-graded $\mathbb{F}$-algebras). *Let $\mathcal{A}_1, \mathcal{A}_2, \mathcal{B}$ be $\mathbb{Z}/2\mathbb{Z}$-graded $\mathbb{F}$-algebras. Consider $\mathbb{Z}/2\mathbb{Z}$-graded $\mathbb{F}$-algebra homomorphisms:*

$$\psi_1 : \mathcal{A}_1 \to \mathcal{B}, \qquad\qquad \psi_2 : \mathcal{A}_2 \to \mathcal{B}, \tag{90}$$

*such that for all homogeneous elements $a_1 \in \mathcal{A}_1$ and $a_2 \in \mathcal{A}_2$ we have:*

$$\psi_1(a_1) \cdot \psi_2(a_2) = (-1)^{\mathrm{prt}(a_1)\cdot\mathrm{prt}(a_2)} \cdot \psi_2(a_2) \cdot \psi_1(a_1). \tag{91}$$

*Then there exists a unique $\mathbb{Z}/2\mathbb{Z}$-graded $\mathbb{F}$-algebra homomorphism:*

$$\psi : \mathcal{A}_1\hat{\otimes}\mathcal{A}_2 \to \mathcal{B}, \tag{92}$$

*such that:*

$$\psi \circ \phi_1 = \psi_1, \qquad\qquad \psi \circ \phi_2 = \psi_2, \qquad\qquad (93)$$

*where $\phi_1, \phi_2$ are the following $\mathbb{Z}/2\mathbb{Z}$-graded $\mathbb{F}$-algebra homomorphisms:*

$$\phi_1 : \mathcal{A}_1 \to \mathcal{A}_1 \hat{\otimes} \mathcal{A}_2, \qquad\qquad a_1 \mapsto a_1 \hat{\otimes} 1, \qquad\qquad (94)$$

$$\phi_2 : \mathcal{A}_2 \to \mathcal{A}_1 \hat{\otimes} \mathcal{A}_2, \qquad\qquad a_2 \mapsto 1 \hat{\otimes} a_2, \qquad\qquad (95)$$

*which satisfy for all homogeneous elements $a_1 \in \mathcal{A}_1$ and $a_2 \in \mathcal{A}_2$:*

$$\phi_1(a_1) \cdot \phi_2(a_2) = (-1)^{\mathrm{prt}(a_1) \cdot \mathrm{prt}(a_2)} \cdot \phi_2(a_2) \cdot \phi_1(a_1). \qquad\qquad (96)$$

*Furthermore, $\mathcal{A}_1 \hat{\otimes} \mathcal{A}_2$ together with $\phi_1, \phi_2$ is uniquely characterized as a $\mathbb{Z}/2\mathbb{Z}$-graded $\mathbb{F}$-algebra by the above property (when considering all possible such $\mathcal{B}$ and $\psi_1, \psi_2$).*

**Proposition D.14.** *Let $(V, \mathfrak{q})$ be a finite dimensional quadratic vector space over $\mathbb{F}$, $\mathrm{char}(\mathbb{F}) \neq 2$, with an orthogonal sum decomposition:*

$$(V, \mathfrak{q}) = (V_1, \mathfrak{q}_1) \oplus (V_2, \mathfrak{q}_2). \qquad\qquad (97)$$

*Then the inclusion maps: $\mathrm{Cl}(V_i, \mathfrak{q}_i) \to \mathrm{Cl}(V, \mathfrak{q})$ induce an isomorphism of $\mathbb{Z}/2\mathbb{Z}$-graded $\mathbb{F}$-algebras:*

$$\mathrm{Cl}(V, \mathfrak{q}) \cong \mathrm{Cl}(V_1, \mathfrak{q}_1) \hat{\otimes} \mathrm{Cl}(V_2, \mathfrak{q}_2). \qquad\qquad (98)$$

*In particular:*

$$\dim \mathrm{Cl}(V, \mathfrak{q}) = \dim \mathrm{Cl}(V_1, \mathfrak{q}_1) \cdot \dim \mathrm{Cl}(V_2, \mathfrak{q}_2). \qquad\qquad (99)$$

*Proof.* First consider the canonical inclusion maps, $l = 1, 2$:

$$\phi_l : \mathrm{Cl}(V_l, \mathfrak{q}_l) \to \mathrm{Cl}(V, \mathfrak{q}), \qquad\qquad x_l \mapsto x_l. \qquad\qquad (100)$$

These satisfy for all (parity) homogeneous elements $x_l \in \mathrm{Cl}(V_l, \mathfrak{q}_l)$, $l = 1, 2$, the condition:

$$\phi_1(x_1)\phi_2(x_2) = x_1 x_2 \overset{!}{=} (-1)^{\mathrm{prt}(x_1) \cdot \mathrm{prt}(x_2)} \cdot x_2 x_1 = (-1)^{\mathrm{prt}(x_1) \cdot \mathrm{prt}(x_2)} \cdot \phi_2(x_2)\phi_1(x_1). \quad (101)$$

Indeed, we can by linearity reduce to the case that $x_1$ and $x_2$ are products of vectors $v_1 \in V_1$ and $v_2 \in V_2$, resp. Since $V_1$ and $V_2$ are orthogonal to each other by assumption, for those elements we get by the identities D.3:

$$v_1 v_2 = -v_2 v_1 + 2 \underbrace{\mathfrak{b}(v_1, v_2)}_{=0} = -v_2 v_1. \qquad\qquad (102)$$

This shows the condition above for $x_1$ and $x_2$.

We then define the $\mathbb{F}$-bilinear map:

$$\mathrm{Cl}(V_1, \mathfrak{q}_1) \times \mathrm{Cl}(V_2, \mathfrak{q}_2) \to \mathrm{Cl}(V, \mathfrak{q}), \qquad\qquad (x_1, x_2) \mapsto x_1 x_2, \qquad\qquad (103)$$

which thus factorizes through the $\mathbb{F}$-linear map:

$$\phi : \mathrm{Cl}(V_1, \mathfrak{q}_1) \hat{\otimes} \mathrm{Cl}(V_2, \mathfrak{q}_2) \to \mathrm{Cl}(V, \mathfrak{q}), \qquad\qquad x_1 \hat{\otimes} x_2 \mapsto x_1 x_2. \qquad\qquad (104)$$

We now show that $\phi$ also respects multiplication. For this let $x_1, y_1 \in \mathrm{Cl}(V_1, \mathfrak{q}_1)$ and $x_2, y_2 \in \mathrm{Cl}(V_2, \mathfrak{q}_2)$ (parity) homogeneous elements. We then get:

$$\phi\left((x_1 \hat{\otimes} x_2) \cdot (y_1 \hat{\otimes} y_2)\right) \qquad\qquad (105)$$

$$= \phi\left((-1)^{\mathrm{prt}(x_2) \cdot \mathrm{prt}(y_1)} \cdot (x_1 y_1) \hat{\otimes} (x_2 y_2)\right) \qquad\qquad (106)$$

$$= (-1)^{\mathrm{prt}(x_2) \cdot \mathrm{prt}(y_1)} \cdot x_1 y_1 x_2 y_2 \qquad\qquad (107)$$

$$= x_1 x_2 y_1 y_2 \qquad\qquad (108)$$

$$= \phi(x_1 \hat{\otimes} x_2) \cdot \phi(y_1 \hat{\otimes} y_2). \qquad\qquad (109)$$

This shows the multiplicativity of $\phi$ on homogeneous elements. The general case follows by the $\mathbb{F}$-linearity of $\phi$. Note that $\phi$ also respects the parity grading.

Now consider the $\mathbb{Z}/2\mathbb{Z}$-graded $\mathbb{F}$-algebra $\mathrm{Cl}(V_1, \mathfrak{q}_1) \hat{\otimes} \mathrm{Cl}(V_2, \mathfrak{q}_2)$ and $\mathbb{Z}/2\mathbb{Z}$-graded $\mathbb{F}$-algebra homomorphisms:

$$\psi_1 : \mathrm{Cl}(V_1, \mathfrak{q}_1) \to \mathrm{Cl}(V_1, \mathfrak{q}_1) \hat{\otimes} \mathrm{Cl}(V_2, \mathfrak{q}_2), \qquad x_1 \mapsto x_1 \hat{\otimes} 1, \tag{110}$$

$$\psi_2 : \mathrm{Cl}(V_2, \mathfrak{q}_2) \to \mathrm{Cl}(V_1, \mathfrak{q}_1) \hat{\otimes} \mathrm{Cl}(V_2, \mathfrak{q}_2), \qquad x_2 \mapsto 1 \hat{\otimes} x_2. \tag{111}$$

Note that for all homogeneous elements $x_l \in \mathrm{Cl}(V_l, \mathfrak{q}_l)$, $l = 1, 2$, we have:

$$\psi_2(x_2) \cdot \psi_1(x_1) = (1 \hat{\otimes} x_2) \cdot (x_1 \hat{\otimes} 1) \tag{112}$$

$$= (-1)^{\mathrm{prt}(x_1) \cdot \mathrm{prt}(x_2)} \cdot x_1 \hat{\otimes} x_2 \tag{113}$$

$$= (-1)^{\mathrm{prt}(x_1) \cdot \mathrm{prt}(x_2)} \cdot (x_1 \hat{\otimes} 1) \cdot (1 \hat{\otimes} x_2) \tag{114}$$

$$= (-1)^{\mathrm{prt}(x_1) \cdot \mathrm{prt}(x_2)} \cdot \psi_1(x_1) \cdot \psi_2(x_2). \tag{115}$$

We then define the $\mathbb{F}$-linear map:

$$\psi : V = V_1 \oplus V_2 \to \mathrm{Cl}(V_1, \mathfrak{q}_1) \hat{\otimes} \mathrm{Cl}(V_2, \mathfrak{q}_2), \quad v = v_1 + v_2 \mapsto \psi_1(v_1) + \psi_2(v_2) =: \psi(v). \tag{116}$$

Note that we have:

$$\psi(v)^2 = (\psi_1(v_1) + \psi_2(v_2)) \cdot (\psi_1(v_1) + \psi_2(v_2)) \tag{117}$$

$$= \psi_1(v_1)^2 + \psi_2(v_2)^2 + \psi_1(v_1) \cdot \psi_2(v_2) + \psi_2(v_2) \cdot \psi_1(v_1) \tag{118}$$

$$= \psi_1(v_1)^2 + \psi_2(v_2)^2 + \underbrace{\psi_1(v_1) \cdot \psi_2(v_2) + (-1)^{\mathrm{prt}(v_1) \cdot \mathrm{prt}(v_2)} \psi_1(v_1) \cdot \psi_2(v_2)}_{=0} \tag{119}$$

$$= \psi_1(v_1^2) + \psi_2(v_2^2) \tag{120}$$

$$= \mathfrak{q}_1(v_1) \cdot \psi_1(1_{\mathrm{Cl}(V_1, \mathfrak{q}_1)}) + \mathfrak{q}_2(v_2) \cdot \psi_2(1_{\mathrm{Cl}(V_2, \mathfrak{q}_2)}) \tag{121}$$

$$= \mathfrak{q}_1(v_1) \cdot 1_{\mathrm{Cl}(V_1, \mathfrak{q}_1) \hat{\otimes} \mathrm{Cl}(V_2, \mathfrak{q}_2)} + \mathfrak{q}_2(v_2) \cdot 1_{\mathrm{Cl}(V_1, \mathfrak{q}_1) \hat{\otimes} \mathrm{Cl}(V_2, \mathfrak{q}_2)} \tag{122}$$

$$= (\mathfrak{q}_1(v_1) + \mathfrak{q}_2(v_2)) \cdot 1_{\mathrm{Cl}(V_1, \mathfrak{q}_1) \hat{\otimes} \mathrm{Cl}(V_2, \mathfrak{q}_2)} \tag{123}$$

$$= \mathfrak{q}(v) \cdot 1_{\mathrm{Cl}(V_1, \mathfrak{q}_1) \hat{\otimes} \mathrm{Cl}(V_2, \mathfrak{q}_2)}. \tag{124}$$

By the universal property of the Clifford algebra $\psi$ uniquely extends to an $\mathbb{F}$-algebra homomorphism:

$$\psi : \mathrm{Cl}(V, \mathfrak{q}) \to \mathrm{Cl}(V_1, \mathfrak{q}_1) \hat{\otimes} \mathrm{Cl}(V_2, \mathfrak{q}_2), \tag{125}$$

$$v_{i_1} \cdots v_{i_k} \mapsto (\psi_1(v_{i_1,1}) + \psi_2(v_{i_1,2})) \cdots (\psi_1(v_{i_k,1}) + \psi_2(v_{i_k,2})). \tag{126}$$

One can see from this, by explicit calculation, that $\psi$ also respects the $\mathbb{Z}/2\mathbb{Z}$-grading. Furthermore, we see that $\psi \circ \phi_l = \psi_l$ for $l = 1, 2$, and, also, $\phi \circ \psi_l = \phi_l$ for $l = 1, 2$.

One easily sees that $\phi$ and $\psi$ are inverse to each other and the claim follows. $\qquad \square$

**Theorem D.15** (The dimension of the Clifford algebra). *Let $(V, \mathfrak{q})$ be a finite dimensional quadratic vector space over $\mathbb{F}$, $\mathrm{char}(\mathbb{F}) \neq 2$, $n := \dim V < \infty$. Then we have:*

$$\dim \mathrm{Cl}(V, \mathfrak{q}) = 2^n. \tag{127}$$

*Proof.* Let $e_1, \ldots, e_n$ be an orthogonal basis of $(V, \mathfrak{q})$ and $V_i := \mathrm{span}(e_i) \subseteq V$, $\mathfrak{q}_i := \mathfrak{q}|_{V_i}$. Then we get the orthogonal sum decomposition:

$$(V, \mathfrak{q}) = (V_1, \mathfrak{q}_1) \oplus \cdots \oplus (V_n, \mathfrak{q}_n), \tag{128}$$

and thus by Proposition D.14 the isomorphism of $\mathbb{Z}/2\mathbb{Z}$-graded algebras:

$$\mathrm{Cl}(V, \mathfrak{q}) \cong \mathrm{Cl}(V_1, \mathfrak{q}_1) \hat{\otimes} \cdots \hat{\otimes} \mathrm{Cl}(V_n, \mathfrak{q}_n), \tag{129}$$

and thus:

$$\dim \mathrm{Cl}(V, \mathfrak{q}) = \dim \mathrm{Cl}(V_1, \mathfrak{q}_1) \cdots \dim \mathrm{Cl}(V_n, \mathfrak{q}_n), \tag{130}$$

Since $\dim V_i = 1$ and thus:

$$\dim \mathrm{Cl}(V_i, \mathfrak{q}_i) = \dim (\mathbb{F} \oplus V_i) = 2, \tag{131}$$

for $i = 1, \ldots, n$, we get the claim:

$$\dim \mathrm{Cl}(V, \mathfrak{q}) = 2^n. \tag{132}$$

$\qquad \square$

**Corollary D.16** (Bases for the Clifford algebra)**.** *Let* $(V, \mathfrak{q})$ *be a finite dimensional quadratic vector space over* $\mathbb{F}$, $\mathrm{char}(\mathbb{F}) \neq 2$, $n := \dim V < \infty$. *Let* $e_1, \ldots, e_n$ *be any basis of* $V$, *then* $(e_A)_{A \subseteq [n]}$ *is a basis for* $\mathrm{Cl}(V, \mathfrak{q})$, *where we put for a subset* $A \subseteq [n] := \{1, \ldots, n\}$:

$$e_A := \prod_{i \in A}^{<} e_i, \qquad\qquad e_{\emptyset} := 1_{\mathrm{Cl}(V,\mathfrak{q})}. \tag{133}$$

*where the product is taken in increasing order of the indices* $i \in A$.

*Proof.* Since $\mathrm{Cl}(V, \mathfrak{q}) = \mathrm{T}(V)/I(\mathfrak{q})$ and

$$\mathrm{T}(V) = \mathrm{span}\left\{ e_{i_1} \otimes \cdots \otimes e_{i_m} \mid m \geq 0, i_j \in [n], j \in [m] \right\}, \tag{134}$$

we see that:

$$\mathrm{Cl}(V, \mathfrak{q}) = \mathrm{span}\left\{ e_{i_1} \cdots e_{i_m} \mid m \geq 0, i_j \in [n], j \in [m] \right\}. \tag{135}$$

By several applications of the fundamental identities D.3:

$$e_{i_k} e_{i_l} = -e_{i_l} e_{i_k} + 2\mathfrak{b}(e_{i_k}, e_{i_l}), \qquad\qquad e_{i_k} e_{i_k} = \mathfrak{q}(e_{i_k}), \tag{136}$$

we can turn products $e_{i_1} \cdots e_{i_m}$ into sums of smaller products if some of the occuring indices agree, say $i_k = i_l$. Furthermore, we can also use those identities to turn the indices in increasing order. This then shows:

$$\mathrm{Cl}(V, \mathfrak{q}) = \mathrm{span}\left\{ e_A \mid A \subseteq [n] \right\}. \tag{137}$$

Since $\# \{A \subseteq [n]\} = 2^n$ and $\dim \mathrm{Cl}(V, \mathfrak{q}) = 2^n$ by Theorem D.15 we see that $\{e_A \mid A \subseteq [n]\}$ must already be a basis for $\mathrm{Cl}(V, \mathfrak{q})$. $\square$

## D.5 Extending the Quadratic Form to the Clifford Algebra

We provide the extension of the quadratic form from the vector space to the Clifford algebra, which will lead to the constrution of an *orthogonal* basis of the Clifford algebra.

**Definition D.17** (The opposite algebra of an algebra)**.** *Let* $(\mathcal{A}, +, \cdot)$ *be an algebra. The* opposite algebra $(\mathcal{A}^{\mathrm{op}}, +, \bullet)$ *is defined to consist of the same underlying vector space* $(\mathcal{A}^{\mathrm{op}}, +) = (\mathcal{A}, +)$, *but where the multiplication is reversed in comparison to* $\mathcal{A}$, *i.e. for* $x, y \in \mathcal{A}^{\mathrm{op}}$ *we have:*

$$x \bullet y := y \cdot x. \tag{138}$$

*Note that this really turns* $(\mathcal{A}^{\mathrm{op}}, +, \bullet)$ *into an algebra.*

**Definition D.18** (The main anti-involution of the Clifford algebra)**.** *Consider the following linear map:*

$$\beta : V \to \mathrm{Cl}(V, \mathfrak{q})^{\mathrm{op}}, \qquad\qquad v \mapsto v, \tag{139}$$

*which also satisfies* $v \bullet v = vv = \mathfrak{q}(v)$. *By the universal property of the Clifford algebra we get a unique extension to an algebra homomorphism:*

$$\beta : \mathrm{Cl}(V, \mathfrak{q}) \to \mathrm{Cl}(V, \mathfrak{q})^{\mathrm{op}}, \qquad\qquad \beta\left( c_0 + \sum_{i \in I} c_i \cdot v_{i,1} \cdots v_{i,k_i} \right) \tag{140}$$

$$= c_0 + \sum_{i \in I} c_i \cdot v_{i,1} \bullet \cdots \bullet v_{i,k_i} \tag{141}$$

$$= c_0 + \sum_{i \in I} c_i \cdot v_{i,k_i} \cdots v_{i,1}, \tag{142}$$

*for any finite sum representation with* $v_{i,j} \in V$ *and* $c_i \in \mathbb{F}$. *We call* $\beta$ *the* main anti-involution *of* $\mathrm{Cl}(V, \mathfrak{q})$.

**Definition D.19** (The combined anti-involution of the Clifford algebra)**.** *The* combined anti-involution *or* Clifford conjugation *of* $\mathrm{Cl}(V, \mathfrak{q})$ *is defined to be the* $\mathbb{F}$-*algebra homomorphism:*

$$\gamma : \mathrm{Cl}(V, \mathfrak{q}) \to \mathrm{Cl}(V, \mathfrak{q})^{\mathrm{op}}, \qquad\qquad \gamma(x) := \beta(\alpha(x)). \tag{143}$$

*More explicitely, it is given by the formula:*

$$\gamma\left( c_0 + \sum_{i \in I} c_i \cdot v_{i,1} \cdots v_{i,k_i} \right) = c_0 + \sum_{i \in I} (-1)^{k_i} \cdot c_i \cdot v_{i,k_i} \cdots v_{i,1}, \tag{144}$$

*for any finite sum representation with* $v_{i,j} \in V$ *and* $c_i \in \mathbb{F}$.

**Remark D.20.** *If $e_1, \ldots, e_n \in V$ is an orthogonal basis for $(V, \mathfrak{q})$. Then we have for $A \subseteq [n]$:*

$$\alpha(e_A) = (-1)^{|A|} e_A, \qquad \beta(e_A) = (-1)^{\binom{|A|}{2}} e_A, \qquad \gamma(e_A) = (-1)^{\binom{|A|+1}{2}} e_A. \qquad (145)$$

Recall the definition of a trace of a linear map:

**Definition D.21** (The trace of an endomorphism). *Let $\mathcal{Y}$ be a vector space over a field $\mathbb{F}$ of dimension $\dim \mathcal{Y} = m < \infty$ and $\Phi : \mathcal{Y} \to \mathcal{Y}$ a vector space endomorphism[14]. Let $B = \{b_1, \ldots, b_m\}$ be a basis for $\mathcal{Y}$ and $B^* = \{b_1^*, \ldots, b_m^*\}$ be the corresponding dual basis of $\mathcal{Y}^*$, defined via: $b_j^*(b_i) := \delta_{i,j}$. Let $A = (a_{i,j})_{\substack{i=1,\ldots,m \\ j=1,\ldots,m}}$, be the matrix representation of $\Phi$ w.r.t. $B$:*

$$\forall j \in [m]. \qquad \Phi(b_j) = \sum_{i=1}^{m} a_{i,j} b_i. \qquad (146)$$

*Then the* trace *of $\Phi$ is defined via:*

$$\mathrm{Tr}(\Phi) := \sum_{j=1}^{m} a_{j,j} = \sum_{j=1}^{m} b_j^*(\Phi(b_j)) \in \mathbb{F}. \qquad (147)$$

*It is a well known fact that $\mathrm{Tr}(\Phi)$ is not dependent on the initial choice of the basis $B$. Furthermore, $\mathrm{Tr}$ is a well-defined $\mathbb{F}$-linear map (homomorphism of vector spaces):*

$$\mathrm{Tr} : \; \mathrm{End}_{\mathbb{F}}(\mathcal{Y}) \to \mathbb{F}, \qquad\qquad \Phi \mapsto \mathrm{Tr}(\Phi). \qquad (148)$$

We now want to define the projection of $x \in \mathrm{Cl}(V, \mathfrak{q})$ onto its zero component $x^{(0)} \in \mathbb{F}$ in a basis independent way.

**Definition D.22** (The projection onto the zero component). *We define the $\mathbb{F}$-linear map:*

$$\zeta : \; \mathrm{Cl}(V, \mathfrak{q}) \to \mathbb{F}, \qquad\qquad \zeta(x) := 2^{-n} \mathrm{Tr}(x), \qquad (149)$$

*where $n := \dim V$ and $\mathrm{Tr}(x) := \mathrm{Tr}(L_x)$, where $L_x$ is the endomorphism of $\mathrm{Cl}(V, \mathfrak{q})$ given by left multiplication with $x$:*

$$L_x : \; \mathrm{Cl}(V, \mathfrak{q}) \to \mathrm{Cl}(V, \mathfrak{q}), \qquad\qquad y \mapsto L_x(y) := xy. \qquad (150)$$

*We call $\zeta$ the* projection onto the zero component. *We also often write for $x \in \mathrm{Cl}(V, \mathfrak{q})$:*

$$x^{(0)} := \zeta(x). \qquad (151)$$

The name is justified by following property:

**Lemma D.23.** *Let $e_1, \ldots, e_n$ be a fixed orthogonal basis of $(V, \mathfrak{q})$. Then we know that $(e_A)_{A \subseteq [n]}$ is a basis for $\mathrm{Cl}(V, \mathfrak{q})$. So we can write every $x \in \mathrm{Cl}(V, \mathfrak{q})$ as:*

$$x = \sum_{A \subseteq [n]} x_A \cdot e_A, \qquad (152)$$

*with $x_A \in \mathbb{F}$, $A \subseteq [n]$. The claim is now that we have:*

$$\zeta(x) \overset{!}{=} x_\emptyset. \qquad (153)$$

*Proof.* By the linearity of the trace we only need to investigate $\mathrm{Tr}(e_A)$ for $A \subseteq [n]$. For $A = \emptyset$, we have $e_\emptyset = 1$ and we get:

$$\mathrm{Tr}(1) = \sum_{B \subseteq [n]} e_B^*(1 \cdot e_B) = \sum_{B \subseteq [n]} 1 = 2^n, \qquad (154)$$

---

[14]A map from a mathematical object space to itself.

which shows: $\zeta(1) = 2^{-n} \operatorname{Tr}(1) = 1$. Now, consider $A \subseteq [n]$ with $A \neq \emptyset$. Let $\triangle$ denote the symmetric difference of two sets. Further, we can write $\pm$ to refrain from distinguishing between signs, which will not affect the result.

$$\operatorname{Tr}(e_A) = \sum_{B \subseteq [n]} e_B^*(e_A e_B) \tag{155}$$

$$= \sum_{B \subseteq [n]} \pm \prod_{i \in A \cap B} \mathfrak{q}(e_i) \cdot e_B^*(e_{A \triangle B}) \tag{156}$$

$$= \sum_{B \subseteq [n]} \pm \prod_{i \in A \cap B} \mathfrak{q}(e_i) \cdot \delta_{B, A \triangle B} \tag{157}$$

$$= \sum_{B \subseteq [n]} \pm \prod_{i \in A \cap B} \mathfrak{q}(e_i) \cdot \delta_{\emptyset, A} \tag{158}$$

$$= 0, \tag{159}$$

where the third equality follows from the fact that $B = A \triangle B$ holds if and only if $A = \emptyset$, regardless of $B$. However, $A = \emptyset$ was ruled out by assumption, so then in the last equality we always have $\delta_{\emptyset, A} = 0$. So for $A \neq \emptyset$ we have: $\zeta(e_A) = 2^{-n} \operatorname{Tr}(e_A) = 0$. Altogether we get:

$$\zeta(e_A) = \delta_{A, \emptyset} = \begin{cases} 1, & \text{if } A = \emptyset, \\ 0, & \text{else.} \end{cases} \tag{160}$$

With this and linearity we get:

$$\zeta(x) = \zeta\left(\sum_{A \subseteq [n]} x_A \cdot e_A\right) = \sum_{A \subseteq [n]} x_A \cdot \zeta(e_A) = \sum_{A \subseteq [n]} x_A \cdot \delta_{A, \emptyset} = x_\emptyset. \tag{161}$$

This shows the claim. $\qquad\square$

**Definition D.24** (The bilinear form on the Clifford algebra)**.** *For our quadratic $\mathbb{F}$-vector space $(V, \mathfrak{q})$ with corresponding bilinear form $\mathfrak{b}$ and corresponding Clifford algebra $\operatorname{Cl}(V, \mathfrak{q})$ we now define the following $\mathbb{F}$-bilinear form on $\operatorname{Cl}(V, \mathfrak{q})$:*

$$\bar{\mathfrak{b}} : \operatorname{Cl}(V, \mathfrak{q}) \times \operatorname{Cl}(V, \mathfrak{q}) \to \mathbb{F}, \qquad\qquad \bar{\mathfrak{b}}(x, y) := \zeta(\beta(x) y). \tag{162}$$

*We also define the corresponding quadratic form on $\operatorname{Cl}(V, \mathfrak{q})$ via:*

$$\bar{\mathfrak{q}} : \operatorname{Cl}(V, \mathfrak{q}) \to \mathbb{F}, \qquad\qquad \bar{\mathfrak{q}}(x) := \bar{\mathfrak{b}}(x, x) = \zeta(\beta(x) x). \tag{163}$$

*We will see below that $\bar{\mathfrak{b}}$ and $\bar{\mathfrak{q}}$ will agree with $\mathfrak{b}$ and $\mathfrak{q}$, resp., when they are restricted to $V$. From that point on we will denote $\bar{\mathfrak{b}}$ just by $\mathfrak{b}$, and $\bar{\mathfrak{q}}$ with $\mathfrak{q}$, resp., without (much) ambiguity.*

**Lemma D.25.** *For $v, w \in V$ we have:*

$$\bar{\mathfrak{b}}(v, w) = \mathfrak{b}(v, w). \tag{164}$$

*Proof.* We pick an orthogonal basis $e_1, \ldots, e_n$ for $V$ and write:

$$v = \sum_{i=1}^{n} a_i \cdot e_i, \qquad\qquad w = \sum_{j=1}^{n} c_j \cdot e_j. \tag{165}$$

We then get by linearity:

$$\bar{\mathfrak{b}}(v,w) = \sum_{i=1}^{n}\sum_{j=1}^{n} a_i c_j \cdot \bar{\mathfrak{b}}(e_i, e_j) \tag{166}$$

$$= \sum_{i=1}^{n}\sum_{j=1}^{n} a_i c_j \cdot \zeta(\beta(e_i)e_j) \tag{167}$$

$$= \sum_{i=1}^{n}\sum_{j=1}^{n} a_i c_j \cdot \zeta(e_i e_j) \tag{168}$$

$$= \sum_{i \neq j} a_i c_j \cdot \underbrace{\zeta(e_i e_j)}_{=0} + \sum_{i=j} a_i c_j \cdot \zeta(e_i e_j) \tag{169}$$

$$= \sum_{i=1}^{n} a_i c_i \cdot \mathfrak{q}(e_i) \cdot \underbrace{\zeta(1)}_{=1} \tag{170}$$

$$= \sum_{i=1}^{n}\sum_{j=1}^{n} a_i c_j \cdot \overbrace{\bar{\mathfrak{b}}(e_i, e_j)}^{\mathfrak{q}(e_i)\cdot\delta_{i,j}=} \tag{171}$$

$$= \mathfrak{b}\left(\sum_{i=1}^{n} a_i \cdot e_i, \sum_{j=1}^{n} c_j \cdot e_j\right) \tag{172}$$

$$= \mathfrak{b}(v, w). \tag{173}$$

This shows the claim. $\qquad\square$

**Theorem D.26.** *Let $e_1, \ldots, e_n$ be an orthogonal basis for $(V, \mathfrak{q})$ then $(e_A)_{A \subseteq [n]}$ is an orthogonal basis for $\mathrm{Cl}(V, \mathfrak{q})$ w.r.t. the induced bilinear form $\bar{\mathfrak{b}}$. Furthermore, for $x, y \in \mathrm{Cl}(V, \mathfrak{q})$ of the form:*

$$x = \sum_{A \subseteq [n]} x_A \cdot e_A, \qquad\qquad y = \sum_{A \subseteq [n]} y_A \cdot e_A, \tag{174}$$

*with $x_A, y_A \in \mathbb{F}$ we get:*

$$\bar{\mathfrak{b}}(x, y) = \sum_{A \subseteq [n]} x_A \cdot y_A \cdot \prod_{i \in A} \mathfrak{q}(e_i), \qquad\qquad \bar{\mathfrak{q}}(x) = \sum_{A \subseteq [n]} x_A^2 \cdot \prod_{i \in A} \mathfrak{q}(e_i). \tag{175}$$

*Note that: $\bar{\mathfrak{q}}(e_\emptyset) = \bar{\mathfrak{q}}(1) = 1$.*

*Proof.* We already know that $(e_A)_{A \subseteq [n]}$ is a basis for $\mathrm{Cl}(V, \mathfrak{q})$. So we only need to check the orthogonality condition. First note that for $e_C = e_{i_1} \cdots e_{i_r}$ we get:

$$\bar{\mathfrak{q}}(e_C) = \zeta(\beta(e_C)e_C) = \zeta(e_{i_r} \cdots e_{i_1} \cdot e_{i_1} \cdots e_{i_r}) = \mathfrak{q}(e_{i_1}) \cdots \mathfrak{q}(e_{i_r}). \tag{176}$$

Now let $A, B \subseteq [n]$ with $A \neq B$, i.e. with $A \triangle B \neq \emptyset$. We then get:

$$\bar{\mathfrak{b}}(e_A, e_B) = \zeta(\beta(e_A)e_B) = \pm \prod_{i \in A \cap B} \mathfrak{q}(e_i) \cdot \zeta(e_{A \triangle B}) = 0. \tag{177}$$

This shows that $(e_A)_{A \subseteq [n]}$ is an orthogonal basis for $\mathrm{Cl}(V, \mathfrak{q})$. For $x, y \in \mathrm{Cl}(V, \mathfrak{q})$ from above we get:

$$\bar{\mathfrak{b}}(x, y) = \bar{\mathfrak{b}}\left( \sum_{A \subseteq [n]} x_A \cdot e_A, \sum_{B \subseteq [n]} y_B \cdot e_B \right) \tag{178}$$

$$= \sum_{A \subseteq [n]} \sum_{B \subseteq [n]} x_A \cdot y_B \cdot \bar{\mathfrak{b}}(e_A, e_B) \tag{179}$$

$$= \sum_{A \subseteq [n]} \sum_{B \subseteq [n]} x_A \cdot y_B \cdot \bar{\mathfrak{q}}(e_A) \cdot \delta_{A,B} \tag{180}$$

$$= \sum_{A \subseteq [n]} x_A \cdot y_A \cdot \bar{\mathfrak{q}}(e_A) \tag{181}$$

$$= \sum_{A \subseteq [n]} x_A \cdot y_A \cdot \prod_{i \in A} \mathfrak{q}(e_i). \tag{182}$$

This shows the claim. $\qquad\square$

## D.6 The Multivector Grading

Now that we have an orthogonal basis for the algebra, we show that the Clifford algebra allows a vector space grading that is independent of the chosen orthogonal basis.

Let $(V, \mathfrak{q})$ be a quadratic space over a field $\mathbb{F}$ with $\mathrm{char}(\mathbb{F}) \neq 2$ and $\dim V = n < \infty$.

In the following we present a technical proof that works for fields $\mathbb{F}$ with $\mathrm{char}(\mathbb{F}) \neq 2$. An alternative, simpler and more structured proof, but for the more restrictive case of $\mathrm{char}(\mathbb{F}) = 0$, can be found in Theorem D.37 later.

**Theorem D.27** (The multivector grading of the Clifford algebra). *Let $e_1, \ldots, e_n$ be an orthogonal basis of $(V, \mathfrak{q})$. Then for every $m = 0, \ldots, n$ we define the following sub-vector space of $\mathrm{Cl}(V, \mathfrak{q})$:*

$$\mathrm{Cl}^{(m)}(V, \mathfrak{q}) := \mathrm{span}\left\{ e_{i_1} \cdots e_{i_m} \,|\, 1 \leq i_1 < \cdots < i_m \leq n \right\} \tag{183}$$

$$= \mathrm{span}\left\{ e_A \,|\, A \subseteq [n], |A| = m \right\}, \tag{184}$$

*where $\mathrm{Cl}^{(0)}(V, \mathfrak{q}) := \mathbb{F}$.*

*Then the sub-vector spaces $\mathrm{Cl}^{(m)}(V, \mathfrak{q})$, $m = 0, \ldots, n$, are independent of the choice of the orthogonal basis, i.e. if $b_1, \ldots, b_n$ is another orthogonal basis of $(V, \mathfrak{q})$, then:*

$$\mathrm{Cl}^{(m)}(V, \mathfrak{q}) = \mathrm{span}\left\{ b_{i_1} \cdots b_{i_m} \,|\, 1 \leq i_1 < \cdots < i_m \leq n \right\}. \tag{185}$$

*Proof.* First note that by the orthogonality and the fundamental relation of the Clifford algebra we have for all $i \neq j$:

$$e_i e_j = -e_j e_i, \qquad\qquad\qquad b_i b_j = -b_j b_i. \tag{186}$$

We now abbreviate:

$$B^{(m)} := \mathrm{span}\left\{ b_{j_1} \cdots b_{j_m} \,|\, 1 \leq j_1 < \cdots < j_m \leq n \right\}, \tag{187}$$

and note that:

$$\mathrm{Cl}^{(m)}(V, \mathfrak{q}) = \mathrm{span}\left\{ e_{i_1} \cdots e_{i_m} \,|\, 1 \leq i_1 < \cdots < i_m \leq n \right\} \tag{188}$$

$$= \mathrm{span}\left\{ e_{i_1} \cdots e_{i_m} \,|\, 1 \leq i_1, \ldots, i_m \leq n, \forall s \neq t. i_s \neq i_t \right\}. \tag{189}$$

We want to show that for $1 \leq j_1 < \cdots < j_m \leq n$ we have that:

$$b_{j_1} \cdots b_{j_m} \in \mathrm{Cl}^{(m)}(V, \mathfrak{q}). \tag{190}$$

Since we have two bases we can write an orthogonal change of basis:

$$b_j = \sum_{i=1}^{n} a_{i,j} e_i \in V = \mathrm{Cl}^{(1)}(V, \mathfrak{q}). \tag{191}$$

Using this, we can now write the above product as the sum of two terms:

$$b_{j_1} \cdots b_{j_m} = \sum_{i_1,\ldots,i_m} a_{i_1,j_1} \cdots a_{i_m,j_m} \cdot e_{i_1} \cdots e_{i_m} \tag{192}$$

$$= \sum_{\substack{i_1,\ldots,i_m \\ \forall s \neq t.\, i_s \neq i_t}} a_{i_1,j_1} \cdots a_{i_m,j_m} \cdot e_{i_1} \cdots e_{i_m} + \sum_{\substack{i_1,\ldots,i_m \\ \exists s \neq t.\, i_s = i_t}} a_{i_1,j_1} \cdots a_{i_m,j_m} \cdot e_{i_1} \cdots e_{i_m}. \tag{193}$$

Our claim is equivalent to the vanishing of the second term. Note that the above equation for $b_j$ already shows the claim for $m = 1$. The case $m = 0$ is trivial.

We now prove the claim for $m = 2$ by hand before doing induction after. Recall that $j_1 \neq j_2$:

$$b_{j_1} b_{j_2} = \sum_{\substack{i_1,i_2 \\ i_1 \neq i_2}} a_{i_1,j_1} a_{i_2,j_2} \cdot e_{i_1} e_{i_2} + \sum_{i=1}^{n} a_{i,j_1} a_{i,j_2} \cdot e_i e_i \tag{194}$$

$$= \sum_{\substack{i_1,i_2 \\ i_1 \neq i_2}} a_{i_1,j_1} a_{i_2,j_2} \cdot e_{i_1} e_{i_2} + \sum_{i=1}^{n} a_{i,j_1} \underbrace{a_{i,j_2} \cdot \mathfrak{q}(e_i)}_{= \mathfrak{b}(e_i, b_{j_2})} \tag{195}$$

$$= \sum_{\substack{i_1,i_2 \\ i_1 \neq i_2}} a_{i_1,j_1} a_{i_2,j_2} \cdot e_{i_1} e_{i_2} + \underbrace{\mathfrak{b}(b_{j_1}, b_{j_2})}_{=0} \tag{196}$$

$$= \sum_{\substack{i_1,i_2 \\ i_1 \neq i_2}} a_{i_1,j_1} a_{i_2,j_2} \cdot e_{i_1} e_{i_2} \tag{197}$$

$$\in \mathrm{Cl}^{(2)}(V, \mathfrak{q}). \tag{198}$$

This shows the claim for $m = 2$.

By way of induction we now assume that we have shown the claim until some $m \geq 2$, i.e. we have:

$$b_{j_1} \cdots b_{j_m} = \sum_{\substack{i_1,\ldots,i_m \\ \forall k \neq l.\, i_k \neq i_l}} a_{i_1,j_1} \cdots a_{i_m,j_m} e_{i_1} \cdots e_{i_m} \in \mathrm{Cl}^{(m)}(V, \mathfrak{q}). \tag{199}$$

Now consider another $b_j$ with $j := j_{m+1} \neq j_k$, $k = 1, \ldots, m$. We then get:

$$b_{j_1} \cdots b_{j_m} b_j = \left( \sum_{\substack{i_1,\ldots,i_m \\ \forall k \neq l.\ i_k \neq i_l}} a_{i_1,j_1} \cdots a_{i_m,j_m} e_{i_1} \cdots e_{i_m} \right) \left( \sum_{i=1}^{n} a_{i,j} e_i \right) \tag{200}$$

$$= \sum_{i=1}^{n} \sum_{\substack{i_1,\ldots,i_m \\ \forall k \neq l.\ i_k \neq i_l}} a_{i_1,j_1} \cdots a_{i_m,j_m} a_{i,j} e_{i_1} \cdots e_{i_m} e_i \tag{201}$$

$$= \sum_{i=1}^{n} \sum_{\substack{i_1,\ldots,i_m \\ \forall k \neq l.\ i_k \neq i_l \\ i \notin \{i_1,\ldots,i_m\}}} a_{i_1,j_1} \cdots a_{i_m,j_m} a_{i,j} e_{i_1} \cdots e_{i_m} e_i \tag{202}$$

$$+ \sum_{i=1}^{n} \sum_{\substack{i_1,\ldots,i_m \\ \forall k \neq l.\ i_k \neq i_l \\ i \in \{i_1,\ldots,i_m\}}} a_{i_1,j_1} \cdots a_{i_m,j_m} a_{i,j} e_{i_1} \cdots e_{i_m} e_i \tag{203}$$

$$= \sum_{\substack{i_1,\ldots,i_m,i_{m+1} \\ \forall k \neq l.\ i_k \neq i_l}} a_{i_1,j_1} \cdots a_{i_m,j_m} a_{i_{m+1},j_{m+1}} e_{i_1} \cdots e_{i_m} e_{i_{m+1}} \tag{204}$$

$$+ \sum_{i=1}^{n} \sum_{s=1}^{m} \sum_{\substack{i_1,\ldots,i_m \\ \forall k \neq l.\ i_k \neq i_l \\ i_s = i}} a_{i_1,j_1} \cdots a_{i_m,j_m} a_{i,j} e_{i_1} \cdots e_{i_m} e_i. \tag{205}$$

We have to show that the last term vanishes. The last term can be written as:

$$\sum_{i=1}^{n}\sum_{s=1}^{m}\sum_{\substack{i_1,\ldots,i_m\\ \forall k\neq l.\, i_k\neq i_l\\ i_s=i}} a_{i_1,j_1}\cdots a_{i_m,j_m}a_{i,j}\cdot e_{i_1}\cdots e_{i_m}e_i \tag{206}$$

$$=\sum_{i=1}^{n}\sum_{s=1}^{m}\sum_{\substack{i_1,\ldots,i_m\\ \forall k\neq l.\, i_k\neq i_l\\ i_s=i}} a_{i_1,j_1}\cdots a_{i_s,j_s}\cdots a_{i_m,j_m}a_{i,j}\cdot e_{i_1}\cdots e_{i_s}\cdots e_{i_m}e_i \tag{207}$$

$$=\sum_{i=1}^{n}\sum_{s=1}^{m}\sum_{\substack{i_1,\ldots,i_m\\ \forall k\neq l.\, i_k\neq i_l\\ i_s=i}} a_{i_1,j_1}\cdots a_{i_s,j_s}\cdots a_{i_m,j_m}a_{i,j}\cdot e_{i_1}\cdots \cancel{e_{i_s}}\cdots e_{i_m}e_{i_s}e_i\cdot(-1)^{m-s} \tag{208}$$

$$=\sum_{i=1}^{n}\sum_{s=1}^{m}\sum_{\substack{i_1,\ldots,i_m\\ \forall k\neq l.\, i_k\neq i_l\\ i_s=i}} a_{i_1,j_1}\cdots a_{i_s,j_s}\cdots a_{i_m,j_m}a_{i,j}\cdot e_{i_1}\cdots \cancel{e_{i_s}}\cdots e_{i_m}\cdot(-1)^{m-s}\mathfrak{q}(e_i) \tag{209}$$

$$=\sum_{s=1}^{m}\sum_{i=1}^{n}\sum_{\substack{i_1,\ldots,i_m\\ \forall k\neq l.\, i_k\neq i_l\\ i_s=i}} a_{i_1,j_1}\cdots a_{i_s,j_s}\cdots a_{i_m,j_m}a_{i,j}\cdot e_{i_1}\cdots \cancel{e_{i_s}}\cdots e_{i_m}\cdot(-1)^{m-s}\mathfrak{q}(e_i) \tag{210}$$

$$=\sum_{s=1}^{m}\sum_{\substack{i_1,\ldots,i_m\\ \forall k\neq l.\, i_k\neq i_l}} a_{i_1,j_1}\cdots a_{i_s,j_s}\cdots a_{i_m,j_m}a_{i_s,j}\cdot e_{i_1}\cdots \cancel{e_{i_s}}\cdots e_{i_m}\cdot(-1)^{m-s}\mathfrak{q}(e_{i_s}) \tag{211}$$

$$=\sum_{s=1}^{m}\sum_{\substack{i_1,\ldots,\cancel{i_s},\ldots,i_m\\ \forall k\neq l.\, i_k\neq i_l}}\sum_{\substack{i_s=1\\ i_s\notin\{i_1,..,\cancel{i_s},..,i_m\}}} a_{i_1,j_1}\cdots a_{i_s,j_s}\cdots a_{i_m,j_m}a_{i_s,j}\cdot e_{i_1}\cdots \cancel{e_{i_s}}\cdots e_{i_m}\cdot(-1)^{m-s}\mathfrak{q}(e_{i_s})$$
$$\tag{212}$$

$$=\sum_{s=1}^{m}\sum_{\substack{i_1,\ldots,\cancel{i_s},\ldots,i_m\\ \forall k\neq l.\, i_k\neq i_l}}\sum_{i_s=1}^{n} a_{i_1,j_1}\cdots a_{i_s,j_s}\cdots a_{i_m,j_m}a_{i_s,j}\cdot e_{i_1}\cdots \cancel{e_{i_s}}\cdots e_{i_m}\cdot(-1)^{m-s}\mathfrak{q}(e_{i_s}) \tag{213}$$

$$-\sum_{s=1}^{m}\sum_{\substack{i_1,\ldots,\cancel{i_s},\ldots,i_m\\ \forall k\neq l.\, i_k\neq i_l}}\sum_{i_s\in\{i_1,..,\cancel{i_s},..,i_m\}} a_{i_1,j_1}\cdots a_{i_s,j_s}\cdots a_{i_m,j_m}a_{i_s,j}\cdot e_{i_1}\cdots \cancel{e_{i_s}}\cdots e_{i_m}\cdot(-1)^{m-s}\mathfrak{q}(e_{i_s})$$
$$\tag{214}$$

$$=\sum_{s=1}^{m}\sum_{\substack{i_1,\ldots,\cancel{i_s},\ldots,i_m\\ \forall k\neq l.\, i_k\neq i_l}} a_{i_1,j_1}\cdots \cancel{a_{i_s,j_s}}\cdots a_{i_m,j_m}\cdot e_{i_1}\cdots \cancel{e_{i_s}}\cdots e_{i_m}\cdot(-1)^{m-s}\underbrace{\sum_{i_s=1}^{n} a_{i_s,j_s}a_{i_s,j}\mathfrak{q}(e_{i_s})}_{=\mathfrak{b}(b_j,b_{j_s})=0}$$
$$\tag{215}$$

$$-\sum_{s=1}^{m}\sum_{\substack{i_1,\ldots,\cancel{i_s},\ldots,i_m\\ \forall k\neq l.\, i_k\neq i_l}}\sum_{i_s\in\{i_1,..,\cancel{i_s},..,i_m\}} a_{i_1,j_1}\cdots a_{i_s,j_s}\cdots a_{i_m,j_m}a_{i_s,j}\cdot e_{i_1}\cdots \cancel{e_{i_s}}\cdots e_{i_m}\cdot(-1)^{m-s}\mathfrak{q}(e_{i_s})$$
$$\tag{216}$$

$$=-\sum_{s=1}^{m}\sum_{\substack{i_1,\ldots,\cancel{i_s},\ldots,i_m\\ \forall k\neq l.\, i_k\neq i_l}}\sum_{i\in\{i_1,..,\cancel{i_s},..,i_m\}} a_{i_1,j_1}\cdots a_{i,j_s}\cdots a_{i_m,j_m}a_{i,j}\cdot e_{i_1}\cdots \cancel{e_{i_s}}\cdots e_{i_m}\cdot(-1)^{m-s}\mathfrak{q}(e_i)$$
$$\tag{217}$$

$$=-\sum_{s=1}^{m}\sum_{\substack{t=1\\ t\neq s}}^{m}\sum_{\substack{i_1,..,\cancel{i_s},..,i_t,..,i_m\\ \forall k\neq l.\, i_k\neq i_l}} a_{i_1,j_1}\cdots a_{i_t,j_s}\cdots a_{i_m,j_m}a_{i_t,j}\cdot e_{i_1}\cdots \cancel{e_{i_s}}\cdots e_{i_m}\cdot(-1)^{m-s}\mathfrak{q}(e_{i_t}).$$
$$\tag{218}$$

Note that the positional index $t$ can occure before or after the positional index $s$. Depending of its position the elements $e_{i_t}$ will appear before or after the element $e_{i_s}$ in the product. We look at both cases separately and suppress the dots in between for readability.

First consider $t > s$:

$$\sum_{\substack{s=1 \\ t>s}}^{m} \sum_{t=1}^{m} \sum_{\substack{i_1,..,\cancel{i_s},..,i_t,..,i_m \\ \forall k \neq l.\, i_k \neq i_l}} a_{i_1,j_1} \cdot a_{i_t,j_s} \cdot a_{i_t,j_t} \cdot a_{i_m,j_m} a_{i_t,j} \cdot e_{i_1} \cdot \cancel{e_{i_s}} \cdot e_{i_t} \cdot e_{i_m} \cdot (-1)^{m-s} \mathfrak{q}(e_{i_t}) \quad (219)$$

$$= \sum_{\substack{s=1 \\ t>s}}^{m} \sum_{t=1}^{m} \sum_{\substack{i_1,..,\cancel{i_s},..,i_t,..,i_m \\ \forall k \neq l.\, i_k \neq i_l}} a_{i_1,j_1} \cdot a_{i_t,j_s} \cdot a_{i_t,j_t} \cdot a_{i_m,j_m} a_{i_t,j} \cdot (-1)^{t-2} e_{i_t} \cdot e_{i_1} \cdot \cancel{e_{i_s}} \cdot \cancel{e_{i_t}} \cdot e_{i_m} \cdot (-1)^{m-s} \mathfrak{q}(e_{i_t})$$

$$(220)$$

$$= \sum_{\substack{s=1 \\ t>s}}^{m} \sum_{t=1}^{m} \sum_{i_t=1}^{n} a_{i_t,j_s} \cdot a_{i_t,j_t} \cdot a_{i_t,j} (-1)^{m-s} \mathfrak{q}(e_{i_t})(-1)^t e_{i_t} \cdot \pi_{s,t}(i_t) \quad (221)$$

with $\quad \pi_{s,t}(i) := \sum_{\substack{i_1,..,\cancel{i_s},..,\cancel{i_t},..,i_m \\ \forall k \neq l.\, i_k \neq i_l \\ \forall k.\, i_k \neq i}} a_{i_1,j_1} \cdot \cancel{a_{i_t,j_s}} \cdot \cancel{a_{i_t,j_t}} \cdot a_{i_m,j_m} \cdot e_{i_1} \cdot \cancel{e_{i_s}} \cdot \cancel{e_{i_t}} \cdot e_{i_m} \quad (222)$

$$= \sum_{\substack{s=1 \\ t>s}}^{m} \sum_{t=1}^{m} \sum_{i=1}^{n} (-1)^{m+s+t} a_{i,j_s} \cdot a_{i,j_t} \cdot a_{i,j} \cdot \mathfrak{q}(e_i) \cdot e_i \cdot \pi_{s,t}(i) \quad (223)$$

$$= \sum_{\substack{s=1 \\ t>s}}^{m} \sum_{t=1}^{m} y(s,t), \quad (224)$$

with $\quad y(s,t) := \sum_{i=1}^{n} (-1)^{m+s+t} a_{i,j_s} \cdot a_{i,j_t} \cdot a_{i,j} \cdot \mathfrak{q}(e_i) \cdot e_i \cdot \pi_{s,t}(i). \quad (225)$

It is important to note that for all $s \neq t$ we have:

$$y(s,t) = y(t,s). \quad (226)$$

Now consider $t < s$:

$$\sum_{\substack{s=1 \\ t<s}}^{m} \sum_{\substack{t=1 \\ t<s}}^{m} \sum_{\substack{i_1,..,i_t,..,\cancel{j_s},..,i_m \\ \forall k\neq l.\ i_k \neq i_l}} a_{i_1,j_1} \cdot a_{i_t,j_t} \cdot a_{i_t,j_s} \cdot a_{i_m,j_m} a_{i_t,j} \cdot e_{i_1} \cdot e_{i_t} \cdot \cancel{e_{i_s}} \cdot e_{i_m} \cdot (-1)^{m-s} \mathfrak{q}(e_{i_t}) \quad (227)$$

$$= \sum_{\substack{s=1 \\ t<s}}^{m} \sum_{\substack{t=1 \\ t<s}}^{m} \sum_{\substack{i_1,..,i_t,..,\cancel{j_s},..,i_m \\ \forall k\neq l.\ i_k \neq i_l}} a_{i_1,j_1} \cdot a_{i_t,j_t} \cdot a_{i_t,j_s} \cdot a_{i_m,j_m} a_{i_t,j} \cdot (-1)^{t-1} e_{i_t} \cdot e_{i_1} \cdot \cancel{e_{i_t}} \cdot \cancel{e_{i_s}} \cdot e_{i_m} \cdot (-1)^{m-s} \mathfrak{q}(e_{i_t})$$

$$\hspace{12cm} (228)$$

$$= \sum_{\substack{s=1 \\ t<s}}^{m} \sum_{\substack{t=1 \\ t<s}}^{m} \sum_{i_t=1}^{n} a_{i_t,j_s} \cdot a_{i_t,j_t} \cdot a_{i_t,j}(-1)^{m-s}\mathfrak{q}(e_{i_t})(-1)^{t-1} e_{i_t} \cdot \pi_{s,t}(i_t) \quad (229)$$

$$= \sum_{\substack{s=1 \\ t<s}}^{m} \sum_{\substack{t=1 \\ t<s}}^{m} \sum_{i=1}^{n} (-1)^{m+s+t+1} a_{i,j_s} \cdot a_{i,j_t} \cdot a_{i,j} \cdot \mathfrak{q}(e_i) \cdot e_i \cdot \pi_{s,t}(i) \quad (230)$$

$$= -\sum_{\substack{s=1 \\ t<s}}^{m} \sum_{\substack{t=1 \\ t<s}}^{m} y(s,t) \quad (231)$$

$$= -\sum_{\substack{t=1 \\ s<t}}^{m} \sum_{\substack{s=1 \\ s<t}}^{m} y(t,s) \quad (232)$$

$$= -\sum_{\substack{t=1 \\ s<t}}^{m} \sum_{\substack{s=1 \\ s<t}}^{m} y(s,t) \quad (233)$$

$$= -\sum_{\substack{s=1 \\ t>s}}^{m} \sum_{\substack{t=1 \\ t>s}}^{m} y(s,t). \quad (234)$$

In total we see that both terms appear with a different sign and thus cancel out. This shows the claim. $\qquad\square$

**Corollary D.28.** *Let $e_1, \ldots, e_n$ and $b_1, \ldots, b_n$ be two orthogonal bases of $(V, \mathfrak{q})$ with basis transition matrix $C = (c_{i,j})_{i\in[n],j\in[n]}$:*

$$\forall j \in [n]. \qquad b_j = \sum_{i\in[n]} c_{i,j} \cdot e_i. \quad (235)$$

*Then $C$ is invertible and we have the following matrix relations:*

$$\mathrm{diag}(\mathfrak{q}(b_1), \ldots, \mathfrak{q}(b_n)) = C^\top \mathrm{diag}(\mathfrak{q}(e_1), \ldots, \mathfrak{q}(e_n))C. \quad (236)$$

*Furthermore, for every subset $J \subseteq [n]$ we have the formula:*

$$b_J = \sum_{\substack{I\subseteq[n] \\ |I|=|J|}} \det C_{I,J} \cdot e_I, \quad (237)$$

*with the submatrix: $C_{I,J} = (c_{i,j})_{i\in I,j\in J}$.*

*Proof.* For $j, l \in [n]$ we have:

$$\mathrm{diag}(\mathfrak{q}(b_1), \ldots, \mathfrak{q}(b_n))_{j,l} = \mathfrak{q}(b_j) \cdot \delta_{j,l} \tag{238}$$

$$= \mathfrak{b}(b_j, b_l) \tag{239}$$

$$= \sum_{i=1}^{n} \sum_{k=1}^{n} c_{i,j} \cdot c_{k,l} \cdot \mathfrak{b}(e_i, e_k) \tag{240}$$

$$= \sum_{i=1}^{n} \sum_{k=1}^{n} c_{i,j} \cdot c_{k,l} \cdot \mathfrak{q}(e_i) \cdot \delta_{i,k} \tag{241}$$

$$= \left( C^\top \mathrm{diag}(\mathfrak{q}(e_1), \ldots, \mathfrak{q}(e_n)) C \right)_{j,l}. \tag{242}$$

This shows the matrix identity:

$$\mathrm{diag}(\mathfrak{q}(b_1), \ldots, \mathfrak{q}(b_n)) = C^\top \mathrm{diag}(\mathfrak{q}(e_1), \ldots, \mathfrak{q}(e_n)) C. \tag{243}$$

Furthermore, we have the following identites:

$$b_J = b_{j_1} \cdots b_{j_m} \tag{244}$$

$$= \left( \sum_{i_1 \in [n]} c_{i_1,j_1} \cdot e_{i_1} \right) \cdots \left( \sum_{i_m \in [n]} c_{i_m,j_m} \cdot e_{i_m} \right) \tag{245}$$

$$= \sum_{i_1 \in [n], \ldots, i_m \in [n]} (c_{i_1,j_1} \cdots c_{i_m,j_m}) \cdot (e_{i_1} \cdots e_{i_m}) \tag{246}$$

$$\overset{D.27}{=} \sum_{\substack{i_1 \in [n], \ldots, i_m \in [n] \\ |\{i_1, \ldots, i_m\}| = m}} (c_{i_1,j_1} \cdots c_{i_m,j_m}) \cdot (e_{i_1} \cdots e_{i_m}) \tag{247}$$

$$= \sum_{\substack{i_1 \in [n], \ldots, i_m \in [n] \\ |\{i_1, \ldots, i_m\}| = m}} \mathrm{sgn}(i_1, \ldots, i_m) \cdot (c_{i_1,j_1} \cdots c_{i_m,j_m}) \cdot e_{\{i_1, \ldots, i_m\}} \tag{248}$$

$$= \sum_{\substack{i_1, \ldots, i_m \in [n] \\ i_1 < \cdots < i_m}} \left( \sum_{\sigma \in \mathrm{S}_m} \mathrm{sgn}(\sigma) \cdot c_{i_{\sigma(1)}, j_1} \cdots c_{i_{\sigma(m)}, j_m} \right) \cdot e_{\{i_1, \ldots, i_m\}} \tag{249}$$

$$= \sum_{\substack{I \subseteq [n] \\ |I| = m}} \det C_{I,J} \cdot e_I. \tag{250}$$

This shows the claim. $\qquad \square$

**Notation D.29.** *For $m \notin \{0, \ldots, n\}$ it is sometimes convenient to put:*

$$\mathrm{Cl}^{(m)}(V, \mathfrak{q}) := 0. \tag{251}$$

**Corollary D.30.** *We have the following orthogonal sum decomposition (w.r.t. $\bar{\mathfrak{q}}$) of the Clifford algebra $\mathrm{Cl}(V, \mathfrak{q})$ into its $\mathbb{F}$-vector spaces of multivector components:*

$$\mathrm{Cl}(V, \mathfrak{q}) = \bigoplus_{m=0}^{n} \mathrm{Cl}^{(m)}(V, \mathfrak{q}), \tag{252}$$

*which is independent of the choice of orthogonal basis of $(V, \mathfrak{q})$. Also note that for all $m = 0, \ldots, n$:*

$$\dim \mathrm{Cl}^{(m)}(V, \mathfrak{q}) = \binom{n}{m}. \tag{253}$$

**Definition D.31.** *We call an element $x \in \mathrm{Cl}^{(m)}(V, \mathfrak{q})$ an $m$-multivector or an element of $\mathrm{Cl}(V, \mathfrak{q})$ of pure grade $m$. For $x \in \mathrm{Cl}(V, \mathfrak{q})$ we have a decomposition:*

$$x = x^{(0)} + x^{(1)} + \cdots + x^{(n)}, \tag{254}$$

*with $x^{(m)} \in \mathrm{Cl}^{(m)}(V, \mathfrak{q})$, $m = 0, \ldots, n$. We call $x^{(m)}$ the grade-$m$-component of $x$.*

**Remark D.32.** *Note that the multivector grading of $\mathrm{Cl}(V, \mathfrak{q})$ is only a grading of $\mathbb{F}$-vector spaces, but not of $\mathbb{F}$-algebras. The reason is that multiplication can make the grade drop. For instance, for $v \in V = \mathrm{Cl}^{(1)}(V, \mathfrak{q})$ we have $vv = \mathfrak{q}(v) \in \mathrm{Cl}^{(0)}(V, \mathfrak{q})$, while a grading for algebras would require that $vv \in \mathrm{Cl}^{(2)}(V, \mathfrak{q})$, which is here not the case.*

**Remark D.33** (Parity grading and multivector filtation in terms of multivector grading)**.**

1. *We clearly have:*

$$\mathrm{Cl}^{[0]}(V, \mathfrak{q}) = \bigoplus_{\substack{m=0,\ldots,n \\ m \text{ even}}} \mathrm{Cl}^{(m)}(V, \mathfrak{q}), \qquad \mathrm{Cl}^{[1]}(V, \mathfrak{q}) = \bigoplus_{\substack{m=1,\ldots,n \\ m \text{ odd}}} \mathrm{Cl}^{(m)}(V, \mathfrak{q}). \qquad (255)$$

2. *It is also clear that for general $m$ we have:*

$$\mathrm{Cl}^{(\leq m)}(V, \mathfrak{q}) = \bigoplus_{l=0}^{m} \mathrm{Cl}^{(m)}(V, \mathfrak{q}) \quad \subseteq \quad \mathrm{Cl}(V, \mathfrak{q}). \qquad (256)$$

A simpler proof of Theorem D.27 can be obtained if we assume that $\mathrm{char}(\mathbb{F}) = 0$. We would then argue as follows.

**Definition D.34** (Antisymmetrization)**.** *For $m \in \mathbb{N}$ and $x_1, \ldots, x_m \in \mathrm{Cl}(V, \mathfrak{q})$ we define their antisymmetrization as:*

$$[x_1; \ldots; x_m] := \frac{1}{m!} \sum_{\sigma \in \mathrm{S}_m} \mathrm{sgn}(\sigma) \cdot x_{\sigma(1)} \cdots x_{\sigma(m)}, \qquad (257)$$

*where $\mathrm{S}_m$ denotes the group of all permuations of $[m]$. Note that, due to the division by $m!$ we need that $\mathrm{char}(\mathbb{F}) = 0$ if we want to accommodate arbitrary $m \in \mathbb{N}$.*

**Lemma D.35.** *Let $e_1, \ldots, e_n$ be an orthogonal basis of $(V, \mathfrak{q})$ and $x_1, \ldots, x_m \in \mathrm{Cl}(V, \mathfrak{q})$. Then we have:*

1. *$[x_1; \ldots; x_m]$ is linear in each of its arguments (if the other arguments are fixed).*

2. *$[x_1; \ldots; x_k; \ldots; x_l; \ldots; x_m] = -[x_1; \ldots; x_l; \ldots; x_k; \ldots; x_m]$.*

3. *$[x_1; \ldots; x_k; \ldots; x_l; \ldots; x_m] = 0$ if $x_k = x_l$.*

4. *$[e_{i_1}; \ldots; e_{i_m}] = e_{i_1} \cdots e_{i_m}$ if $|\{i_1, \ldots, i_m\}| = m$ (i.e. if all indices are different).*

**Definition D.36** (Multivector grading - alternative, basis independent definition)**.** *For $m = 0, \ldots, n$ we (re-)define:*

$$\mathrm{Cl}^{(m)}(V, \mathfrak{q}) := \mathrm{span}\left\{[v_1; \ldots; v_m] \mid v_1, \ldots, v_m \in V\right\}. \qquad (258)$$

**Theorem D.37.** *Let $e_1, \ldots, e_n$ be an orthogonal basis of $(V, \mathfrak{q})$, $\mathrm{char}(\mathbb{F}) = 0$. Then for every $m = 0, \ldots, n$ we have the equality:*

$$\mathrm{Cl}^{(m)}(V, \mathfrak{q}) = \mathrm{span}\left\{e_A \mid A \subseteq [n], |A| = m\right\}. \qquad (259)$$

*Note that the rhs is seemingly dependent of the choice of the orthogonal basis while the lhs is defined in a basis independent way.*

*Proof.* We can write every $v_k \in V$ as a linear combination of its basis vectors:

$$v_k = \sum_{j_k \in [n]} c_{k,j_k} \cdot e_{j_k}. \qquad (260)$$

With this we get:

$$[v_1; \ldots; v_m] = \sum_{j_1,\ldots,j_m \in [n]} \prod_{k \in [m]} c_{k,j_k} \cdot [e_{j_1}; \ldots; e_{j_m}] \tag{261}$$

$$= \sum_{\substack{j_1,\ldots,j_m \in [n] \\ |\{j_1,\ldots,j_m\}|=m}} \prod_{k \in [m]} c_{k,j_k} \cdot [e_{j_1}; \ldots; e_{j_m}] \tag{262}$$

$$= \sum_{\substack{j_1,\ldots,j_m \in [n] \\ |\{j_1,\ldots,j_m\}|=m}} \prod_{k \in [m]} c_{k,j_k} \cdot e_{j_1} \cdots e_{j_m} \tag{263}$$

$$= \sum_{\substack{j_1,\ldots,j_m \in [n] \\ |\{j_1,\ldots,j_m\}|=m}} \pm \prod_{k \in [m]} c_{k,j_k} \cdot e_{\{j_1,\ldots,j_m\}} \tag{264}$$

$$\in \mathrm{span} \left\{ e_A \mid A \subseteq [n], |A| = m \right\}. \tag{265}$$

This shows the inclusion:

$$\mathrm{Cl}^{(m)}(V, \mathfrak{q}) \subseteq \mathrm{span} \left\{ e_A \mid A \subseteq [n], |A| = m \right\}. \tag{266}$$

The reverse inclusion is also clear as:

$$e_A = [e_{j_1}; \ldots; e_{j_m}] \in \mathrm{Cl}^{(m)}(V, \mathfrak{q}), \tag{267}$$

where $A = \{j_1, \ldots, j_m\}$ and $|A| = m$. This shows the equality of both sets. $\qquad \square$

### D.7 The Radical Subalgebra of the Clifford Algebra

Again, let $(V, \mathfrak{q})$ be a quadratic vector space of finite dimensions $\dim V = n < \infty$ over a field $\mathbb{F}$ of $\mathrm{char}(\mathbb{F}) \neq 2$. Let $\mathfrak{b}$ the corresponding bilinear form of $\mathfrak{q}$.

**Notation D.38.** *We denote the group of the* invertible elements *of* $\mathrm{Cl}(V, \mathfrak{q})$:

$$\mathrm{Cl}^{\times}(V, \mathfrak{q}) := \left\{ x \in \mathrm{Cl}(V, \mathfrak{q}) \mid \exists y \in \mathrm{Cl}(V, \mathfrak{q}). \, xy = yx = 1 \right\}. \tag{268}$$

Let $\mathcal{R} \subseteq V$ be $V$'s radical subspace. Recall:

$$\mathcal{R} := \left\{ f \in V \mid \forall v \in V. \, \mathfrak{b}(f, v) = 0 \right\}. \tag{269}$$

**Definition D.39** (The radical subalgebra)**.** *We define the* radical subalgebra *of* $\mathrm{Cl}(V, \mathfrak{q})$ *to be:*

$$\bigwedge(\mathcal{R}) := \mathrm{span} \left\{ 1, f_1 \cdots f_k \mid k \in \mathbb{N}_0, f_l \in \mathcal{R}, l = 1, \ldots, k \right\} \subseteq \mathrm{Cl}(V, \mathfrak{q}). \tag{270}$$

*Note that* $\mathfrak{q}|_{\mathcal{R}} = 0$ *and that* $\bigwedge(\mathcal{R})$ *coincides with* $\mathrm{Cl}(\mathcal{R}, \mathfrak{q}|_{\mathcal{R}})$.

**Notation D.40.** *We make the following further abbreviations:*

$$\bigwedge^{[i]}(\mathcal{R}) := \bigwedge(\mathcal{R}) \cap \mathrm{Cl}^{[i]}(V, \mathfrak{q}), \tag{271}$$

$$\bigwedge^{(\geq 1)}(\mathcal{R}) := \mathrm{span} \left\{ f_1 \cdots f_k \mid k \geq 1, f_l \in \mathcal{R}, l = 1, \ldots, k \right\}, \tag{272}$$

$$\bigwedge^{\times}(\mathcal{R}) := \mathbb{F}^{\times} + \bigwedge^{(\geq 1)}(\mathcal{R}), \tag{273}$$

$$\bigwedge^{[\times]}(\mathcal{R}) := \mathbb{F}^{\times} + \mathrm{span} \left\{ f_1 \cdots f_k \mid k \geq 2 \text{ even}, f_l \in \mathcal{R}, l = 1, \ldots, k \right\}, \tag{274}$$

$$\bigwedge^{*}(\mathcal{R}) := 1 + \bigwedge^{(\geq 1)}(\mathcal{R}), \tag{275}$$

$$\bigwedge^{[*]}(\mathcal{R}) := 1 + \mathrm{span} \left\{ f_1 \cdots f_k \mid k \geq 2 \text{ even}, f_l \in \mathcal{R}, l = 1, \ldots, k \right\}. \tag{276}$$

Here, $\mathbb{F}^{\times}$ denotes the set of invertible elements of $\mathbb{F}$.

**Lemma D.41.** *1. For every* $h \in \bigwedge^{(\geq 1)}(\mathcal{R})$ *there exists a* $k \geq 0$ *such that:*

$$h^{k+1} = 0. \tag{277}$$

*In particular, no* $h \in \bigwedge^{(\geq 1)}(\mathcal{R})$ *is ever invertible.*

2. *Every $y \in \bigwedge^{\times}(\mathcal{R}) = \bigwedge(\mathcal{R}) \setminus \bigwedge^{(\geq 1)}(\mathcal{R})$ is invertible. Its inverse is given by:*

$$y^{-1} = c^{-1} \left(1 - h + h^2 - \cdots + (-1)^k h^k\right), \tag{278}$$

*where we write: $y = c \cdot (1 + h)$ with $c \in \mathbb{F}^{\times}$, $h \in \bigwedge^{(\geq 1)}(\mathcal{R})$, and $k$ is such that $h^{k+1} = 0$.*

3. *In particular, we get:*

$$\bigwedge^{\times}(\mathcal{R}) = \bigwedge(\mathcal{R}) \cap \mathrm{Cl}^{\times}(V, \mathfrak{q}). \tag{279}$$

*Proof.* Items 1 and 3 are clear. For item 2 we refer to Example E.21. $\qquad\square$

**Lemma D.42** (Twisted commutation relationships). *1. For every $f \in \mathcal{R}$ and $v \in V$ we have the anticommutation relationship:*

$$fv = -vf + 2\underbrace{\mathfrak{b}(f, v)}_{=0} = -vf. \tag{280}$$

2. *For every $f \in \mathcal{R}$ and $x \in \mathrm{Cl}(V, \mathfrak{q})$ we get the following twisted commutation relationship:*

$$\alpha(x)f = (x^{[0]} - x^{[1]})f = f(x^{[0]} + x^{[1]}) = fx. \tag{281}$$

3. *For every $y \in \bigwedge(\mathcal{R})$ and every $x \in \mathrm{Cl}^{[0]}(V, \mathfrak{q})$ we get:*

$$xy = yx. \tag{282}$$

4. *For every $y \in \bigwedge(\mathcal{R})$ and $v \in V$ we get:*

$$\alpha(y)v = (y^{[0]} - y^{[1]})v = v(y^{[0]} + y^{[1]}) = vy. \tag{283}$$

5. *For every $y \in \bigwedge^{[0]}(\mathcal{R})$ (of even parity) and every $x \in \mathrm{Cl}(V, \mathfrak{q})$ we get:*

$$yx = xy. \tag{284}$$

**Remark D.43.** *A direct consequence from Lemma D.42 is that the even parity parts: $\bigwedge^{[0]}(\mathcal{R})$, $\bigwedge^{[\times]}(\mathcal{R})$ and $\bigwedge^{[*]}(\mathcal{R})$ all lie in the center of $\mathrm{Cl}(V, \mathfrak{q})$, which we denote by $\mathfrak{Z}(\mathrm{Cl}(V, \mathfrak{q}))$:*

$$\bigwedge^{[*]}(\mathcal{R}) \subseteq \bigwedge^{[\times]}(\mathcal{R}) \subseteq \bigwedge^{[0]}(\mathcal{R}) \subseteq \mathfrak{Z}(\mathrm{Cl}(V, \mathfrak{q})), \tag{285}$$

*i.e. every $y \in \bigwedge^{[0]}(\mathcal{R})$ (of even parity) commutes with every $x \in \mathrm{Cl}(V, \mathfrak{q})$. For more detail we refer to Theorem D.47.*

In the following we will study the center of the Clifford algebra $\mathfrak{Z}(\mathrm{Cl}(V, \mathfrak{q}))$ more carefully.

## D.8 The Center of the Clifford Algebra

In the following final subsections, we study additional properties of the Clifford algebra that will aid us in studying group representations and actions on the algebra in the upcoming sections.

Let $(V, \mathfrak{q})$ be a quadratic space over a field $\mathbb{F}$ with $\mathrm{char}(\mathbb{F}) \neq 2$ and $\dim V = n < \infty$.

**Definition D.44** (The center of an algebra). *The* center *of an algebra $\mathcal{A}$ is defined to be:*

$$\mathfrak{Z}(\mathcal{A}) := \{z \in \mathcal{A} \mid \forall x \in \mathcal{A}. \, xz = zx\}. \tag{286}$$

**Lemma D.45.** *Let $e_1, \ldots, e_n$ be an orthogonal basis of $(V, \mathfrak{q})$. For $A \subseteq [n] := \{1, \ldots, n\}$ let $e_A := \prod_{i \in A}^{<} e_i$ be the product in $\mathrm{Cl}(V, \mathfrak{q})$ in increasing index order, $e_{\emptyset} := 1$. Then we get for two subsets $A, B \subseteq [n]$:*

$$e_A e_B = (-1)^{|A| \cdot |B| - |A \cap B|} \cdot e_B e_A. \tag{287}$$

*In particular, for $j \notin A$ we get:*

$$e_A e_j = (-1)^{|A|} \cdot e_j e_A, \tag{288}$$

*and:*

$$e_A e_j - e_j e_A = ((-1)^{|A|} - 1) \cdot e_j e_A = (-1)^t ((-1)^{|A|+1} + 1) e_{A \dot\cup \{j\}}, \tag{289}$$

*where $t$ is the position of $j$ in the ordered set $A \dot\cup \{j\}$.*

*For $i \in A$ we get:*

$$e_A e_i = (-1)^{|A|-1} \cdot e_i e_A = (-1)^{|A|-s} \mathfrak{q}(e_i) e_{A \setminus \{i\}}, \tag{290}$$

*and:*

$$e_A e_i - e_i e_A = (-1)^s ((-1)^{|A|} + 1) \mathfrak{q}(e_i) e_{A \setminus \{i\}}, \tag{291}$$

*where $s$ is the position of $i$ in the ordered set $A$.*

*Proof.* Let $B := \{j_1, \ldots, j_{|B|}\} \subseteq [n]$.

$$e_A e_B = \prod_{i \in A}^{<} e_i \prod_{j \in B}^{<} e_j \tag{292}$$

$$= (-1)^{|A| - \mathbb{1}[j_1 \in A]} e_{j_1} \prod_{i \in A}^{<} e_i \prod_{j \in B \setminus j_1}^{<} e_j \tag{293}$$

$$= (-1)^{|B||A| - \sum_{j \in B} \mathbb{1}[j \in A]} e_B e_A \tag{294}$$

$$= (-1)^{|B||A| - |A \cap B|} e_B e_A \tag{295}$$

$\square$

For the other two identities, we similarly make use of the fundamental Clifford identity.

**Lemma D.46.** *Let $(V, \mathfrak{q})$ be a quadratic space with $\dim V = n < \infty$. Let $e_1, \ldots, e_n$ be an orthogonal basis for $(V, \mathfrak{q})$. For $x \in \mathrm{Cl}(V, \mathfrak{q})$ we have the equivalence:*

$$x \in \mathfrak{z}(\mathrm{Cl}(V, \mathfrak{q})) \qquad \Longleftrightarrow \qquad \forall i \in [n]. \quad x e_i = e_i x. \tag{296}$$

*Proof.* This is clear as $\mathrm{Cl}(V, \mathfrak{q})$ is generated by $e_{k_1} \cdots e_{k_l}$.

$\square$

**Theorem D.47** (The center of the Clifford algebra)**.** *Let $(V, \mathfrak{q})$ be a quadratic space with $\dim V = n < \infty$, $\mathrm{char}\,\mathbb{F} \neq 2$, and let $\mathcal{R} \subseteq V$ be the radical subspace of $(V, \mathfrak{q})$. Then for the center of $\mathrm{Cl}(V, \mathfrak{q})$ we have the following cases:*

1. *If $n$ is odd then:*

$$\mathfrak{z}(\mathrm{Cl}(V, \mathfrak{q})) = \bigwedge^{[0]}(\mathcal{R}) \oplus \mathrm{Cl}^{(n)}(V, \mathfrak{q}). \tag{297}$$

2. *If $n$ is even then:*

$$\mathfrak{z}(\mathrm{Cl}(V, \mathfrak{q})) = \bigwedge^{[0]}(\mathcal{R}). \tag{298}$$

*In all cases we have:*

$$\bigwedge^{[0]}(\mathcal{R}) \subseteq \mathfrak{z}(\mathrm{Cl}(V, \mathfrak{q})). \tag{299}$$

*Proof.* Let $e_1, \ldots, e_n$ be an orthogonal basis for $(V, \mathfrak{q})$. The statement can then equivalently be expressed as:

1. If $n$ is odd or $\mathfrak{q} = 0$ (on all vectors), then:
$$\mathfrak{Z}(\mathrm{Cl}(V, \mathfrak{q})) = \mathrm{span}\left\{1, e_A, e_{[n]} \,\middle|\, |A| \text{ even}, \forall i \in A.\, \mathfrak{q}(e_i) = 0\right\} \tag{300}$$
$$= \bigwedge\nolimits^{[0]}(\mathcal{R}) + \mathrm{Cl}^{(n)}(V, \mathfrak{q}). \tag{301}$$

2. If $n$ is even and $\mathfrak{q} \neq 0$ (on some vector), then:
$$\mathfrak{Z}(\mathrm{Cl}(V, \mathfrak{q})) = \mathrm{span}\left\{1, e_A \,\middle|\, |A| \text{ even}, \forall i \in A.\, \mathfrak{q}(e_i) = 0\right\} \tag{302}$$
$$= \bigwedge\nolimits^{[0]}(\mathcal{R}). \tag{303}$$

Note that if $n$ is even and $\mathfrak{q} = 0$ then both points would give the same answer, as then $V = \mathcal{R}$ and thus:
$$\mathrm{Cl}^{(n)}(V, \mathfrak{q}) = \bigwedge\nolimits^{(n)}(\mathcal{R}) \subseteq \bigwedge\nolimits^{[0]}(\mathcal{R}). \tag{304}$$

We first only consider basis elements $e_A$ with $A \subseteq [n]$ and check when we have $e_A \in \mathfrak{Z}(\mathrm{Cl}(V, \mathfrak{q}))$.

We now consider three cases:

1. $A = \emptyset$. We always have: $e_\emptyset = 1 \in \mathfrak{Z}(\mathrm{Cl}(V, \mathfrak{q}))$.

2. $A = [n]$. For every $i \in [n]$ we have:
$$e_{[n]}e_i = (-1)^{n-1} \cdot e_i e_{[n]} = \pm \mathfrak{q}(e_i) \cdot e_{[n] \setminus \{i\}}. \tag{305}$$
So $e_{[n]} \in \mathfrak{Z}(\mathrm{Cl}(V, \mathfrak{q}))$ iff either $n$ is odd or $\mathfrak{q} = 0$ (on all $e_i$).

3. $\emptyset \neq A \subsetneq [n]$. Note that for $i \in A$ we have:
$$e_A e_i = (-1)^{|A|-1} \cdot e_i e_A = \pm \mathfrak{q}(e_i) \cdot e_{A \setminus \{i\}}, \tag{306}$$
while for $j \notin A$ we have:
$$e_A e_j = (-1)^{|A|} \cdot e_j e_A. \tag{307}$$
The latter implies that for $e_A \in \mathfrak{Z}(\mathrm{Cl}(V, \mathfrak{q}))$ to hold, $|A|$ necessarily needs to be even. Since in that case $(-1)^{|A|-1} = -1$, we necessarily need by the former case that for every $i \in A$, $\mathfrak{q}(e_i) = 0$.

Now consider a linear combination $x = \sum_{A \subseteq [n]} c_A \cdot e_A \in \mathrm{Cl}(V, \mathfrak{q})$ with $c_A \in \mathbb{F}$. Then abbreviate:
$$\mathcal{Z} := \mathrm{span}\left\{e_A \,\middle|\, A \subseteq [n], e_A \in \mathfrak{Z}(\mathrm{Cl}(V, \mathfrak{q}))\right\} \subseteq \mathfrak{Z}(\mathrm{Cl}(V, \mathfrak{q})). \tag{308}$$
We need to show that $x \in \mathcal{Z}$. We thus write:
$$x = y + z, \qquad y := \sum_{\substack{A \subseteq [n] \\ e_A \notin \mathcal{Z}}} c_A \cdot e_A, \qquad z := \sum_{\substack{A \subseteq [n] \\ e_A \in \mathcal{Z}}} c_A \cdot e_A \in \mathcal{Z}. \tag{309}$$
So with $x, z \in \mathfrak{Z}(\mathrm{Cl}(V, \mathfrak{q}))$ also $y \in \mathfrak{Z}(\mathrm{Cl}(V, \mathfrak{q}))$ and we are left to show that $y = 0$.

First note that, since $e_\emptyset = 1 \in \mathcal{Z}$, there is no $e_\emptyset$-component in $y$.

We now have for every $i \in [n]$:
$$0 = ye_i - e_i y \tag{310}$$
$$= \sum_{\substack{A \subseteq [n] \\ e_A \notin \mathcal{Z}}} c_A \cdot (e_A e_i - e_i e_A) \tag{311}$$
$$= \sum_{\substack{A \subseteq [n] \\ e_A \notin \mathcal{Z} \\ i \in A}} c_A \cdot (e_A e_i - e_i e_A) + \sum_{\substack{A \subseteq [n] \\ e_A \notin \mathcal{Z} \\ i \notin A}} c_A \cdot (e_A e_i - e_i e_A) \tag{312}$$
$$= \sum_{\substack{A \subseteq [n] \\ e_A \notin \mathcal{Z} \\ i \in A}} \pm((-1)^{|A|} + 1)\mathfrak{q}(e_i) c_A \cdot e_{A \setminus \{i\}} + \sum_{\substack{A \subseteq [n] \\ e_A \notin \mathcal{Z} \\ i \notin A}} \pm((-1)^{|A|+1} + 1) c_A \cdot e_{A \,\dot\cup\, \{i\}}. \tag{313}$$

Note that for $A, B \subseteq [n]$ with $A \neq B$ we always have:

$$A \setminus \{i\} \neq B \setminus \{i\}, \qquad \text{if } i \in A, i \in B, \qquad (314)$$
$$A \setminus \{i\} \neq B \,\dot\cup\, \{i\}, \qquad \text{if } i \in A, i \notin B, \qquad (315)$$
$$A \,\dot\cup\, \{i\} \neq B \setminus \{i\}, \qquad \text{if } i \notin A, i \in B, \qquad (316)$$
$$A \,\dot\cup\, \{i\} \neq B \,\dot\cup\, \{i\}, \qquad \text{if } i \notin A, i \notin B. \qquad (317)$$

So the above representation for $ye_i - e_i y$ is already given in basis form. By their linear independence we then get that for every $A \subseteq [n]$ with $e_A \notin \mathcal{Z}$ and every $i \in [n]$:

$$0 = ((-1)^{|A|} + 1)\mathfrak{q}(e_i)c_A, \qquad \text{for } i \in A, \qquad (318)$$
$$0 = ((-1)^{|A|+1} + 1)c_A, \qquad \text{for } i \notin A. \qquad (319)$$

First consider the case $e_{[n]} \notin \mathcal{Z}$. By the previous result we then know that $n$ is even and $\mathfrak{q} \neq 0$. So there exists $e_i$ with $\mathfrak{q}(e_i) \neq 0$. So the above condition for $i \in [n]$ then reads:

$$0 = 2\mathfrak{q}(e_i)c_{[n]}, \qquad \text{for } i \in [n], \qquad (320)$$

which implies $c_{[n]} = 0$ as $\mathfrak{q}(e_i) \neq 0$. So $y$ does not have a $e_{[n]}$-component.

Similarly, for $A \subseteq [n]$ with $A \neq [n]$ and $e_A \notin \mathcal{Z}$ and $|A|$ odd there exists $i \notin A$ and the above condition reads:

$$0 = 2c_A, \qquad \text{for } i \notin A, \qquad (321)$$

which implies $c_A = 0$. So $y$ does not have any $e_A$-components with odd $|A|$.

Now let $A \subseteq [n]$ with $A \neq [n]$ and $e_A \notin \mathcal{Z}$ and $|A|$ even. Then by our previous analysis we know that there exists $i \in A$ with $\mathfrak{q}(e_i) \neq 0$. Otherwise $e_A \in \mathcal{Z}$. So the above condition reads:

$$0 = 2\mathfrak{q}(e_i)c_A, \qquad \text{for } i \in A, \qquad (322)$$

which implies $c_A = 0$ as $\mathfrak{q}(e_i) \neq 0$. This shows that $y$ does not have any $e_A$-component with even $|A|$.

Overall, this shows that $y = 0$ and thus the claim. $\qquad\square$

### D.9  The Twisted Center of the Clifford Algebra

**Notation D.48.** *Let $e_1, \ldots, e_n$ be an orthogonal basis of $(V, \mathfrak{q})$. For $A \subseteq [n] := \{1, \ldots, n\}$ let $e_A := \prod_{i \in A}^{<} e_i$ be the product in $\mathrm{Cl}(V, \mathfrak{q})$ in increasing index order, $e_\emptyset := 1$. Then $(e_A)_{A \subseteq [n]}$ forms a basis for $\mathrm{Cl}(V, \mathfrak{q})$.*

**Definition D.49** (The twisted center of a $\mathbb{Z}/2\mathbb{Z}$-graded algebra)**.** *We define the* twisted center *of a $\mathbb{Z}/2\mathbb{Z}$-graded algebra $\mathcal{A}$ as the following subset:*

$$\mathfrak{K}(\mathcal{A}) := \left\{ y \in \mathcal{A} \,\middle|\, \forall x \in \mathcal{A}.\ yx^{[0]} + (y^{[0]} - y^{[1]})x^{[1]} = xy \right\}. \qquad (323)$$

**Theorem D.50.** *We have the following identification of the twisted center with the radical subalgebra of the Clifford algebra $\mathrm{Cl}(V, \mathfrak{q})$ and the set:*

$$\mathfrak{K}(\mathrm{Cl}(V, \mathfrak{q})) = \bigwedge(\mathcal{R}) = \{y \in \mathrm{Cl}(V, \mathfrak{q}) \,|\, \forall v \in V.\ \alpha(y)v = vy\}. \qquad (324)$$

*Proof.* Let $y \in \bigwedge(\mathcal{R})$ then by Lemma D.42 we get:

$$yx^{[0]} + \alpha(y)x^{[1]} = x^{[0]}y + x^{[1]}y = xy. \qquad (325)$$

This shows that:

$$y \in \mathfrak{K}(\mathrm{Cl}(V, \mathfrak{q})), \qquad (326)$$

and thus:

$$\bigwedge(\mathcal{R}) \subseteq \mathfrak{K}(\mathrm{Cl}(V, \mathfrak{q})). \qquad (327)$$

Note that the following inclusion is clear as $V \subseteq \mathrm{Cl}(V, \mathfrak{q})$:

$$\mathfrak{K}(\mathrm{Cl}(V, \mathfrak{q})) \subseteq \{y \in \mathrm{Cl}(V, \mathfrak{q}) \mid \forall v \in V. \ \alpha(y)v = vy\}. \tag{328}$$

For the final inclusion, let $y = \sum_{B \subseteq [n]} c_B \cdot e_B \in \mathrm{Cl}(V, \mathfrak{q})$ such that for all $v \in V$ we have $\alpha(y)v = vy$. Then for all orthogonal basis vectors $e_i$ we get the requirement:

$$e_i y = \alpha(y)e_i, \tag{329}$$

which always holds if $\mathfrak{q}(e_i) = 0$, and is only a condition for $\mathfrak{q}(e_i) \neq 0$. For such $e_i$ we get:

$$e_i \left( \sum_{\substack{B \subseteq [n] \\ |B| \text{ even} \\ i \in B}} c_B \cdot e_B + \sum_{\substack{B \subseteq [n] \\ |B| \text{ even} \\ i \notin B}} c_B \cdot e_B + \sum_{\substack{B \subseteq [n] \\ |B| \text{ odd} \\ i \in B}} c_B \cdot e_B + \sum_{\substack{B \subseteq [n] \\ |B| \text{ odd} \\ i \notin B}} c_B \cdot e_B \right) \tag{330}$$

$$= e_i y \tag{331}$$

$$= \alpha(y)e_i \tag{332}$$

$$= \alpha \left( \sum_{\substack{B \subseteq [n] \\ |B| \text{ even} \\ i \in B}} c_B \cdot e_B + \sum_{\substack{B \subseteq [n] \\ |B| \text{ even} \\ i \notin B}} c_B \cdot e_B + \sum_{\substack{B \subseteq [n] \\ |B| \text{ odd} \\ i \in B}} c_B \cdot e_B + \sum_{\substack{B \subseteq [n] \\ |B| \text{ odd} \\ i \notin B}} c_B \cdot e_B \right) e_i \tag{333}$$

$$= \left( \sum_{\substack{B \subseteq [n] \\ |B| \text{ even} \\ i \in B}} c_B \cdot e_B + \sum_{\substack{B \subseteq [n] \\ |B| \text{ even} \\ i \notin B}} c_B \cdot e_B - \sum_{\substack{B \subseteq [n] \\ |B| \text{ odd} \\ i \in B}} c_B \cdot e_B - \sum_{\substack{B \subseteq [n] \\ |B| \text{ odd} \\ i \notin B}} c_B \cdot e_B \right) e_i \tag{334}$$

$$= e_i \left( - \sum_{\substack{B \subseteq [n] \\ |B| \text{ even} \\ i \in B}} c_B \cdot e_B + \sum_{\substack{B \subseteq [n] \\ |B| \text{ even} \\ i \notin B}} c_B \cdot e_B - \sum_{\substack{B \subseteq [n] \\ |B| \text{ odd} \\ i \in B}} c_B \cdot e_B + \sum_{\substack{B \subseteq [n] \\ |B| \text{ odd} \\ i \notin B}} c_B \cdot e_B \right). \tag{335}$$

Since $e_i$ with $\mathfrak{q}(e_i) \neq 0$ is invertible, we can cancel $e_i$ on both sides and make use of the linear independence of $(e_B)_{B \subseteq [n]}$ to get that:

$$c_B = 0 \qquad \text{if} \qquad i \in B. \tag{336}$$

Since for given $B$ this can be concluded from every $e_i$ with $\mathfrak{q}(e_i) \neq 0$ we can only have $c_B \neq 0$ if $\mathfrak{q}(e_j) = 0$ for all $j \in B$. Note that elements $e_j$ with $\mathfrak{q}(e_j) = 0$ that are part of an orthogonal basis satisfy $e_j \in \mathcal{R}$. This shows that:

$$y = \sum_{\substack{B \subseteq [n] \\ \forall j \in B. \ \mathfrak{q}(e_j) = 0}} c_B \cdot e_B \in \mathrm{span}\{e_A \mid A \subseteq [n], \forall i \in A. \ e_i \in \mathcal{R}\} = \bigwedge(\mathcal{R}). \tag{337}$$

This shows the remaining inclusion:

$$\{y \in \mathrm{Cl}(V, \mathfrak{q}) \mid \forall v \in V. \ \alpha(y)v = vy\} \subseteq \bigwedge(\mathcal{R}). \tag{338}$$

This shows the equality of all three sets. $\qquad\square$

## E    The Clifford Group and its Clifford Algebra Representations

We saw that Cartan-Dieudonné (Theorem C.13) generates the orthogonal group of a (non-degenerate) quadratic space by composing reflections. Considering this, we seek in the following a group representation that acts on the entire Clifford algebra, but reduces to a reflection when restricted to $V$. Further, we ensure that the action is an algebra homomorphism and will therefore respect the geometric product.

## E.1 Adjusting the Twisted Conjugation

Recall the notation $\mathrm{Cl}^\times(V, \mathfrak{q}) := \{x \in \mathrm{Cl}(V, \mathfrak{q}) \mid \exists y \in \mathrm{Cl}(V, \mathfrak{q}). \, xy = yx = 1\}$.

**Motivation E.1** (Generalizing reflection operations). *For $v, w \in V$ with $\mathfrak{q}(w) \neq 0$ the reflection of $v$ onto the hyperplane that is normal to $w$ is given by the following formula, which we then simplify:*

$$r_w(v) = v - 2\frac{\mathfrak{b}(w, v)}{\mathfrak{b}(w, w)}w \tag{339}$$

$$= wwv/\mathfrak{q}(w) - 2\frac{\mathfrak{b}(w, v)}{\mathfrak{q}(w)}w \tag{340}$$

$$= -w(-wv + 2\mathfrak{b}(w, v))/\mathfrak{q}(w) \tag{341}$$

$$= -wvw/\mathfrak{q}(w) \tag{342}$$

$$= -wvw^{-1}. \tag{343}$$

*So we have $r_w(v) = -wvw^{-1}$ for $v, w \in V$ with $\mathfrak{q}(w) \neq 0$. We would like to generalize this to an operation $\rho$ for $w$ from a subgroup of $\Gamma \subseteq \mathrm{Cl}^\times(V, \mathfrak{q})$ (as large as possible) onto all elements $x \in \mathrm{Cl}(V, \mathfrak{q})$. More explicitely, we want for all $w_1, w_2 \in \Gamma$ and $x_1, x_2 \in \mathrm{Cl}(V, \mathfrak{q})$, $v, w \in V$, $\mathfrak{q}(w) \neq 0$:*

$$\rho(w)(v) = -wvw^{-1} = r_w(v), \tag{344}$$

$$(\rho(w_2) \circ \rho(w_1))(x_1) = \rho(w_2 w_1)(x_1), \tag{345}$$

$$\rho(w_1)(x_1 + x_2) = \rho(w_1)(x_1) + \rho(w_1)(x_2), \tag{346}$$

$$\rho(w_1)(x_1 x_2) = \rho(w_1)(x_1)\rho(w_1)(x_2). \tag{347}$$

*The second condition makes sure that $\mathrm{Cl}(V, \mathfrak{q})$ will be a group representation of $\Gamma$, i.e. we get a group homomorphism:*

$$\rho : \Gamma \to \mathrm{Aut}(\mathrm{Cl}(V, \mathfrak{q})), \tag{348}$$

*where $\mathrm{Aut}(\mathrm{Cl}(V, \mathfrak{q}))$ denotes the set of automorphisms $\mathrm{Cl}(V, \mathfrak{q}) \to \mathrm{Cl}(V, \mathfrak{q})$.*

*In the literature, the following versions of reflection operations were studied:*

$$\rho_0(w) : x \mapsto wxw^{-1}, \qquad \rho_1(w) : x \mapsto \alpha(w)xw^{-1}, \qquad \rho_2(w) : x \mapsto wx\alpha(w)^{-1}. \tag{349}$$

*Since $\rho_0(w)$ is missing the minus sign it only generalizes compositions of reflections for elements $w = w_1 \cdots w_k$, $w_l \in V$, $\mathfrak{q}(v_l) \neq 0$, of even parity $k$. The map $\rho_1(w)$, on the other hand, takes the minus sign into account and generalizes also to such elements $w = w_1 \cdots w_k$ of odd party $k = \mathrm{prt}(w)$ as then: $\alpha(w) = (-1)^{\mathrm{prt}(w)}w$. However, in contrast to $\rho_0(w)$, which is an algebra homomorphism for all $w \in \mathrm{Cl}^\times(V, \mathfrak{q})$, the map $\rho_1(w)$ is not multiplicative in $x$, as can be seen with $v_1, v_2 \in V$ and $w \in V$ with $\mathfrak{q}(w) \neq 0$:*

$$\rho_1(w)(v_1 v_2) = \alpha(w)v_1 v_2 w^{-1} \tag{350}$$

$$= (-1)^{\mathrm{prt}(w)}wv_1 w^{-1}(-1)^{\mathrm{prt}(w)}(-1)^{\mathrm{prt}(w)}wv_2 w^{-1} \tag{351}$$

$$= (-1)^{\mathrm{prt}(w)}\rho_1(w)(v_1)\rho_1(w)(v_2) \tag{352}$$

$$\neq \rho_1(w)(v_1)\rho_1(w)(v_2). \tag{353}$$

*The lack of multiplicativity means that reflection and taking geometric product does not commute.*

*To fix this, it makes sense to first restrict $\rho_1(w)$ to $V$, where it coincides with $r_w$ and also still is multiplicative in $w$, and then study under which conditions on $w$ it extends to an algebra homomorphism $\mathrm{Cl}(V, \mathfrak{q}) \to \mathrm{Cl}(V, \mathfrak{q})$.*

*More formally, by the universal property of the Clifford algebra, the obstruction for:*

$$\rho(w) : V \to \mathrm{Cl}(V, \mathfrak{q}), \quad \rho(w)(v) := \alpha(w)vw^{-1} = w\eta(w)vw^{-1}, \quad \eta(w) := w^{-1}\alpha(w), \tag{354}$$

*with general invertible $w \in \mathrm{Cl}^\times(V, \mathfrak{q})$, to extend to an $\mathbb{F}$-algebra homomorphism:*

$$\rho(w) : \mathrm{Cl}(V, \mathfrak{q}) \to \mathrm{Cl}(V, \mathfrak{q}), \tag{355}$$

*is the following:*

$$\forall v \in V. \qquad \mathfrak{q}(v) \overset{!}{=} (\rho(w)(v))^2 \tag{356}$$

$$= \rho(w)(v)\rho(w)(v) \tag{357}$$

$$= (w\eta(w)vw^{-1})(w\eta(w)vw^{-1}) \tag{358}$$

$$= w\eta(w)v\eta(w)vw^{-1}, \tag{359}$$

*which reduces to:*

$$\forall v \in V. \qquad \mathfrak{q}(v) \overset{!}{=} \eta(w)v\eta(w)v. \tag{360}$$

*The latter is, for instance, satisfied if $\eta(w)$ commutes with every $v \in V$ and $1 = \eta(w)^2$. In particular, the above requirement is satisfied for all $w \in \mathrm{Cl}^{\times}(V, \mathfrak{q})$ with $\eta(w) \in \{\pm 1\}$, which is equivalent to $\alpha(w) = \pm w$, which means that $w$ is a homogeneous element of $\mathrm{Cl}(V, \mathfrak{q})$ in the parity grading. This discussion motivates the following definitions and analysis.*

**Notation E.2** (The coboundary of $\alpha$)**.** *The* coboundary $\eta$ *of $\alpha$ is defined on $w \in \mathrm{Cl}^{\times}(V, \mathfrak{q})$ as follows:*

$$\eta : \mathrm{Cl}^{\times}(V, \mathfrak{q}) \to \mathrm{Cl}^{\times}(V, \mathfrak{q}), \qquad\qquad \eta(w) := w^{-1}\alpha(w). \tag{361}$$

**Remark E.3.** *1. $\eta$ is a* crossed group homomorphism *(aka 1-cocycle), i.e. for $w_1, w_2 \in \mathrm{Cl}^{\times}(V, \mathfrak{q})$ we have:*

$$\eta(w_1 w_2) = \eta(w_1)^{w_2}\eta(w_2), \qquad with \qquad \eta(w_1)^{w_2} := w_2^{-1}\eta(w_1)w_2. \tag{362}$$

*2. For $w \in \mathrm{Cl}^{\times}(V, \mathfrak{q})$ we have:*

$$\alpha(\eta(w)) = \alpha(w)^{-1}w = \eta(w)^{-1}. \tag{363}$$

*3. For $w \in \mathrm{Cl}^{\times}(V, \mathfrak{q})$ we have that $w$ is an homogeneous element in $\mathrm{Cl}(V, \mathfrak{q})$, in the sense of parity, if and only if $\eta(w) \in \{\pm 1\}$.*

**Definition E.4** (The group of homogeneous invertible elements)**.** *With the introduced notation we can define the group of all invertible elements of $\mathrm{Cl}(V, \mathfrak{q})$ that are also homogeneous (in the sense of parity) as:*

$$\mathrm{Cl}^{[\times]}(V, \mathfrak{q}) := \left(\mathrm{Cl}^{\times}(V, \mathfrak{q}) \cap \mathrm{Cl}^{[0]}(V, \mathfrak{q})\right) \cup \left(\mathrm{Cl}^{\times}(V, \mathfrak{q}) \cap \mathrm{Cl}^{[1]}(V, \mathfrak{q})\right) \tag{364}$$

$$= \left\{ w \in \mathrm{Cl}^{\times}(V, \mathfrak{q}) \,\middle|\, \eta(w) \in \{\pm 1\} \right\}. \tag{365}$$

**Notation E.5** (The main involution - revisited)**.** *We now make the following abbreviations:*

$$\alpha^0 := \mathrm{id} : \mathrm{Cl}(V, \mathfrak{q}) \to \mathrm{Cl}(V, \mathfrak{q}), \qquad \alpha^0(x) := x^{[0]} + x^{[1]} = x, \tag{366}$$

$$\alpha^1 := \alpha : \mathrm{Cl}(V, \mathfrak{q}) \to \mathrm{Cl}(V, \mathfrak{q}), \qquad \alpha^1(x) := x^{[0]} - x^{[1]}. \tag{367}$$

*For $w \in \mathrm{Cl}(V, \mathfrak{q})$ we then have:*

$$\alpha^{\mathrm{prt}(w)} : \mathrm{Cl}(V, \mathfrak{q}) \to \mathrm{Cl}(V, \mathfrak{q}), \qquad \alpha^{\mathrm{prt}(w)}(x) = x^{[0]} + (-1)^{\mathrm{prt}(w)}x^{[1]}. \tag{368}$$

*Note that $\alpha^{\mathrm{prt}(w)}$ is an $\mathbb{F}$-algebra involution of $\mathrm{Cl}(V, \mathfrak{q})$ that preserves the parity grading of $\mathrm{Cl}(V, \mathfrak{q})$.*

*We also need the following slight variation $\alpha^w$, which in many, but not all cases, coincides with $\alpha^{\mathrm{prt}(w)}$. For $w \in \mathrm{Cl}^{\times}(V, \mathfrak{q})$ we define the $w$-twisted map:*

$$\alpha^w : \mathrm{Cl}(V, \mathfrak{q}) \to \mathrm{Cl}(V, \mathfrak{q}), \qquad \alpha^w(x) := x^{[0]} + \eta(w)x^{[1]}. \tag{369}$$

**Remark E.6.** *Note that for $w \in \mathrm{Cl}^{[\times]}(V, \mathfrak{q})$ we have that $\eta(w) = (-1)^{\mathrm{prt}(w)}$ and thus $\alpha^w = \alpha^{\mathrm{prt}(w)}$, in which case $\alpha^w$ is an $\mathbb{F}$-algebra involution of $\mathrm{Cl}(V, \mathfrak{q})$ that preserves the parity grading of $\mathrm{Cl}(V, \mathfrak{q})$.*

**Definition E.7** (Adjusted twisted conjugation)**.** *For $w \in \mathrm{Cl}^{\times}(V, \mathfrak{q})$ we define the* twisted conjugation*:*

$$\rho(w) : \mathrm{Cl}(V, \mathfrak{q}) \to \mathrm{Cl}(V, \mathfrak{q}), \qquad \rho(w)(x) := wx^{[0]}w^{-1} + \alpha(w)x^{[1]}w^{-1}. \tag{370}$$

$$= w\left(x^{[0]} + \eta(w)x^{[1]}\right)w^{-1} \tag{371}$$

$$= w\alpha^w(x)w^{-1}. \tag{372}$$

We now want to re-investigate the action of $\rho$ on $\mathrm{Cl}(V, \mathfrak{q})$.

**Lemma E.8.** *For every $w \in \mathrm{Cl}^{[\times]}(V, \mathfrak{q})$ the map:*

$$\rho(w) : \mathrm{Cl}(V, \mathfrak{q}) \to \mathrm{Cl}(V, \mathfrak{q}), \qquad x \mapsto \rho(w)(x) = w\left(x^{[0]} + \eta(w)x^{[1]}\right)w^{-1}, \qquad (373)$$

*is an $\mathbb{F}$-algebra automorphism that preserves the parity grading of $\mathrm{Cl}(V, \mathfrak{q})$. Its inverse it given by $\rho(w^{-1})$.*

*Proof.* First note that $\alpha^w$ agrees with the involution $\alpha^{\mathrm{prt}(w)}$ for $w \in \mathrm{Cl}^{[\times]}(V, \mathfrak{q})$. Since $\alpha^{\mathrm{prt}(w)}$ is an $\mathbb{F}$-algebra automorphism that preserves the parity grading of $\mathrm{Cl}(V, \mathfrak{q})$ so is $\alpha^w$ for $w \in \mathrm{Cl}^{[\times]}(V, \mathfrak{q})$.

Furthermore, the conjugation $\rho_0(w) : x \mapsto wxw^{-1}$ is an $\mathbb{F}$-algebra automorphism, which preserves the parity grading of $\mathrm{Cl}(V, \mathfrak{q})$ if $w \in \mathrm{Cl}^{[\times]}(V, \mathfrak{q})$. To see this, note that $0 = \mathrm{prt}(1) = \mathrm{prt}(ww^{-1}) = \mathrm{prt}(w) + \mathrm{prt}(w^{-1})$. As such, $\mathrm{prt}(w) = \mathrm{prt}(w^{-1})$. Then, $\mathrm{prt}(wxw^{-1}) = \mathrm{prt}(w) + \mathrm{prt}(x) + \mathrm{prt}(w^{-1}) = 2\mathrm{prt}(w) + \mathrm{prt}(x) = \mathrm{prt}(x)$. Here, we use the fact that the Clifford algebra is $\mathbb{Z}/2\mathbb{Z}$-graded.

So, their composition $\rho(w) = \rho_0(w) \circ \alpha^w$ is also an $\mathbb{F}$-algebra automorphism that preserves the parity grading $\mathrm{Cl}(V, \mathfrak{q})$ for $w \in \mathrm{Cl}^{[\times]}(V, \mathfrak{q})$. $\qquad \square$

As a direct corollary we get:

**Corollary E.9.** *Let $F(T_{0,1}, \dots, T_{1,\ell}) \in \mathbb{F}[T_{0,1}, \dots, T_{1,\ell}]$ be a polynomial in $2\ell$ variables with coefficients in $\mathbb{F}$. Let $x_1, \dots, x_\ell \in \mathrm{Cl}(V, \mathfrak{q})$ be $\ell$ elements of the Clifford algebra and $w \in \mathrm{Cl}^{[\times]}(V, \mathfrak{q})$ be a homogeneous invertible element of $\mathrm{Cl}(V, \mathfrak{q})$. Then we have the following equivariance property:*

$$\rho(w)\left(F(x_1^{[0]}, \dots, x_l^{[i]}, \dots, x_\ell^{[1]})\right) = F(\rho(w)(x_1)^{[0]}, \dots, \rho(w)(x_l)^{[i]}, \dots, \rho(w)(x_\ell)^{[1]}). \quad (374)$$

*Proof.* This directly follows from Lemma E.8. $\qquad \square$

Futhermore, we get the following result:

**Theorem E.10.** *The map:*

$$\rho : \mathrm{Cl}^{[\times]}(V, \mathfrak{q}) \to \mathrm{Aut}_{\mathbf{Alg},\mathrm{prt}}\left(\mathrm{Cl}(V, \mathfrak{q})\right), \qquad\qquad w \mapsto \rho(w), \qquad (375)$$

*is a well-defined group homomorphism from the group of all homogeneous invertible elements of $\mathrm{Cl}(V, \mathfrak{q})$ to the group of $\mathbb{F}$-algebra automorphisms of $\mathrm{Cl}(V, \mathfrak{q})$ that preserve the parity grading of $\mathrm{Cl}(V, \mathfrak{q})$. In particular, $\mathrm{Cl}(V, \mathfrak{q})$, $\mathrm{Cl}^{[0]}(V, \mathfrak{q})$, $\mathrm{Cl}^{[1]}(V, \mathfrak{q})$ are group representations of $\mathrm{Cl}^{[\times]}(V, \mathfrak{q})$ via $\rho$.*

*Proof.* By the previous Lemma E.8 we already know that $\rho$ is a well-defined map. We only need to check if it is a group homomorphism. Let $w_1, w_2 \in \mathrm{Cl}^{[\times]}(V, \mathfrak{q})$ and $x \in \mathrm{Cl}(V, \mathfrak{q})$, then we get:

$$(\rho(w_2) \circ \rho(w_1))(x) = \rho(w_2)(\rho(w_1)(x)) \qquad\qquad\qquad\qquad (376)$$

$$= \rho(w_2)\left(w_1 \alpha^{\mathrm{prt}(w_1)}(x)w_1^{-1}\right) \qquad\qquad\qquad (377)$$

$$= w_2 \alpha^{\mathrm{prt}(w_2)}\left(w_1 \alpha^{\mathrm{prt}(w_1)}(x)w_1^{-1}\right)w_2^{-1} \qquad\qquad (378)$$

$$= w_2 \alpha^{\mathrm{prt}(w_2)}(w_1)\alpha^{\mathrm{prt}(w_2)}(\alpha^{\mathrm{prt}(w_1)}(x))\alpha^{\mathrm{prt}(w_2)}(w_1)^{-1}w_2^{-1} \qquad (379)$$

$$= w_2(-1)^{\mathrm{prt}(w_2)\,\mathrm{prt}(w_1)}w_1\alpha^{\mathrm{prt}(w_2)+\mathrm{prt}(w_1)}(x)(-1)^{\mathrm{prt}(w_2)\,\mathrm{prt}(w_1)}w_1^{-1}w_2^{-1}$$
$$(380)$$

$$= w_2 w_1 \alpha^{\mathrm{prt}(w_2)+\mathrm{prt}(w_1)}(x)w_1^{-1}w_2^{-1} \qquad\qquad\qquad (381)$$

$$= (w_2 w_1)\alpha^{\mathrm{prt}(w_2 w_1)}(x)(w_2 w_1)^{-1} \qquad\qquad\qquad (382)$$

$$= \rho(w_2 w_1)(x), \qquad\qquad\qquad\qquad\qquad (383)$$

where we used the multiplicativity of $\alpha^{\mathrm{prt}(w)}(x)$:

$$\alpha^w(x)\alpha^w(y) = \left((-1)^{\mathrm{prt}(w)}x^{[1]} + x^{[0]}\right)\left((-1)^{\mathrm{prt}(w)}y^{[1]} + y^{[0]}\right) \tag{384}$$

$$= (-1)^{\mathrm{prt}(w)}\left(x^{[0]}y^{[1]} + x^{[1]}y^{[0]}\right) + x^{[1]}y^{[1]} + x^{[0]}y^{[0]} \tag{385}$$

$$= \alpha^w\left((xy)^{[1]}\right) + (xy)^{[0]} \tag{386}$$

$$= \alpha^w(xy). \tag{387}$$

This implies:

$$\rho(w_2) \circ \rho(w_1) = \rho(w_2 w_1), \tag{388}$$

which shows the claim. $\qquad\square$

Finally, we want to re-check that our newly defined $\rho$, despite its different appearance, still has the proper interpretation of a reflection.

**Remark E.11.** *Let $w, v \in V$ with $\mathfrak{q}(w) \neq 0$. Then $\rho(w)(v)$ is the reflection of $v$ w.r.t. the hyperplane that is normal to $w$:*

$$\rho(w)(v) = w\alpha^w(v)w^{-1} = -wvw^{-1} = r_w(v). \tag{389}$$

**Remark E.12.** *The presented results in this subsection can be slightly generalized as follows. If $w \in \mathrm{Cl}^\times(V, \mathfrak{q})$ such that $\eta(w) \in \bigwedge(\mathcal{R})$ then by Theorem D.50 we get for all $v \in V$:*

$$v\eta(w) = \alpha(\eta(w))v = \eta(w)^{-1}v. \tag{390}$$

*This implies that for all $v \in V$:*

$$\eta(w)v\eta(w)v = \eta(w)\eta(w)^{-1}vv = \mathfrak{q}(v), \tag{391}$$

*and thus for all $v \in V$:*

$$(\alpha(w)vw^{-1})(\alpha(w)vw^{-1}) = \mathfrak{q}(v). \tag{392}$$

*By the universal property of the Clifford algebra the map:*

$$\rho(w): \ V \to \mathrm{Cl}(V, \mathfrak{q}), \qquad\qquad v \mapsto \alpha(w)vw^{-1}, \tag{393}$$

*uniquely extends to an $\mathbb{F}$-algebra homomorphism:*

$$\rho(w): \ \mathrm{Cl}(V, \mathfrak{q}) \to \mathrm{Cl}(V, \mathfrak{q}), \tag{394}$$

*with:*

$$x = c_0 + \sum_{i \in I} c_i \cdot v_{i,1} \cdots v_{i,k_i} \tag{395}$$

$$\mapsto c_0 + \sum_{i \in I} c_i \cdot \alpha(w)v_{i,1}w^{-1} \cdots \alpha(w)v_{i,k_i}w^{-1} \tag{396}$$

$$= w\left(c_0 + \sum_{i \in I} c_i \cdot \eta(w)v_{i,1} \cdots \eta(w)v_{i,k_i}\right)w^{-1} \tag{397}$$

$$= w\left(c_0 + \sum_{\substack{i \in I \\ k_i \text{ even}}} c_i \cdot v_{i,1} \cdots v_{i,k_i} + \eta(w) \cdot \sum_{\substack{i \in I \\ k_i \text{ odd}}} c_i \cdot v_{i,1} \cdots v_{i,k_i}\right)w^{-1} \tag{398}$$

$$= w\left(x^{[0]} + \eta(w) \cdot x^{[1]}\right)w^{-1}. \tag{399}$$

*To further ensure that the elements $w \in \mathrm{Cl}^\times(V, \mathfrak{q})$ with the above property form a group we might need to further restrict to require that $\eta(w) \in \bigwedge^{[\times]}(\mathcal{R})$. At least in this case we get for $w_1, w_2 \in \mathrm{Cl}^\times(V, \mathfrak{q})$ with $\eta(w_1), \eta(w_2) \in \bigwedge^{[\times]}(\mathcal{R}) \subseteq \mathfrak{Z}(\mathrm{Cl}(V, \mathfrak{q}))$, see Remark D.43, that:*

$$\eta(w_2 w_1) = w_1^{-1}\eta(w_2)w_1\eta(w_1) = w_1^{-1}w_1\eta(w_2)\eta(w_1) = \eta(w_2)\eta(w_1) \in \bigwedge^{[\times]}(\mathcal{R}). \tag{400}$$

*So the following set:*

$$C := \left\{w \in \mathrm{Cl}^\times(V, \mathfrak{q}) \ \middle|\ \eta(w) \in \bigwedge^{[\times]}(\mathcal{R})\right\} \tag{401}$$

*is a subgroup of $\mathrm{Cl}^\times(V, \mathfrak{q})$ where every $w \in C$ defines an algebra homomorphism:*

$$\rho(w): \ \mathrm{Cl}(V, \mathfrak{q}) \to \mathrm{Cl}(V, \mathfrak{q}), \qquad\qquad \rho(w)(x) = w\left(x^{[0]} + \eta(w)x^{[1]}\right)w^{-1}. \tag{402}$$

## E.2 The Clifford Group

**Motivation E.13.** *We have seen in the last section that if we choose homogeneous invertible elements* $w \in \mathrm{Cl}^{[\times]}(V, \mathfrak{q})$ *then the action* $\rho(w)$*, the (adjusted) twisted conjugation, is an algebra automorphism of* $\mathrm{Cl}(V, \mathfrak{q})$ *that also preserves the parity grading of* $\mathrm{Cl}(V, \mathfrak{q})$*, in particular,* $\rho(w)$ *is linear and multiplicative.*

*We now want to investigate under which conditions on* $w$ *the algebra automorphism* $\rho(w)$ *also preserves the multivector grading:*

$$\mathrm{Cl}(V, \mathfrak{q}) = \mathrm{Cl}^{(0)}(V, \mathfrak{q}) \oplus \mathrm{Cl}^{(1)}(V, \mathfrak{q}) \oplus \cdots \oplus \mathrm{Cl}^{(m)}(V, \mathfrak{q}) \oplus \cdots \oplus \mathrm{Cl}^{(n)}(V, \mathfrak{q}). \tag{403}$$

*If this was the case then each component* $\mathrm{Cl}^{(m)}(V, \mathfrak{q})$ *would give rise to a corresponding group representation.*

*To preserve the multivector grading we at least need that for* $v \in V = \mathrm{Cl}^{(1)}(V, \mathfrak{q})$ *we have that also* $\rho(w)(v) \in \mathrm{Cl}^{(1)}(V, \mathfrak{q}) = V$*. We will see that for* $w \in \mathrm{Cl}^{\times}(V, \mathfrak{q})$ *such that* $\rho(w)$ *is an algebra homomorphism, i.e. for* $w$ *homogeneous and invertible, and such that* $\rho(w)(v) \in V$ *for all* $v \in V$*, we already get the preservation of the whole multivector grading of* $\mathrm{Cl}(V, \mathfrak{q})$*.*

Again, let $(V, \mathfrak{q})$ be a quadratic space over a field $\mathbb{F}$ with $\mathrm{char}(\mathbb{F}) \neq 2$ and $\dim V = n < \infty$ and $e_1, \ldots, e_n$ and orthogonal basis of $(V, \mathfrak{q})$.

**Remark E.14.** *In the following, we elaborate on the term* Clifford group *in constrast to previous literature. First, the* unconstrained *Clifford group is also referred to as the* Lipschitz group *or* Clifford-Lipschitz *group in honor of its creator Rudolf Lipschitz [LS09]. Throughout its inventions, several generalizations and versions have been proposed, often varying in details, leading to slightly non-equivalent definitions. For instance, some authors utilize conjugation as an action, others apply the the twisted conjugation, while yet another group employs the twisting on the other side of the conjugation operation. Furthermore, some authors require that the elements are homogeneous in the parity grading, where others do not. We settle for a definition that requires homogeneous invertible elements of the Clifford algebra that act via our* adjusted *twisted conjugation such that elements from the vector space* $V$ *land also in* $V$*. The reason for our definition is that we want the adjusted twisted conjugation to act on the whole Clifford algebra* $\mathrm{Cl}(V, \mathfrak{q})$*, not just on the vector space* $V$*. Furthermore, we want it to respect, besides the vector space structure of* $\mathrm{Cl}(V, \mathfrak{q})$*, also the product structure, leading to algebra homomorphisms. In addition, we also want that the action to respect the extended bilinear form* $\mathfrak{b}$*, the orthogonal structure, and the multivector grading. These properties might not (all) be ensured in other definitions with different nuances.*

*Also note that the name Clifford group might be confused with different groups with the same name in other literature, e.g., with the group of unitary matrices that normalize the Pauli group or with the finite group inside the Clifford algebra that is generated by an orthogonal basis via the geometric product.*

**Definition E.15.**     *1. We denote the* unconstrained Clifford group *of* $\mathrm{Cl}(V, \mathfrak{q})$ *as follows:*

$$\tilde{\Gamma}(V, \mathfrak{q}) := \left\{ w \in \mathrm{Cl}^{\times}(V, \mathfrak{q}) \,\middle|\, \forall v \in V.\, \rho(w)(v) \in V \right\}. \tag{404}$$

*2. We denote the* Clifford group *of* $\mathrm{Cl}(V, \mathfrak{q})$ *as follows:*

$$\Gamma(V, \mathfrak{q}) := \mathrm{Cl}^{[\times]}(V, \mathfrak{q}) \cap \tilde{\Gamma}(V, \mathfrak{q}) \tag{405}$$

$$= \left\{ w \in \mathrm{Cl}^{\times}(V, \mathfrak{q}) \,\middle|\, \eta(w) \in \{\pm 1\} \wedge \forall v \in V.\, \rho(w)(v) \in V \right\}. \tag{406}$$

*3. We define the* special Clifford group *as follows:*

$$\Gamma^{[0]}(V, \mathfrak{q}) := \tilde{\Gamma}(V, \mathfrak{q}) \cap \mathrm{Cl}^{[0]}(V, \mathfrak{q}) = \Gamma(V, \mathfrak{q}) \cap \mathrm{Cl}^{[0]}(V, \mathfrak{q}). \tag{407}$$

**Theorem E.16.** *For* $w \in \Gamma(V, \mathfrak{q})$ *and* $x \in \mathrm{Cl}(V, \mathfrak{q})$ *we have for all* $m = 0, \ldots, n$*:*

$$\rho(w)(x^{(m)}) = \rho(w)(x)^{(m)}. \tag{408}$$

*In particular, for* $x \in \mathrm{Cl}^{(m)}(V, \mathfrak{q})$ *we also have* $\rho(w)(x) \in \mathrm{Cl}^{(m)}(V, \mathfrak{q})$*.*

*Proof.* We first claim that for $w \in \Gamma(V, \mathfrak{q})$ the set of elements:

$$b_1 := \rho(w)(e_1), \ldots, b_n := \rho(w)(e_n), \tag{409}$$

forms an orthogonal basis of $(V, \mathfrak{q})$. Indeed, since $\rho(w)(e_t) \in V$, by definition of $\Gamma(V, \mathfrak{q})$, the orthogonality relation, $i \neq j$:

$$0 = 2\mathfrak{b}(e_i, e_j) = e_i e_j + e_j e_i, \tag{410}$$

transforms under $\rho(w)$ to:

$$0 = \rho(w)(0) = \rho(w)\left(e_i e_j + e_j e_i\right) \tag{411}$$
$$= \rho(w)(e_i)\rho(w)(e_j) + \rho(w)(e_j)\rho(w)(e_i) \tag{412}$$
$$= b_i b_j + b_j b_i \tag{413}$$
$$= 2\mathfrak{b}(b_i, b_j). \tag{414}$$

This shows that $b_1, \ldots, b_n$ is an orthogonal system in $V$. Using $\rho(w^{-1})$ we also see that the system is linear independent and thus an orthognal basis of $V$. By the basis-independence of the multivector grading $\mathrm{Cl}^{(m)}(V, \mathfrak{q})$, see Theorem D.27, we then get for:

$$x = \sum_{i_1 < \cdots < i_m} c_{i_1, \ldots, i_m} \cdot e_{i_1} \cdots e_{i_m} \in \mathrm{Cl}^{(m)}(V, \mathfrak{q}), \tag{415}$$

the relation:

$$\rho(w)(x) = \sum_{i_1 < \cdots < i_m} c_{i_1, \ldots, i_m} \cdot b_{i_1} \cdots b_{i_m} \in \mathrm{Cl}^{(m)}(V, \mathfrak{q}). \tag{416}$$

This shows the claim. $\qquad\square$

**Corollary E.17.** *The map:*

$$\rho : \Gamma(V, \mathfrak{q}) \to \mathrm{Aut}_{\mathbf{Alg},\mathrm{grd}}\left(\mathrm{Cl}(V, \mathfrak{q})\right), \qquad\qquad w \mapsto \rho(w), \tag{417}$$

*is a well-defined group homomorphism from the Clifford group to the group of $\mathbb{F}$-algebra automorphisms of $\mathrm{Cl}(V, \mathfrak{q})$ that preserve the multivector grading of $\mathrm{Cl}(V, \mathfrak{q})$. In particular, $\mathrm{Cl}(V, \mathfrak{q})$ and $\mathrm{Cl}^{(m)}(V, \mathfrak{q})$ for $m = 0, \ldots, n$, are group representations of $\Gamma(V, \mathfrak{q})$ via $\rho$.*

**Corollary E.18.** *Let $F(T_1, \ldots, T_\ell) \in \mathbb{F}[T_1, \ldots, T_\ell]$ be a polynomial in $\ell$ variables with coefficients in $\mathbb{F}$ and let $k \in \{0, \ldots, n\}$. Further, consider $\ell$ elements $x_1, \ldots, x_\ell \in \mathrm{Cl}(V, \mathfrak{q})$. Then for every $w \in \Gamma(V, \mathfrak{q})$ we get the equivariance property:*

$$\rho(w)\left(F(x_1, , \ldots, x_\ell)^{(k)}\right) = F(\rho(w)(x_1), \ldots, \rho(w)(x_\ell))^{(k)}, \tag{418}$$

*where the superscript $(k)$ indicates the projection onto the multivector grade-$k$-part of the whole expression.*

**Example E.19.** *Let $w \in V$ with $\mathfrak{q}(w) \neq 0$, then $w \in \Gamma(V, \mathfrak{q})$.*

*Proof.* It is clear that $w$ is homogeneous in the parity grading, as $\eta(w) = -1$. For $v \in V$ we get:

$$\rho(w)(v) = r_w(v) = v - 2\frac{\mathfrak{b}(v, w)}{\mathfrak{q}(w)}w \in V. \tag{419}$$

This shows $w \in \Gamma(V, \mathfrak{q})$. $\qquad\square$

**Example E.20.** *Let $e, f \in V$ with $\mathfrak{b}(v, f) = 0$ for all $v \in V$, and put: $\gamma := 1 + ef \in \mathrm{Cl}^{[0]}(V, \mathfrak{q})$. Then $\gamma \in \Gamma(V, \mathfrak{q})$.*

*Proof.* First, note that for all $v \in V$ we get:

$$fv = -vf + 2\mathfrak{b}(f, v) = -vf. \tag{420}$$

Next, we see that:

$$\gamma^{-1} = 1 - ef. \tag{421}$$

Indeed, we get:

$$(1 + ef)(1 - ef) = 1 + ef - ef - efef \tag{422}$$

$$= 1 + effe \tag{423}$$

$$= 1 + \underbrace{\mathfrak{q}(f)}_{=0}\mathfrak{q}(e) \tag{424}$$

$$= 1. \tag{425}$$

Now consider $v \in V$. We then have:

$$\rho(\gamma)(v) = \alpha(\gamma)v\gamma^{-1} \tag{426}$$

$$= (1 + ef)v(1 - ef) \tag{427}$$

$$= (1 + ef)(v - vef) \tag{428}$$

$$= v + efv - vef - efvef \tag{429}$$

$$= v - evf - vef - eveff \tag{430}$$

$$= v - (ev + ve)f - \underbrace{\mathfrak{q}(f)}_{=0}eve \tag{431}$$

$$= v - 2\mathfrak{b}(e,v)f \tag{432}$$

$$\in V. \tag{433}$$

This shows $\gamma \in \Gamma(V, \mathfrak{q})$. $\qquad\square$

**Example E.21.** *Let $f_1, \ldots, f_r \in V$ be a basis of the radical subspace $\mathcal{R}$ of $(V, \mathfrak{q})$. In particular, we have $\mathfrak{b}(v, f_j) = 0$ for all $v \in V$. Then we put:*

$$g := 1 + h, \quad \text{with} \quad h \in \operatorname{span}\{f_{k_1}\cdots f_{k_s} \mid s \geq 2 \ \underline{even}, \ 1 \leq k_1 < k_2 < \ldots k_s \leq r\} \subseteq \operatorname{Cl}(V, \mathfrak{q}). \tag{434}$$

*Then we claim that $g \in \Gamma(V, \mathfrak{q})$ and $\rho(g) = \operatorname{id}_{\operatorname{Cl}(V,\mathfrak{q})}$.*

*Proof.* Since we restrict to even products it is clear that $g \in \operatorname{Cl}^{[0]}(V, \mathfrak{q})$. Furthermore, note that, since $h$ lives in the radical subalgebra, there exists a number $k \geq 1$ such that:

$$h^{k+1} = 0. \tag{435}$$

Then we get that:

$$g^{-1} = 1 - h + h^2 + \cdots + (-1)^k h^k. \tag{436}$$

Indeed, we get:

$$(1 + h)\left(\sum_{l=0}^{k}(-1)^l h^l\right) = \sum_{l=0}^{k}(-1)^l h^l + \sum_{l=0}^{k}(-1)^l h^{l+1} \tag{437}$$

$$= 1 + \sum_{l=1}^{k}(-1)^l h^l - \sum_{l=1}^{k}(-1)^l h^l + (-1)^k\underbrace{h^{k+1}}_{=0} \tag{438}$$

$$= 1. \tag{439}$$

Furthermore, $g$ lies in the center $\mathfrak{Z}(\operatorname{Cl}(V, \mathfrak{q}))$ of $\operatorname{Cl}(V, \mathfrak{q})$ by Theorem D.47. Then for $v \in V$ we get:

$$\rho(g)(v) = \alpha(g)vg^{-1} \tag{440}$$

$$= gvg^{-1} \tag{441}$$

$$= vgg^{-1} \tag{442}$$

$$= v \tag{443}$$

$$\in V. \tag{444}$$

So, $g \in \Gamma(V, \mathfrak{q})$ and acts as the identity on $V$, and thus on $\operatorname{Cl}(V, \mathfrak{q})$, via $\rho$. $\qquad\square$

## E.3  The Structure of the Clifford Group

We have identified the Clifford group and its action on the algebra. In particular, our adjusted twisted conjugation preserves the parity and multivector grading, and reduces to a reflection when restricted to $V$. We now want to further investigate how the Clifford and the Clifford groups act via the twisted conjugation $\rho$ on $V$ and $\mathrm{Cl}(V, \mathfrak{q})$. We again denote by $\mathcal{R} \subseteq V$ the radical subspace of $V$ w.r.t. $\mathfrak{q}$. We follow and extend the analysis of [Cru80, Cru90, DKL10].

We first investigate the kernel of the twisted action.

**Corollary E.22** (The kernel of the twisted conjugation). *1. We have the following identity for the twisted conjugation:*

$$\ker\left(\rho|_{\mathrm{Cl}^{\times}(V, \mathfrak{q})}\right) := \left\{w \in \mathrm{Cl}^{\times}(V, \mathfrak{q}) \,\middle|\, \rho(w) = \mathrm{id}_{\mathrm{Cl}(V, \mathfrak{q})}\right\} \stackrel{!}{=} \bigwedge\nolimits^{\times}(\mathcal{R}). \tag{445}$$

*2. For the twisted conjugation restricted to the unconstrained Clifford, group and to $V$ the kernel is given by:*

$$\ker\left(\rho : \tilde{\Gamma}(V, \mathfrak{q}) \to \mathrm{GL}(V)\right) := \left\{w \in \tilde{\Gamma}(V, \mathfrak{q}) \,\middle|\, \rho(w)|_V = \mathrm{id}_V\right\} \stackrel{!}{=} \bigwedge\nolimits^{\times}(\mathcal{R}). \tag{446}$$

*3. For the twisted conjugation restricted to the Clifford group, the kernel is given by:*

$$\ker\left(\rho : \Gamma(V, \mathfrak{q}) \to \mathrm{Aut}_{\mathbf{Alg}}(\mathrm{Cl}(V, \mathfrak{q}))\right) := \left\{w \in \Gamma(V, \mathfrak{q}) \,\middle|\, \rho(w) = \mathrm{id}_{\mathrm{Cl}(V, \mathfrak{q})}\right\} \tag{447}$$

$$\stackrel{!}{=} \bigwedge\nolimits^{[\times]}(\mathcal{R}). \tag{448}$$

*Proof.* This follows directly from the characterizing of the twisted center of $\mathrm{Cl}(V, \mathfrak{q})$ by Theorem D.50. Note that we have for $w \in \mathrm{Cl}^{\times}(V, \mathfrak{q})$:

$$\rho(w) = \mathrm{id}_{\mathrm{Cl}(V, \mathfrak{q})} \iff \forall x \in \mathrm{Cl}(V, \mathfrak{q}).\ \rho(w)(x) = x \tag{449}$$

$$\iff \forall x \in \mathrm{Cl}(V, \mathfrak{q}).\ wx^{[0]}w^{-1} + \alpha(w)x^{[1]}w^{-1} = x \tag{450}$$

$$\iff \forall x \in \mathrm{Cl}(V, \mathfrak{q}).\ wx^{[0]} + \alpha(w)x^{[1]} = xw \tag{451}$$

$$\iff w \in \mathfrak{K}(\mathrm{Cl}(V, \mathfrak{q})) = \bigwedge(\mathcal{R}). \tag{452}$$

From this follows that:

$$w \in \mathrm{Cl}^{\times}(V, \mathfrak{q}) \cap \bigwedge(\mathcal{R}) = \bigwedge\nolimits^{\times}(\mathcal{R}). \tag{453}$$

The other points follow similarly with Theorem D.50.

For the last point also note:

$$\Gamma(V, \mathfrak{q}) = \tilde{\Gamma}(V, \mathfrak{q}) \cap \mathrm{Cl}^{[\times]}(V, \mathfrak{q}), \qquad \bigwedge\nolimits^{\times}(\mathcal{R}) \cap \mathrm{Cl}^{[\times]}(V, \mathfrak{q}) = \bigwedge\nolimits^{[\times]}(\mathcal{R}). \tag{454}$$

This shows the claims. $\qquad\square$

**Lemma E.23.** *1. For every $w \in \mathrm{Cl}^{\times}(V, \mathfrak{q})$ we have:*

$$\rho(w)|_{\mathcal{R}} = \mathrm{id}_{\mathcal{R}}. \tag{455}$$

*2. For every $w \in \mathrm{Cl}^{[\times]}(V, \mathfrak{q})$ we have:*

$$\rho(w)|_{\bigwedge(\mathcal{R})} = \mathrm{id}_{\bigwedge(\mathcal{R})}. \tag{456}$$

*3. For every $g \in \bigwedge^{\times}(\mathcal{R})$ we have:*

$$\rho(g)|_V = \mathrm{id}_V. \tag{457}$$

*4. For every $g \in \bigwedge^{[\times]}(\mathcal{R})$ we have:*

$$\rho(g)|_{\mathrm{Cl}(V, \mathfrak{q})} = \mathrm{id}_{\mathrm{Cl}(V, \mathfrak{q})}. \tag{458}$$

*Proof.* This directly follows from the twisted commutation relationship, see Lemma D.42. For $w \in \mathrm{Cl}^{\times}(V, \mathfrak{q})$ and $f \in \mathcal{R} \subseteq V$ we have:

$$\rho(w)(f) = \alpha(w)fw^{-1} = fww^{-1} = f. \tag{459}$$

For $w \in \mathrm{Cl}^{[\times]}(V, \mathfrak{q})$ the map $\rho(w)$ is an algebra automorphism of $\mathrm{Cl}(V, \mathfrak{q})$ and satisfies $\rho(w)(f) = f$ for all $f \in \mathcal{R}$. So for every $y \in \bigwedge(\mathcal{R})$ we have:

$$\rho(w)(y) = \rho(w) \left( \sum_{i \in I} c_i \cdot f_{k_i} \cdots f_{l_i} \right) \tag{460}$$

$$= \sum_{i \in I} c_i \cdot \rho(w)(f_{k_i}) \cdots \rho(w)(f_{l_i}) \tag{461}$$

$$= \sum_{i \in I} c_i \cdot f_{k_i} \cdots f_{l_i} \tag{462}$$

$$= y. \tag{463}$$

For $g \in \bigwedge^{\times}(\mathcal{R})$ and $v \in V$ we get:

$$\rho(g)(v) = \alpha(g)vg^{-1} = vgg^{-1} = v. \tag{464}$$

For $g \in \bigwedge^{[\times]}(\mathcal{R})$ the map $\rho(g)$ is an algebra automorphism of $\mathrm{Cl}(V, \mathfrak{q})$ and satisfies $\rho(g)(v) = v$ for all $v \in V$. As above we see that for $x \in \mathrm{Cl}(V, q)$ we get:

$$\rho(g)(x) = \rho(g) \left( \sum_{i \in I} c_i \cdot v_{k_i} \cdots v_{l_i} \right) \tag{465}$$

$$= \sum_{i \in I} c_i \cdot \rho(g)(v_{k_i}) \cdots \rho(g)(v_{l_i}) \tag{466}$$

$$= \sum_{i \in I} c_i \cdot v_{k_i} \cdots v_{l_i} \tag{467}$$

$$= x. \tag{468}$$

This shows all the claims. $\qquad\square$

From Lemma E.23 we see that $\rho(w)|_{\bigwedge(\mathcal{R})} = \mathrm{id}_{\bigwedge(\mathcal{R})}$ for $w \in \mathrm{Cl}^{[\times]}(V, \mathfrak{q})$. Together with Corollary E.18 we arrive at a slightly more general version that allows one to parameterize polynomials not just with coefficients from $\mathbb{F}$, but also with elements from $\bigwedge(\mathcal{R})$, and still get the equivariance w.r.t. the Clifford group $\Gamma(V, \mathfrak{q})$:

**Corollary E.24.** *Let* $F(T_1, \ldots, T_{\ell+s}) \in \mathbb{F}[T_1, \ldots, T_{\ell+s}]$ *be a polynomial in* $\ell + s$ *variables with coefficients in* $\mathbb{F}$ *and let* $k \in \{0, \ldots, n\}$. *Further, consider* $\ell$ *elements* $x_1, \ldots, x_\ell \in \mathrm{Cl}(V, \mathfrak{q})$ *and* $s$ *elements* $y_1, \ldots, y_s \in \bigwedge(\mathcal{R})$. *Then for every* $w \in \Gamma(V, \mathfrak{q})$ *we get the equivariance property:*

$$\rho(w) \left( F(x_1,, \ldots, x_\ell, y_1, \ldots, y_s)^{(k)} \right) = F(\rho(w)(x_1), \ldots, \rho(w)(x_\ell), y_1, \ldots, y_s)^{(k)}, \tag{469}$$

*where the superscript* $(k)$ *indicates the projection onto the multivector grade-$k$-part of the whole expression.*

We now investigate the image/range of the twisted conjugation.

**Theorem E.25** (The range of the twisted conjugation). *The image/range of the Clifford group* $\Gamma(V, \mathfrak{q})$ *under the twisted conjugation restricted to* $V$ *coincides with all orthogonal automorphisms of* $(V, \mathfrak{q})$ *that restrict to the identity* $\mathrm{id}_\mathcal{R}$ *of the radical subspace* $\mathcal{R} \subseteq V$ *of* $(V, \mathfrak{q})$:

$$\mathrm{ran}\,(\rho : \Gamma(V, \mathfrak{q}) \to \mathrm{GL}(V)) = \mathrm{O}_\mathcal{R}(V, \mathfrak{q}). \tag{470}$$

*Again, recall that the kernel is given by:*

$$\ker\,(\rho : \Gamma(V, \mathfrak{q}) \to \mathrm{GL}(V)) = \bigwedge^{[\times]}(\mathcal{R}). \tag{471}$$

*Proof.* We first show that the range of $\rho$ when restricted to $V$ will consistute an orthogonal automorphism of $(V, \mathfrak{q})$. For this let $e_1, \ldots, e_n$ be an orthogonal basis of $(V, \mathfrak{q})$ and $\mathcal{R}$ the radical subspace of $(V, \mathfrak{q})$, $r = \dim \mathcal{R}$. W.l.o.g. we can assume that $e_1, \ldots, e_m, e_{m+1}, \ldots, e_{m+r}$ with:

$$\mathcal{R} = \text{span}\{e_{m+1}, \ldots, e_{m+r}\}, \qquad E := \text{span}\{e_1, \ldots, e_m\}. \tag{472}$$

For $w \in \Gamma(V, \mathfrak{q})$ we now apply $\rho(w) \in \text{GL}(V)$ to the basis elements $e_i$. Note that by definition of $\Gamma(V, \mathfrak{q})$ we have $\rho(w)(e_i) \in V$. With this we get:

$$2\mathfrak{b}(\rho(w)(e_i), \rho(w)(e_j)) = \rho(w)(e_i)\rho(w)(e_j) + \rho(w)(e_j)\rho(w)(e_i) \tag{473}$$
$$= \rho(w)\left(e_i e_j + e_j e_i\right) \tag{474}$$
$$= \rho(w)\left(2\mathfrak{b}(e_i, e_j)\right) \tag{475}$$
$$= 2\mathfrak{b}(e_i, e_j). \tag{476}$$

This shows for $v = \sum_{i=1}^n a_i \cdot e_i$:

$$\mathfrak{q}(\rho(w)(v)) = \mathfrak{q}(\sum_{i=1}^n a_i \cdot \rho(w)(e_i)) \tag{477}$$

$$= \sum_{i=1}^n a_i^2 \cdot \mathfrak{q}(\rho(w)(e_i)) \tag{478}$$

$$= \sum_{i=1}^n a_i^2 \cdot \mathfrak{q}(e_i) \tag{479}$$

$$= \mathfrak{q}(\sum_{i=1}^n a_i \cdot e_i) \tag{480}$$

$$= \mathfrak{q}(v). \tag{481}$$

Since $\rho(w)$ is also a linear automorphism of $V$ with inverse $\rho(w^{-1})$ we see that:

$$\rho(w)|_V \in \text{O}(V, \mathfrak{q}). \tag{482}$$

By Lemma E.23 we also see that:

$$\rho(w)|_\mathcal{R} = \text{id}_\mathcal{R}. \tag{483}$$

Together this shows that:

$$\rho(w)|_V \in \text{O}_\mathcal{R}(V, \mathfrak{q}). \tag{484}$$

This shows the inclusion:

$$\text{ran}\left(\rho : \Gamma(V, \mathfrak{q}) \to \text{GL}(V)\right) \subseteq \text{O}_\mathcal{R}(V, \mathfrak{q}). \tag{485}$$

Recall the definition of the set of radical preserving orthogonal automorphisms:

$$\text{O}_\mathcal{R}(V, \mathfrak{q}) := \{\Phi \in \text{O}(V, \mathfrak{q}) \mid \Phi|_\mathcal{R} = \text{id}_\mathcal{R}\} \cong \begin{pmatrix} \text{O}(E, \mathfrak{q}|_E) & 0_{m \times r} \\ \text{M}(r, m) & \text{id}_\mathcal{R} \end{pmatrix} \tag{486}$$

$$\cong \text{O}(E, \mathfrak{q}|_E) \ltimes \text{M}(r, m). \tag{487}$$

So an element $\Phi \in \text{O}_\mathcal{R}(V, \mathfrak{q})$ can equivalently be written as:

$$\Phi = \begin{pmatrix} O & 0 \\ M & I \end{pmatrix} = \begin{pmatrix} O & 0 \\ 0 & I \end{pmatrix} \circ \begin{pmatrix} I & 0 \\ M & I \end{pmatrix} \tag{488}$$

$$= \begin{pmatrix} O_1 & 0 \\ 0 & I \end{pmatrix} \circ \cdots \circ \begin{pmatrix} O_k & 0 \\ 0 & I \end{pmatrix} \circ \begin{pmatrix} I & 0 \\ M_{1,1} & I \end{pmatrix} \circ \cdots \circ \begin{pmatrix} I & 0 \\ M_{m,r} & I \end{pmatrix}, \tag{489}$$

where $O = O_1 \cdots O_k$ is a product of $k \leq m$ reflection matrices $O_l$ by the Theorem of Cartan-Dieudonné C.13, and $M = \sum_{i=1}^m \sum_{j=1}^r M_{i,j}$ where the matrix $M_{i,j} = c_{i,j} \cdot I_{i,j}$ only has the entry $c_{i,j} \in \mathbb{F}$ at $(i, j)$ (and 0 otherwise). Now let $w_l \in V$ be the normal vector of the reflection $O_l$ with $\mathfrak{q}(w_l) \neq 0$ for $l = 1, \ldots, k$, and for $i = 1, \ldots, m$ and $j = 1, \ldots, r$ put:

$$\gamma_{i,j} := 1 + c_{i,j} \cdot e_i e_{m+j} \in \text{Cl}(V, \mathfrak{q}), \tag{490}$$

and further:

$$w := w_1 \cdots w_k \cdot \gamma_{1,1} \cdots \gamma_{m,r} \in \mathrm{Cl}(V, \mathfrak{q}). \tag{491}$$

Note that by Examples E.19 and E.20 we have:

$$\{w \in V \mid \mathfrak{q}(w) \neq 0\} \subseteq \Gamma(V, \mathfrak{q}), \tag{492}$$

$$\left\{ \gamma_i := 1 + e_i \sum_{j=1}^{r} c_{i,j} e_{m+j} \;\middle|\; c_{i,j} \in \mathbb{F}, i = 1, \ldots, m, \; j = 1, \ldots, r \right\} \subseteq \Gamma(V, \mathfrak{q}), \tag{493}$$

which implies that $w \in \Gamma(V, \mathfrak{q})$. With this we get:

$$\rho(w) = \rho(w_1) \circ \cdots \circ \rho(w_k) \circ \rho(\gamma_{1,1}) \circ \cdots \circ \rho(\gamma_{m,r}) \tag{494}$$

$$= \begin{pmatrix} O_1 & 0 \\ 0 & I \end{pmatrix} \circ \cdots \circ \begin{pmatrix} O_k & 0 \\ 0 & I \end{pmatrix} \circ \begin{pmatrix} I & 0 \\ M_{1,1} & I \end{pmatrix} \circ \cdots \circ \begin{pmatrix} I & 0 \\ M_{m,r} & I \end{pmatrix} \tag{495}$$

$$= \Phi. \tag{496}$$

This thus shows the surjectivity of the map:

$$\rho : \Gamma(V, \mathfrak{q}) \to \mathrm{O}_{\mathcal{R}}(V, \mathfrak{q}). \tag{497}$$

This shows the claim. $\qquad\square$

We can summarize our finding in the following statement.

**Corollary E.26.** *We have the short exact sequence:*

$$1 \longrightarrow \bigwedge\nolimits^{[\times]}(\mathcal{R}) \xrightarrow{\mathrm{incl}} \Gamma(V, \mathfrak{q}) \xrightarrow{\rho} \mathrm{O}_{\mathcal{R}}(V, \mathfrak{q}) \longrightarrow 1. \tag{498}$$

From the above Theorem we can now also derive the structure of the elements of the Clifford group.

**Corollary E.27** (Elements of the Clifford group). *Let $(V, \mathfrak{q})$ be a finite dimensional quadratic vector space of dimension $n := \dim V < \infty$ over a fields $\mathbb{F}$ with $\mathrm{char}(\mathbb{F}) \neq 2$. Let $\mathcal{R} \subseteq V$ be the radical vector subspace of $(V, \mathfrak{q})$ with dimension $r := \dim \mathcal{R}$. Put $m := n - r \geq 0$. Let $e_1, \ldots, e_n$ be an orthogonal basis of $(V, \mathfrak{q})$ ordered in such a way that $e_{m+1}, \ldots e_{m+r} = e_n$ are the basis vectors inside $\mathcal{R}$, while $e_1, \ldots, e_m$ are spanning a non-degenerate orthogonal subspace to $\mathcal{R}$ inside $(V, \mathfrak{q})$.*

*Then every element of the Clifford group $w \in \Gamma(V, \mathfrak{q})$ is of the form:*

$$w = c \cdot v_1 \cdots v_k \cdot \gamma_1 \cdots \gamma_m \cdot g, \tag{499}$$

*with $c \in \mathbb{F}^{\times}$, $k \in \mathbb{N}_0$, $v_l \in V$ with $\mathfrak{q}(v_l) \neq 0$ for $l = 1, \ldots, k$,*

$$\gamma_i = 1 + e_i \sum_{j=1}^{r} c_{i,j} e_{m+j}, \tag{500}$$

*with $c_{i,j} \in \mathbb{F}$ for $i = 1, \ldots, m$, $j = 1, \ldots, r$, and some $g \in \bigwedge\nolimits^{[*]}(\mathcal{R})$.*

### E.4 Orthogonal Representations of the Clifford Group

**Lemma E.28.** *Let $e_1, \ldots, e_n$ be an orthogonal basis of $(V, \mathfrak{q})$ and $w \in \Gamma(V, \mathfrak{q})$. If we put for $j \in [n]$:*

$$b_j := \rho(w)(e_j), \tag{501}$$

*and for $A \subseteq [n]$:*

$$b_A := \prod_{i \in A}^{<} b_i = \rho(w)(e_A), \tag{502}$$

*then $b_1, \ldots, b_n$ is an orthogonal basis for $(V, \mathfrak{q})$ and both $(e_A)_{A \in [n]}$ and $(b_A)_{A \in [n]}$ are orthogonal bases for $(\mathrm{Cl}(V, \mathfrak{q}), \bar{\mathfrak{q}})$.*

*Proof.* First note that $b_1, \ldots, b_n$ is a basis of $V$. Indeed, the releation:

$$0 = \sum_{i=1}^{n} a_i \cdot b_i, \tag{503}$$

with $a_i \in \mathbb{F}$ implies:

$$0 = \rho(w)^{-1}(0) \tag{504}$$

$$= \rho(w)^{-1}\left(\sum_{i=1}^{n} a_i \cdot b_i\right) \tag{505}$$

$$= \sum_{i=1}^{n} a_i \cdot \rho(w)^{-1}(b_i) \tag{506}$$

$$= \sum_{i=1}^{n} a_i \cdot e_i. \tag{507}$$

Since $e_1, \ldots, e_n$ is linear independent we get $a_i = 0$ for all $i \in [n]$. So also $b_1, \ldots, b_n$ is linear independent and thus constitute a basis of $V$.

To show that $b_1, \ldots, b_n$ is an orthogonal basis of $(V, \mathfrak{q})$ let $i \neq j$ and then consider the following:

$$2 \cdot \mathfrak{b}(b_i, b_j) = b_i b_j + b_j b_i \tag{508}$$

$$= \rho(w)(e_i)\rho(w)(e_j) + \rho(w)(e_j)\rho(w)(e_i) \tag{509}$$

$$= \rho(w)\left(e_i e_j + e_j e_i\right) \tag{510}$$

$$= \rho(w)(2 \cdot \underbrace{\mathfrak{b}(e_i, e_j)}_{=0}) \tag{511}$$

$$= 0. \tag{512}$$

This shows that $b_1, \ldots, b_n$ is an orthogonal basis of $(V, \mathfrak{q})$.

Theorem D.26 then shows that both $(e_A)_{A \in [n]}$ and $(b_A)_{A \in [n]}$ are orthogonal bases for $(\mathrm{Cl}(V, \mathfrak{q}), \bar{\mathfrak{q}})$. $\qquad \square$

**Theorem E.29.** *Let $w \in \Gamma(V, \mathfrak{q})$ and $x \in \mathrm{Cl}(V, \mathfrak{q})$ then we get:*

$$\bar{\mathfrak{q}}(\rho(w)(x)) = \bar{\mathfrak{q}}(x). \tag{513}$$

*In other words, $\rho(w) \in \mathrm{O}(\mathrm{Cl}(V, \mathfrak{q}), \bar{\mathfrak{q}})$. Furthermore, we have:*

$$\rho(w)|_{\bigwedge(\mathcal{R})} = \mathrm{id}_{\bigwedge(\mathcal{R})}. \tag{514}$$

*In other words, $\rho(w) \in \mathrm{O}_{\bigwedge(\mathcal{R})}(\mathrm{Cl}(V, \mathfrak{q}), \bar{\mathfrak{q}})$.*

*Proof.* Let $e_1, \ldots, e_n$ be an orthogonal basis for $(V, \mathfrak{q})$ and $b_j := \rho(w)(e_j)$ for $j \in [n]$. Then by Lemma E.28 we know that $b_1, \ldots, b_n$ is an orthogonal basis of $(V, \mathfrak{q})$ and both $(e_A)_{A \subseteq [n]}$ and $(b_A)_{A \subseteq [n]}$ are orthogonal basis for $(\mathrm{Cl}(V, \mathfrak{q}), \bar{\mathfrak{q}})$. Now let $x \in \mathrm{Cl}(V, \mathfrak{q})$ and write it as:

$$x = \sum_{A \subseteq [n]} x_A \cdot e_A, \qquad \rho(w)(x) = \sum_{A \subseteq [n]} x_A \cdot \rho(w)(e_A) = \sum_{A \subseteq [n]} x_A \cdot b_A. \tag{515}$$

Then we get:

$$\bar{\mathfrak{q}}(\rho(w)(x)) = \sum_{A\subseteq[n]} x_A \cdot \prod_{i\in A} \mathfrak{q}(b_i) \tag{516}$$

$$= \sum_{A\subseteq[n]} x_A \cdot \prod_{i\in A} \mathfrak{q}(\rho(w)(e_i)) \tag{517}$$

$$= \sum_{A\subseteq[n]} x_A \cdot \prod_{i\in A} \rho(w)(e_i)\rho(w)(e_i) \tag{518}$$

$$= \sum_{A\subseteq[n]} x_A \cdot \prod_{i\in A} \rho(w)(e_i^2) \tag{519}$$

$$= \sum_{A\subseteq[n]} x_A \cdot \prod_{i\in A} \rho(w)(\mathfrak{q}(e_i) \cdot 1) \tag{520}$$

$$= \sum_{A\subseteq[n]} x_A \cdot \prod_{i\in A} \mathfrak{q}(e_i) \cdot \rho(w)(1) \tag{521}$$

$$= \sum_{A\subseteq[n]} x_A \cdot \prod_{i\in A} \mathfrak{q}(e_i) \tag{522}$$

$$= \bar{\mathfrak{q}}(x). \tag{523}$$

This shows the claim. The remaining point follows from Lemma E.23. $\qquad\square$

Similarly, and more detailed, we also get the following, using Theorem D.27, Corollary E.17 and Lemma E.23:

**Corollary E.30.** *For every $w \in \Gamma(V,\mathfrak{q})$ and $m = 0, \ldots, n$ we have:*

$$\rho(w)|_{\mathrm{Cl}^{(m)}(V,\mathfrak{q})} \in \mathrm{O}_{\bigwedge^{(m)}(\mathcal{R})}(\mathrm{Cl}^{(m)}(V,\mathfrak{q}),\bar{\mathfrak{q}}). \tag{524}$$

*In words, $\rho(w)$, when restricted to the $m$-th homogeneous multivector component $\mathrm{Cl}^{(m)}(V,\mathfrak{q})$ of $\mathrm{Cl}(V,\mathfrak{q})$ acts as an orthogonal automorphism of $\mathrm{Cl}^{(m)}(V,\mathfrak{q})$ w.r.t. $\bar{\mathfrak{q}}$. Furthermore, it acts as the identity when further restricted to the $m$-th homogeneous multivector component of the radical subalgebra: $\bigwedge^{(m)}(\mathcal{R}) \subseteq \mathrm{Cl}^{(m)}(V,\mathfrak{q})$.*

**Remark E.31.** *For $m \in [n]$ we use $\rho^{(m)}$ to denote the group homomorphism $\rho$ when restricted to act on the subvector space $\mathrm{Cl}^{(m)}(V,\mathfrak{q})$:*

$$\rho^{(m)} : \Gamma(V,\mathfrak{q}) \to \mathrm{O}_{\bigwedge^{(m)}(\mathcal{R})}(\mathrm{Cl}^{(m)}(V,\mathfrak{q}),\bar{\mathfrak{q}}), \qquad \rho^{(m)}(w) := \rho(w)|_{\mathrm{Cl}^{(m)}(V,\mathfrak{q})}. \tag{525}$$

*By Theorem D.50 or Corollary E.22 we see that $\bigwedge^{[\times]}(\mathcal{R})$ always lies inside the kernel of $\rho^{(m)}$:*

$$\bigwedge^{[\times]}(\mathcal{R}) \subseteq \ker \rho^{(m)} \subseteq \Gamma(V,\mathfrak{q}). \tag{526}$$

*So we get a well-defined group homorphism on the quotient:*

$$\bar{\rho}^{(m)} : \Gamma(V,\mathfrak{q})/\bigwedge^{[\times]}(\mathcal{R}) \to \mathrm{O}_{\bigwedge^{(m)}(\mathcal{R})}(\mathrm{Cl}^{(m)}(V,\mathfrak{q}),\bar{\mathfrak{q}}), \quad \bar{\rho}^{(m)}([w]) := \rho(w)|_{\mathrm{Cl}^{(m)}(V,\mathfrak{q})}. \tag{527}$$

*Furthermore, by Theorem E.25 we have the isomorphism:*

$$\bar{\rho}^{(1)} : \Gamma(V,\mathfrak{q})/\bigwedge^{[\times]}(\mathcal{R}) \cong \mathrm{O}_{\bigwedge^{(1)}(\mathcal{R})}(\mathrm{Cl}^{(1)}(V,\mathfrak{q}),\bar{\mathfrak{q}}) = \mathrm{O}_{\mathcal{R}}(V,\mathfrak{q}), \quad \bar{\rho}^{(1)}([w]) = \rho(w)|_V. \tag{528}$$

*Consequently, for all $m \in [n]$ we get the composition of group homomorphisms:*

$$\tilde{\rho}^{(m)} : \mathrm{O}_{\mathcal{R}}(V,\mathfrak{q}) \overset{(\bar{\rho}^{(1)})^{-1}}{\cong} \Gamma(V,\mathfrak{q})/\bigwedge^{[\times]}(\mathcal{R}) \overset{\bar{\rho}^{(m)}}{\to} \mathrm{O}_{\bigwedge^{(m)}(\mathcal{R})}(\mathrm{Cl}^{(m)}(V,\mathfrak{q}),\bar{\mathfrak{q}}), \tag{529}$$

$$\tilde{\rho}^{(m)}(\Phi) = \rho(w)|_{\mathrm{Cl}^{(m)}(V,\mathfrak{q})}, \quad \text{for any } w \in \Gamma(V,\mathfrak{q}) \text{ with } \rho^{(1)}(w) = \Phi. \tag{530}$$

*Similarly, we get an (injective) group homomorphism:*

$$\tilde{\rho} : \mathrm{O}_{\mathcal{R}}(V, \mathfrak{q}) \to \mathrm{O}_{\bigwedge(\mathcal{R})}(\mathrm{Cl}(V, \mathfrak{q}), \bar{\mathfrak{q}}), \tag{531}$$

$$\tilde{\rho}(\Phi) := \rho(w), \quad \textit{for any } w \in \Gamma(V, \mathfrak{q}) \textit{ with } \rho^{(1)}(w) = \Phi. \tag{532}$$

*So, the group $\mathrm{O}_{\mathcal{R}}(V, \mathfrak{q})$ acts on $\mathrm{Cl}(V, \mathfrak{q})$ and all subvector spaces $\mathrm{Cl}^{(m)}(V, \mathfrak{q})$, $m = 0, \ldots, n$, in the same way as $\Gamma(V, \mathfrak{q})$ does via $\rho$, when using the surjective map $\rho^{(1)}$ to lift elements $\Phi \in \mathrm{O}_{\mathcal{R}}(V, \mathfrak{q})$ to elements $w \in \Gamma(V, \mathfrak{q})$ with $\rho^{(1)}(w) = \Phi$.*

*Specifically, let $x \in \mathrm{Cl}(V, \mathfrak{q})$ be of the form $x = \sum_{i \in I} c_i \cdot v_{i,1} \cdots v_{i,k_i}$ with $v_{i,j} \in V$, $c_i \in \mathbb{F}$ and $\Phi \in \mathrm{O}_{\mathcal{R}}(V, \mathfrak{q})$ is given by $\Phi = \bar{\rho}^{(1)}([w])$ with $w \in \Gamma(V, \mathfrak{q})$, then we have:*

$$\rho(w)(x) = \sum_{i \in I} c_i \cdot \rho(w)(v_{i,1}) \cdots \rho(w)(v_{i,k_i}) = \sum_{i \in I} c_i \cdot \Phi(v_{i,1}) \cdots \Phi(v_{i,k_i}). \tag{533}$$

*This means that the action $\tilde{\rho}$ of $\mathrm{O}_{\mathcal{R}}(V, \mathfrak{q})$ transforms a multivector $x$ by transforming all its vector components $v_{i,j}$ by $\Phi$, acting through the usual orthogonal transformation on vectors.*

As such, we have the following equivariance property with respect to $\mathrm{O}_{\mathcal{R}}(V, \mathfrak{q})$.

**Corollary E.32.** *Let $F(T_1, \ldots, T_{\ell+s}) \in \mathbb{F}[T_1, \ldots, T_{\ell+s}]$ be a polynomial in $\ell + s$ variables with coefficients in $\mathbb{F}$ and let $k \in \{0, \ldots, n\}$. Further, consider $\ell$ elements $x_1, \ldots, x_\ell \in \mathrm{Cl}(V, \mathfrak{q})$ and $s$ elements $y_1, \ldots, y_s \in \bigwedge(\mathcal{R})$. Then for every $\Phi \in \mathrm{O}_{\mathcal{R}}(V, \mathfrak{q})$ we get the equivariance property:*

$$\tilde{\rho}(\Phi) \left( F(x_1,, \ldots, x_\ell, y_1, \ldots, y_s)^{(k)} \right) = F(\tilde{\rho}(\Phi)(x_1), \ldots, \tilde{\rho}(\Phi)(x_\ell), y_1, \ldots, y_s)^{(k)}, \tag{534}$$

*where the superscript $(k)$ indicates the projection onto the multivector grade-$k$-part of the whole expression.*

### E.5  The Spinor Norm and the Clifford Norm

In this subsection we shortly introduce the three slightly different versions of a norm that appear in the literature: the Spinor norm, the Clifford norm and the extended quadratic form. We are interested under which conditions do they have multiplicative behaviour.

**Definition E.33** (The Spinor norm and the Clifford norm)**.** *We define the Spinor norm and the Clifford norm of $\mathrm{Cl}(V, \mathfrak{q})$ as the maps:*

$$\mathrm{SN} : \mathrm{Cl}(V, \mathfrak{q}) \to \mathrm{Cl}(V, \mathfrak{q}), \qquad\qquad \mathrm{SN}(x) := \beta(x)x, \tag{535}$$

$$\mathrm{CN} : \mathrm{Cl}(V, \mathfrak{q}) \to \mathrm{Cl}(V, \mathfrak{q}), \qquad\qquad \mathrm{CN}(x) := \gamma(x)x. \tag{536}$$

*Also recall the extended quadratic form:*

$$\bar{\mathfrak{q}} : \mathrm{Cl}(V, \mathfrak{q}) \to \mathbb{F}, \qquad\qquad \bar{\mathfrak{q}}(x) = \zeta(\beta(x)x) = \zeta(\mathrm{SN}(x)). \tag{537}$$

As a first preliminary Lemma we need to study when the projection onto the zero-component is multiplicative:

**Lemma E.34.** *Let $x \in \bigwedge(\mathcal{R})$ and $y \in \mathrm{Cl}(V, \mathfrak{q})$ then we have:*

$$\zeta(xy) = \zeta(x)\,\zeta(y). \tag{538}$$

*As a result, the projection onto the zero component induces an $\mathbb{F}$-algebra homomorphism:*

$$\zeta : \bigwedge(\mathcal{R}) \to \mathbb{F}, \qquad\qquad y \mapsto \zeta(y). \tag{539}$$

*Proof.* We now use the notations from D.40 and Lemma D.41. We distinguish two cases: $x \in \bigwedge^{(\geq 1)}(\mathcal{R})$ and:

$$x \in \bigwedge(\mathcal{R}) \setminus \bigwedge^{(\geq 1)}(\mathcal{R}) = \bigwedge^{\times}(\mathcal{R}) = \mathbb{F}^{\times} + \bigwedge^{(\geq 1)}(\mathcal{R}. \tag{540}$$

In the first case, we have: $\zeta(xy) = 0 = \zeta(x)\,\zeta(y)$, as multiplying with $x \in \bigwedge^{(\geq 1)}(\mathcal{R})$ can only increase the grade of occurring terms or make them vanish.

In the second case, we can write $x = a + f$ with $a \in \mathbb{F}^{\times}$ and $f \in \bigwedge^{(\geq 1)}(\mathcal{R})$. Clearly, $\zeta(x) = a$. We then get by linearity and the first case:

$$\zeta(xy) = \zeta(ay + fy) = a\,\zeta(y) + \zeta(fy) = \zeta(x)\,\zeta(y) + 0. \tag{541}$$

This shows the claim. $\qquad\square$

**Lemma E.35.** *Let $x_1, x_2 \in \mathrm{Cl}(V, \mathfrak{q})$.*

1. *If $\mathrm{SN}(x_1) \in \mathfrak{Z}(\mathrm{Cl}(V, \mathfrak{q}))$ then we have:*
$$\mathrm{SN}(x_1 x_2) = \mathrm{SN}(x_1)\,\mathrm{SN}(x_2). \tag{542}$$

2. *If $\mathrm{CN}(x_1) \in \mathfrak{Z}(\mathrm{Cl}(V, \mathfrak{q}))$ then we have:*
$$\mathrm{CN}(x_1 x_2) = \mathrm{CN}(x_1)\,\mathrm{CN}(x_2). \tag{543}$$

3. *If $\mathrm{SN}(x_1) \in \bigwedge^{[0]}(\mathcal{R})$ or $\mathfrak{q} = 0$ then we have:*
$$\bar{\mathfrak{q}}(x_1 x_2) = \bar{\mathfrak{q}}(x_1)\bar{\mathfrak{q}}(x_2). \tag{544}$$

*Proof.* $\mathrm{SN}(x_1) \in \mathfrak{Z}(\mathrm{Cl}(V, \mathfrak{q}))$ implies:
$$\mathrm{SN}(x_1 x_2) = \beta(x_1 x_2) x_1 x_2 \tag{545}$$
$$= \beta(x_2)\beta(x_1) x_1 x_2 \tag{546}$$
$$= \beta(x_2)\,\mathrm{SN}(x_1) x_2 \tag{547}$$
$$\overset{\mathrm{SN}(x_1) \in \mathfrak{Z}(\mathrm{Cl}(V,\mathfrak{q}))}{=} \mathrm{SN}(x_1)\beta(x_2) x_2 \tag{548}$$
$$= \mathrm{SN}(x_1)\,\mathrm{SN}(x_2). \tag{549}$$
Similarly for CN.

Together with Lemma E.34 and $\mathrm{SN}(x_1) \in \bigwedge^{[0]}(\mathcal{R}) \subseteq \bigwedge(\mathcal{R}) \cap \mathfrak{Z}(\mathrm{Cl}(V, \mathfrak{q}))$ we get:
$$\bar{\mathfrak{q}}(x_1 x_2) = \zeta\left(\mathrm{SN}(x_1 x_2)\right) \tag{550}$$
$$= \zeta\left(\mathrm{SN}(x_1)\,\mathrm{SN}(x_2)\right) \tag{551}$$
$$= \zeta(\mathrm{SN}(x_1))\,\zeta(\mathrm{SN}(x_2)) \tag{552}$$
$$= \bar{\mathfrak{q}}(x_1)\bar{\mathfrak{q}}(x_2). \tag{553}$$
This shows the claim. $\square$

**Lemma E.36.** *Consider the following subset of $\mathrm{Cl}(V, \mathfrak{q})$:*
$$\Gamma^{[-]}(V, \mathfrak{q}) := \left\{ x \in \mathrm{Cl}^{[0]}(V, \mathfrak{q}) \cup \mathrm{Cl}^{[1]}(V, \mathfrak{q}) \,\middle|\, \forall v \in V \,\exists v' \in V.\, \alpha(x)v = v'x \right\}. \tag{554}$$
*Then $\Gamma^{[-]}(V, \mathfrak{q})$ is closed under multiplication and for every $x \in \Gamma^{[-]}(V, \mathfrak{q})$ we have:*
$$\mathrm{SN}(x) \in \bigwedge^{[0]}(\mathcal{R}), \qquad\qquad \mathrm{CN}(x) \in \bigwedge^{[0]}(\mathcal{R}). \tag{555}$$

*Proof.* For $x, y \in \Gamma^{[-]}(V, \mathfrak{q})$ we also have that $xy$ is homogeneous. Furthermore, we get for $v \in V$:
$$\alpha(xy)v = \alpha(x)\alpha(y)v \tag{556}$$
$$= \alpha(x)v'y \tag{557}$$
$$= \tilde{v}xy, \tag{558}$$
for some $v', \tilde{v} \in V$. So, $xy \in \Gamma^{[-]}(V, \mathfrak{q})$ and $\Gamma^{[-]}(V, \mathfrak{q})$ is closed under multiplication.

With the above conditions on $x$ we get for every $v \in V$:
$$\alpha(\mathrm{SN}(x))v = \alpha(\beta(x))\alpha(x)v \tag{559}$$
$$= \alpha(\beta(x))v'x \tag{560}$$
$$= \alpha(\beta(x))(-\alpha(\beta(v')))x \tag{561}$$
$$= -\alpha(\beta(x)\beta(v'))x \tag{562}$$
$$= -\alpha(\beta(v'x))x \tag{563}$$
$$= -\alpha(\beta(\alpha(x)v))x \tag{564}$$
$$= -\beta(\alpha(\alpha(x)v))x \tag{565}$$
$$= -\beta(x\alpha(v))x \tag{566}$$
$$= \beta(xv)x \tag{567}$$
$$= \beta(v)\beta(x)x \tag{568}$$
$$= v\,\mathrm{SN}(x). \tag{569}$$

This implies by Theorem D.50 that:

$$\mathrm{SN}(x) \in \bigwedge(\mathcal{R}). \tag{570}$$

Since for homogeneous $x \in \mathrm{Cl}(V, \mathfrak{q})$ we have: $\mathrm{prt}(\beta(x)) = \mathrm{prt}(x)$ and thus: $\mathrm{SN}(x) = \beta(x)x \in \mathrm{Cl}^{[0]}(V, \mathfrak{q})$. This implies:

$$\mathrm{SN}(x) \in \bigwedge(\mathcal{R}) \cap \mathrm{Cl}^{[0]}(V, \mathfrak{q}) = \bigwedge^{[0]}(\mathcal{R}). \tag{571}$$

This shows the claim. $\qquad\square$

**Theorem E.37** (Multiplicativity of the three different norms)**.** *Both, the Spinor norm and the Clifford norm, when restricted to the Clifford group, are well-defined group homomorphisms:*

$$\mathrm{SN} : \Gamma(V, \mathfrak{q}) \to \bigwedge^{[\times]}(\mathcal{R}), \qquad\qquad w \mapsto \mathrm{SN}(w) = \beta(w)w, \tag{572}$$

$$\mathrm{CN} : \Gamma(V, \mathfrak{q}) \to \bigwedge^{[\times]}(\mathcal{R}), \qquad\qquad w \mapsto \mathrm{CN}(w) = \gamma(w)w. \tag{573}$$

*Furthermore, the extended quadratic form $\bar{\mathfrak{q}}$ of $\mathrm{Cl}(V, \mathfrak{q})$ restricted to the Clifford group is a well-defined group homomorphism:*

$$\bar{\mathfrak{q}} : \Gamma(V, \mathfrak{q}) \to \mathbb{F}^{\times}, \qquad\qquad w \mapsto \bar{\mathfrak{q}}(w) = \zeta(\beta(w)w). \tag{574}$$

*Proof.* This directly follows from Lemma E.35 and Lemma E.36. Note that $\Gamma(V, \mathfrak{q}) \subseteq \Gamma^{[-]}(V, \mathfrak{q})$.
$\qquad\square$

**Example E.38.**     *1. For $a \in \mathbb{F}$ we get:*

$$\mathrm{SN}(a) = \beta(a)a = a^2. \tag{575}$$

*This also shows: $\bar{\mathfrak{q}}(a) = a^2 \bar{\mathfrak{q}}(1) = a^2$.*

*2. For $w \in V$ we have:*

$$\mathrm{SN}(w) = \beta(w)w = w^2 = \mathfrak{q}(w). \tag{576}$$

*This also shows: $\bar{\mathfrak{q}}(w) = \mathfrak{q}(w)$.*

*3. For $\gamma = 1 + ef$ with $e \in V$ and $f \in \mathcal{R}$ we have:*

$$\mathrm{SN}(\gamma) = (1 + fe)(1 + ef) \tag{577}$$
$$= 1 + fe + ef + feef \tag{578}$$
$$= 1 + 2\mathfrak{b}(e, f) + \mathfrak{q}(f) \cdot \mathfrak{q}(e) \tag{579}$$
$$= 1. \tag{580}$$

*This also shows: $\bar{\mathfrak{q}}(\gamma) = 1$.*

*4. For $g = 1 + h \in \bigwedge^*(\mathcal{R})$ we get:*

$$\mathrm{SN}(g) = (1 + \beta(h))(1 + h) \tag{581}$$
$$= 1 + \beta(h) + h + \beta(h)h, \tag{582}$$

*and thus: $\bar{\mathfrak{q}}(g) = 1$.*

### E.6 The Pin Group and the Spin Group

We have investigated the Clifford group and its action on the algebra through the twisted conjugation. In many fields of study, the Clifford group is further restricted to the Pin or Spin group. There can arise a few issues regarding the exact definition of these groups, especially when also considering fields other than the reals. We elaborate on these concerns here and leave the general definition for future discussion.

**Motivation E.39** (The problem of generalizing the definition of the Spin group)**.** *For a positive definite quadratic form* $\mathfrak{q}$ *on the real vector space* $V = \mathbb{R}^n$ *with* $n \geq 3$ *the* Spin *group* $\mathrm{Spin}(n)$ *is defined via the kernel of the Spinor norm (=extended quadratic form on* $\mathrm{Cl}(V, \mathfrak{q})$*) restricted to the special Clifford group* $\Gamma^{[0]}(V, \mathfrak{q})$*:*

$$\mathrm{Spin}(n) := \ker\left(\bar{\mathfrak{q}} : \Gamma^{[0]}(V, \mathfrak{q}) \to \mathbb{R}^{\times}\right) = \left\{w \in \Gamma^{[0]}(V, \mathfrak{q}) \,\middle|\, \bar{\mathfrak{q}}(w) = 1\right\} = \bar{\mathfrak{q}}|^{-1}_{\Gamma^{[0]}(V,\mathfrak{q})}(1). \quad (583)$$

$\mathrm{Spin}(n)$ *is thus a normal subgroup of the special Clifford group* $\Gamma^{[0]}(V, \mathfrak{q})$*, and, as it turns out, a double cover of the* special orthogonal group $\mathrm{SO}(n)$ *via the twisted conjugation* $\rho$*. The latter can be summarized by the short exact sequence:*

$$1 \longrightarrow \{\pm 1\} \xrightarrow{\mathrm{incl}} \mathrm{Spin}(n) \xrightarrow{\rho} \mathrm{SO}(n) \longrightarrow 1. \quad (584)$$

*We intend to generalize this in several directions: 1. from Spin to Pin group, 2. from* $\mathbb{R}^n$ *to vector spaces* $V$ *over general fields* $\mathbb{F}$ *with* $\mathrm{char}(\mathbb{F}) \neq 2$*, 3. from non-degenerate to degenerate quadratic forms* $\mathfrak{q}$*, 4. from positive (semi-)definite to non-definite quadratic forms* $\mathfrak{q}$*. This comes with several challenges and ambiguities.*

*If we want to generalize the above to define the Pin group we would allow for elements not just of (pure) even parity* $w \in \Gamma^{[0]}(V, \mathfrak{q})$*. Here the question arises if one should generalize to the unconstrained Clifford group* $\tilde{\Gamma}(V, \mathfrak{q})$ *or the (homogeneous) Clifford group* $\Gamma(V, \mathfrak{q})$*. As discussed before, to ensure that the (adjusted) twisted conjugation* $\rho$ *is a well-defined algebra automorphism of* $\mathrm{Cl}(V, \mathfrak{q})$ *the parity homogeneity assumptions is crucial. Furthermore, the elements of* $\tilde{\Gamma}(V, \mathfrak{q})$ *that lead to non-trivial orthogonal automorphisms, e.g.* $v \in V$ *with* $\mathfrak{q}(v) \neq 0$ *and* $\gamma = 1 + ef$ *with* $e \in V$*,* $f \in \mathcal{R}$*, are already homogeneous. So it is arguably safe and reasonable to restrict to the Clifford group* $\Gamma(V, \mathfrak{q})$ *and define the Pin group* $\mathrm{Pin}(V, \mathfrak{q})$ *as some subquotient of* $\Gamma(V, \mathfrak{q})$*.*

*In the non-definite (but still non-degenerate, real) case* $\mathbb{R}^{(p,q)}$*,* $p, q \geq 1$*, the special orthogonal group* $\mathrm{SO}(p, q)$ *contains combinations of reflections* $r_0 \circ r_1$ *where the corresponding normal vectors* $v_0, v_1 \in V$ *satisfy* $\mathfrak{q}(v_0) = 1$ *and* $\mathfrak{q}(v_1) = -1$*. Their product* $v_0 v_1$ *would lie in the special Clifford group* $\Gamma^{[0]}(V, \mathfrak{q})$*. However, their spinor norm would be different from 1:*

$$\bar{\mathfrak{q}}(v_0 v_1) = \mathfrak{q}(v_0)\mathfrak{q}(v_1) = -1 \neq 1. \quad (585)$$

*Here now the question arises if we would like to preserve the former definition of* $\mathrm{Spin}(p, q)$ *as* $\bar{\mathfrak{q}}|^{-1}_{\Gamma^{[0]}(V,\mathfrak{q})}(1)$ *and exclude* $v_0 v_1$ *from* $\mathrm{Spin}(p, q)$*, or, if we adjust the definition of* $\mathrm{Spin}(p, q)$ *and include* $v_0 v_1$*. The former definition has the effect that* $\mathrm{Spin}(p, q)$ *does, in general, not map surjectively onto* $\mathrm{SO}(p, q)$ *anymore, and we would only get a short exact sequence:*

$$1 \longrightarrow \{\pm 1\} \xrightarrow{\mathrm{incl}} \mathrm{Spin}(p, q) \xrightarrow{\rho} \mathrm{SO}(p, q) \xrightarrow{\bar{\bar{\mathfrak{q}}}} \underbrace{\mathbb{R}^{\times}/(\mathbb{R}^{\times})^2}_{\cong \{\pm 1\}}. \quad (586)$$

*The alternative would be to define* $\mathrm{Spin}(p, q)$ *as* $\bar{\mathfrak{q}}|^{-1}_{\Gamma^{[0]}(V,\mathfrak{q})}(\pm 1)$*. This would allow for* $v_0 v_1 \in \mathrm{Spin}(p, q)$ *and lead to the short exact sequence:*

$$1 \longrightarrow \underbrace{\mu_4(\mathbb{R})}_{=\{\pm 1\}} \xrightarrow{\mathrm{incl}} \mathrm{Spin}(p, q) \xrightarrow{\rho} \mathrm{SO}(p, q) \longrightarrow 1, \quad (587)$$

*which exactly recovers the former behaviour for* $\mathrm{Spin}(n)$*, and, which makes* $\mathrm{Spin}(p, q)$ *a double cover of* $\mathrm{SO}(p, q)$*.*

*However, for other fields* $\mathbb{F}$*,* $\mathrm{char}(\mathbb{F}) \neq 2$*, and non-degenerate* $(V, \mathfrak{q})$*,* $\dim V \geq 3$*, one would get, with the last definition* $\bar{\mathfrak{q}}|^{-1}_{\Gamma^{[0]}(V,\mathfrak{q})}(\pm 1)$*, the exact sequence:*

$$1 \longrightarrow \mu_4(\mathbb{F}) \xrightarrow{\mathrm{incl}} \mathrm{Spin}(V, \mathfrak{q}) \xrightarrow{\rho} \mathrm{SO}(V, \mathfrak{q}) \xrightarrow{\bar{\bar{\mathfrak{q}}}} \mathbb{F}^{\times}/(\mathbb{F}^{\times})^2, \quad (588)$$

and one would need to live with the fact that: a) $\mu_4(\mathbb{F}) := \left\{ x \in \mathbb{F}^\times \,\middle|\, x^4 = 1 \right\}$ could contain a number of elements $k$ different from 2, rendering $\rho$ a $(k : 1)$-map, in contrast to a $(2 : 1)$-map, and, again, b) the non-surjectivity of $\rho$. The latter comes from the fact that a combination of reflections $r_0 \circ r_1$ where the corresponding normal vectors $v_0, v_1 \in V$ with $\mathfrak{q}(v_0), \mathfrak{q}(v_1) \neq 0$ cannot, in general, be normalized such that the $\mathfrak{q}(v_i)$'s lie in $\{\pm 1\}$ by multiplying/dividing the $v_i$'s with some scalars $c_i \in \mathbb{F}^\times$. Note that we get:

$$\mathfrak{q}(v_i/c_i) = \mathfrak{q}(v_i)/c_i^2. \tag{589}$$

This shows that we can only normalize the $v_i$'s such that the $\mathfrak{q}(v_i)$'s lie inside a fixed system of representatives $S \subseteq \mathbb{F}^\times$ of $\mathbb{F}^\times/(\mathbb{F}^\times)^2$, which is thus of size:

$$\#S = \# \left( \mathbb{F}^\times/(\mathbb{F}^\times)^2 \right), \tag{590}$$

which can be different from $2 = \# \{\pm 1\}$.

So, in this general setting, the first definition of $\mathrm{Spin}(V, \mathfrak{q})$ as $\bar{\mathfrak{q}}|_{\Gamma^{[0]}(V,\mathfrak{q})}^{-1}(1)$ would at least correct the map $\rho$ to be a $(2 : 1)$-map. We would get the following short exact seequence:

$$1 \longrightarrow \{\pm 1\} \xrightarrow{\mathrm{incl}} \mathrm{Spin}(V, \mathfrak{q}) \xrightarrow{\rho} \mathrm{SO}(V, \mathfrak{q}) \xrightarrow{\bar{\bar{\mathfrak{q}}}} \mathbb{F}^\times/(\mathbb{F}^\times)^2. \tag{591}$$

The normalization argument around Equation 589 for general fields $\mathbb{F}$ now would also give us a third option: we could, instead of restricting elements to have a fixed value $\mathfrak{q}(v_i) \in S$, which depends on the choice of $S$, identify elements $w_1, w_2 \in \Gamma^{[0]}(V, \mathfrak{q})$ if they differ by a scalar $c \in \mathbb{F}^\times$:

$$w_1 = c \cdot w_2. \tag{592}$$

Then for their spinor norms (modulo $(\mathbb{F}^\times)^2$) we would get:

$$\bar{\mathfrak{q}}(w_1) = c^2 \cdot \bar{\mathfrak{q}}(w_2), \qquad\qquad [\bar{\mathfrak{q}}(w_1)] = [\bar{\mathfrak{q}}(w_2)] \in \mathbb{F}^\times/(\mathbb{F}^\times)^2. \tag{593}$$

So, the spinor norms of $w_1$ and $w_2$ would be represented by the same representative $s \in S$, as desired. However, this definition would just identify $\mathrm{Spin}(V, \mathfrak{q})$ with $\mathrm{SO}(V, \mathfrak{q})$ via $\rho$:

$$\mathrm{Spin}(V, \mathfrak{q}) = \Gamma^{[0]}(V, \mathfrak{q})/\mathbb{F}^\times \overset{\rho}{\cong} \mathrm{SO}(V, \mathfrak{q}), \tag{594}$$

and nothing new would emerge from this. Note that the latter isomorphisms always holds for non-degenerate $(V, \mathfrak{q})$ with $\dim V \geq 3$, $\mathrm{char}(\mathbb{F}) \neq 2$, and can be expressed as the exact sequence:

$$1 \longrightarrow \mathbb{F}^\times \xrightarrow{\mathrm{incl}} \Gamma^{[0]}(V, \mathfrak{q}) \xrightarrow{\rho} \mathrm{SO}(V, \mathfrak{q}) \longrightarrow 1. \tag{595}$$

A fourth option would be to mod out the scalar squares $(\mathbb{F}^\times)^2$ instead of $\mathbb{F}^\times$ and use $\Gamma^{[0]}(V, \mathfrak{q})/(\mathbb{F}^\times)^2$ as the definition of $\mathrm{Spin}(V, \mathfrak{q})$. This would lead to the exact sequence:

$$1 \longrightarrow \mathbb{F}^\times/(\mathbb{F}^\times)^2 \xrightarrow{\mathrm{incl}} \mathrm{Spin}(V, \mathfrak{q}) \xrightarrow{\rho} \mathrm{SO}(V, \mathfrak{q}) \longrightarrow 1, \tag{596}$$

which would again coincide with the real case of $\mathrm{Spin}(n)$ as then $\mathbb{R}^\times/(\mathbb{R}^\times)^2 \cong \{\pm 1\}$. However, in the general case, again, $\rho$ is here a $(k : 1)$-map instead of a $(2 : 1)$-map with $k := \# \left( \mathbb{F}^\times/(\mathbb{F}^\times)^2 \right)$. Furthermore, the spinor norm $\bar{\mathfrak{q}}$ on $\mathrm{Spin}(V, \mathfrak{q})$ would not be well-defined anymore, in its current form, as different representatives of elements $[w_1] = [w_2] \in \Gamma^{[0]}(V, \mathfrak{q})/(\mathbb{F}^\times)^2$ would differ by a scalar square: $w_1 = c^2 \cdot w_2$. Their spinor norms would thus differ by by a forth scalar power:

$$\bar{\mathfrak{q}}(w_1) = \bar{\mathfrak{q}}(c^2 \cdot w_2) = c^4 \cdot \bar{\mathfrak{q}}(w_2). \tag{597}$$

So, the spinor norm on $\Gamma^{[0]}(V, \mathfrak{q})/(\mathbb{F}^\times)^2$ would only be well defined modulo $(\mathbb{F}^\times)^4 \subseteq (\mathbb{F}^\times)^2$:

$$[\bar{\mathfrak{q}}] : \Gamma^{[0]}(V, \mathfrak{q})/(\mathbb{F}^\times)^2 \to \mathbb{F}^\times/(\mathbb{F}^\times)^4, \qquad\qquad [w] \mapsto [\bar{\mathfrak{q}}(w)]. \tag{598}$$

Things become even more complicated in the degenerate case. At least we always have a short exact sequence for $m \geq 3$:

$$1 \longrightarrow \bigwedge^{[\times]}(\mathcal{R}) \xrightarrow{\mathrm{incl}} \Gamma^{[0]}(V, \mathfrak{q}) \xrightarrow{\rho} \mathrm{SO}_\mathcal{R}(V, \mathfrak{q}) \longrightarrow 1, \tag{599}$$

*where* $\mathrm{SO}_{\mathcal{R}}(V,\mathfrak{q})$ *indicates the set of those special orthogonal automorphisms* $\Phi$ *of* $(V,\mathfrak{q})$ *with* $\Phi|_{\mathcal{R}} = \mathrm{id}_{\mathcal{R}}$, *where* $\mathcal{R}$ *is the radical subspace of* $(V,\mathfrak{q})$ *with* $r := \dim \mathcal{R}$, $n := \dim V$, $m := n - r$. *Recall that we have:*

$$\bigwedge\nolimits^{[\times]}(\mathcal{R}) = \mathbb{F}^{\times} \cdot \bigwedge\nolimits^{[*]}(\mathcal{R}), \tag{600}$$

$$\bigwedge\nolimits^{[*]}(\mathcal{R}) = 1 + \mathrm{span}\left\{f_1 \cdots f_k \mid k \geq 2 \text{ even}, f_l \in \mathcal{R}, l = 1, \ldots, k\right\}. \tag{601}$$

*Note that for* $g \in \bigwedge^{[*]}(\mathcal{R})$ *we have* $\rho(g)|_V = \mathrm{id}_V$ *and* $\bar{\mathfrak{q}}(g) = 1$. *So the elements from* $\bigwedge^{[*]}(\mathcal{R})$ *are only blowing up the kernels of* $\rho$ *and* $\bar{\mathfrak{q}}$ *and can be considered redundant for our analysis. So one can argue that one can mod out* $\bigwedge^{[*]}(\mathcal{R})$ *in the above groups. We thus get a short exact sequence:*

$$1 \longrightarrow \mathbb{F}^{\times} \longrightarrow \underbrace{\Gamma^{[0]}(V,\mathfrak{q}) / \bigwedge\nolimits^{[*]}(\mathcal{R})}_{=:\tilde{\Gamma}^{[0]}(V,\mathfrak{q})} \overset{\rho}{\longrightarrow} \mathrm{SO}_{\mathcal{R}}(V,\mathfrak{q}) \longrightarrow 1, \tag{602}$$

*which now looks similar to the non-degenerate case. We can now consider the same 4 options for the definition of the Spin group as before:*

$$\bar{\mathfrak{q}}|^{-1}_{\tilde{\Gamma}^{[0]}(V,\mathfrak{q})}(1), \qquad \bar{\mathfrak{q}}|^{-1}_{\tilde{\Gamma}^{[0]}(V,\mathfrak{q})}(\pm 1), \qquad \tilde{\Gamma}^{[0]}(V,\mathfrak{q})/\mathbb{F}^{\times}, \qquad \tilde{\Gamma}^{[0]}(V,\mathfrak{q})/(\mathbb{F}^{\times})^2. \tag{603}$$

*As before, the third option can easily be discarded. If we want to preserve generality, we have the option to either preserve* $\bar{\mathfrak{q}}$ *and the* $(2:1)$-*property of* $\rho$ *and pick the first option, or, preserve the surjectivity of* $\rho$ *and take the fourth option. If we are only interested in the* $\mathbb{R}$-*case, then the second option preserves all properties. Note that in the* $\mathbb{R}$-*case the groups of the second and forth option are isomorphic as groups anyways:*

$$\bar{\mathfrak{q}}|^{-1}_{\tilde{\Gamma}^{[0]}(V,\mathfrak{q})}(\pm 1) \cong \tilde{\Gamma}^{[0]}(V,\mathfrak{q})/(\mathbb{R}^{\times})^2. \tag{604}$$

*The reason is that we always have that:* $\bar{\mathfrak{q}}(\mathbb{F}^{\times}) = (\mathbb{F}^{\times})^2$, *and, in the* $\mathbb{R}$-*case, the left group already contains the relevant elements to also map surjectively onto* $\mathrm{SO}_{\mathcal{R}}(V,\mathfrak{q})$ *via* $\rho$.

*One could further discuss if one wanted to replace the extended quadratic form* $\bar{\mathfrak{q}}$, *which is given by* $\bar{\mathfrak{q}}(x) = \zeta(\beta(x)x) \in \mathbb{F}^{\times}$, *by the other possible definition of the spinor norm* $\mathrm{SN}$, *which is only given by* $\mathrm{SN}(x) = \beta(x)x \in \bigwedge^{[\times]}(\mathcal{R})$. *However, for all relevant elements of* $\Gamma^{[0]}(V,\mathfrak{q})$ *both definitions agree, but the description of the set of the rather irrelevant elements, which satisfy* $\mathrm{SN}(x) = 1$ *and* $\rho(x)|_V = \mathrm{id}_V$, *becomes more complicated than the set* $\bigwedge^{[\times]}(\mathcal{R})$. *As we mod those irrelevant elements out anyways and we prefer to have our "norm map" to map to the scalars* $\mathbb{F}^{\times}$ *instead of to* $\bigwedge^{[\times]}(\mathcal{R})$, *it is safe and reasonable to work with the extended quadratic form* $\bar{\mathfrak{q}}$ *in all definitions.*

*In this paper we are mostly interested in working with the orthogonal groups and thus are interested in preserving the surjectivity of* $\rho$. *Since, for computational reasons, we usually restrict ourselves to the* $\mathbb{R}$-*case, it is easier to work with (a restricted set of) elements of a group than equivalence classes, and, the spinor norm/extended quadratic form has computational meaning, we side with the second definition in this paper, but only state it for the* $\mathbb{R}$-*case below. We leave the general definition for future discussion.*

**Definition E.40** (The real Pin group and the real Spin group). *Let* $V$ *be a finite dimensional* $\mathbb{R}$-*vector space* $V$, $\dim V = n < \infty$, *and* $\mathfrak{q}$ *a (possibly degenerate) quadratic form on* $V$. *We define the (real)* Pin group *and (real)* Spin group, *resp., of* $(V,\mathfrak{q})$ *as the following subquotients of the Clifford group* $\Gamma(V,\mathfrak{q})$ *and its even parity part* $\Gamma^{[0]}(V,\mathfrak{q})$, *resp.:*

$$\mathrm{Pin}(V,\mathfrak{q}) := \left\{x \in \Gamma(V,\mathfrak{q}) \mid \bar{\mathfrak{q}}(x) \in \{\pm 1\}\right\} / \bigwedge\nolimits^{[*]}(\mathcal{R}), \tag{605}$$

$$\mathrm{Spin}(V,\mathfrak{q}) := \left\{x \in \Gamma^{[0]}(V,\mathfrak{q}) \mid \bar{\mathfrak{q}}(x) \in \{\pm 1\}\right\} / \bigwedge\nolimits^{[*]}(\mathcal{R}). \tag{606}$$

*If* $(V,\mathfrak{q}) = \mathbb{R}^{(p,q,r)}$ *is the standard quadratic* $\mathbb{R}$-*vector space with signature* $(p,q,r)$ *then we denote:*

$$\mathrm{Pin}(p,q,r) := \mathrm{Pin}(\mathbb{R}^{(p,q,r)}), \tag{607}$$

$$\mathrm{Spin}(p,q,r) := \mathrm{Spin}(\mathbb{R}^{(p,q,r)}). \tag{608}$$

**Corollary E.41.** *Let $(V, \mathfrak{q})$ be a finite dimensional quadratic vector space over $\mathbb{R}$. Then the twisted conjugation induces a well-defined and surjective group homomorphism onto the group of radical preserving orthogonal automorphisms of $(V, \mathfrak{q})$:*

$$\rho : \text{Pin}(V, \mathfrak{q}) \to \text{O}_{\mathcal{R}}(V, \mathfrak{q}), \tag{609}$$

*with kernel:*

$$\ker(\rho : \text{Pin}(V, \mathfrak{q}) \to \text{O}_{\mathcal{R}}(V, \mathfrak{q})) = \{\pm 1\}. \tag{610}$$

*Correspondingly, for the $\text{Spin}(V, \mathfrak{q})$ group. In short, we have short exact sequences:*

$$1 \longrightarrow \{\pm 1\} \xrightarrow{\text{incl}} \text{Pin}(V, \mathfrak{q}) \xrightarrow{\rho} \text{O}_{\mathcal{R}}(V, \mathfrak{q}) \longrightarrow 1, \tag{611}$$

$$1 \longrightarrow \{\pm 1\} \xrightarrow{\text{incl}} \text{Spin}(V, \mathfrak{q}) \xrightarrow{\rho} \text{SO}_{\mathcal{R}}(V, \mathfrak{q}) \longrightarrow 1. \tag{612}$$

The examples E.19, E.20, E.21, E.38 allow us to describe the elements of the Pin and Spin group as follows.

**Corollary E.42** (The elements of the real Pin group and the real Spin group)**.** *Let $(V, \mathfrak{q})$ be a finite dimensional quadratic vector space over $\mathbb{R}$ with signature $(p, q, r)$. We get the following description of the elements of the Pin group:*

$$\text{Pin}(V, \mathfrak{q}) = \left\{ \pm v_1 \cdots v_k \cdot \gamma_1 \cdots \gamma_{p+q} \cdot g \,\middle|\, k \in \mathbb{N}_0, \mathfrak{q}(v_l) \in \{\pm 1\}, g \in \bigwedge^{[*]}(\mathcal{R}) \right\} / \bigwedge^{[*]}(\mathcal{R}), \tag{613}$$

*where $\gamma_i = 1 + e_i \sum_{j=1}^{r} c_{i,j} e_{p+q+j}$, $c_{i,j} \in \mathbb{R}$, for $i = 1, \ldots, p + q$ and $j = 1, \ldots, r$, and, $v_l \in V$ with $\mathfrak{q}(v_l) \in \{\pm 1\}$ with $l = 1, \ldots, l$ and $k \in \mathbb{N}_0$. Note that $\{e_{p+q+j} \,|\, j \in [r]\}$ is meant to span the radical subspace $\mathcal{R}$ of $(V, \mathfrak{q})$.*

*We similarly can describe the Spin group as follows:*

$$\text{Spin}(V, \mathfrak{q}) = \left\{ \pm v_1 \cdots v_k \cdot \gamma_1 \cdots \gamma_{p+q} \cdot g \,\middle|\, k \in 2\mathbb{N}_0, \mathfrak{q}(v_l) \in \{\pm 1\}, g \in \bigwedge^{[*]}(\mathcal{R}) \right\} / \bigwedge^{[*]}(\mathcal{R}), \tag{614}$$

*with the same conditions as above, but where $k$ needs to be an even number.*

**Corollary E.43.** *Note that we also get well-defined group representations:*

$$\rho : \text{Pin}(V, \mathfrak{q}) \to \text{Aut}_{\mathbf{Alg}, \text{grd}}(\text{Cl}(V, \mathfrak{q})) \cap \text{O}_{\bigwedge(\mathcal{R})}(\text{Cl}(V, \mathfrak{q}), \bar{\mathfrak{q}}), \tag{615}$$

*with kernel $\ker \rho = \{\pm 1\}$.*

*In particular, $\text{Cl}(V, \mathfrak{q})$ and $\text{Cl}^{(m)}(V, \mathfrak{q})$ for $m = 0, \ldots, n$, are orthogonal group representations of $\text{Pin}(V, \mathfrak{q})$ via $\rho$:*

$$\rho : \text{Pin}(V, \mathfrak{q}) \to \text{O}_{\bigwedge^{(m)}(\mathcal{R})}(\text{Cl}^{(m)}(V, \mathfrak{q}), \bar{\mathfrak{q}}). \tag{616}$$

*Also, if $F(T_1, \ldots, T_{\ell+s}) \in \mathbb{R}[T_1, \ldots, T_{\ell+s}]$ is a polynomial in $\ell + s$ variables with coefficients in $\mathbb{R}$ and $x_1, \ldots, x_\ell \in \text{Cl}(V, \mathfrak{q})$, $y_1, \ldots, y_s \in \bigwedge(\mathcal{R})$, and $k \in \{0, \ldots, n\}$. Then for every $w \in \text{Pin}(V, \mathfrak{q})$ we have the equivariance property:*

$$\rho(w)\left(F(x_1, \ldots, x_\ell, y_1, \ldots, y_s)^{(k)}\right) = F(\rho(w)(x_1), \ldots, \rho(w)(x_\ell), y_1, \ldots, y_s)^{(k)}. \tag{617}$$

