= 0, \dots, n$, are orthogonal group representations of $\mathrm{Pin}(V, \mathfrak{q})$ via $\rho$:*

$$\rho : \mathrm{Pin}(V, \mathfrak{q}) \to \mathrm{O}_{\bigwedge^{(m)}(\mathcal{R})}(\mathrm{Cl}^{(m)}(V, \mathfrak{q}), \bar{\mathfrak{q}}). \tag{616}$$

*Also, if $F(T_1, \dots, T_{\ell+s}) \in \mathbb{R}[T_1, \dots, T_{\ell+s}]$ is a polynomial in $\ell + s$ variables with coefficients in $\mathbb{R}$ and $x_1, \dots, x_\ell \in \mathrm{Cl}(V, \mathfrak{q})$, $y_1, \dots, y_s \in \bigwedge(\mathcal{R})$, and $k \in \{0, \dots, n\}$. Then for every $w \in \mathrm{Pin}(V, \mathfrak{q})$ we have the equivariance property:*

$$\rho(w)\left(F(x_1, \dots, x_\ell, y_1, \dots, y_s)^{(k)}\right) = F(\rho(w)(x_1), \dots, \rho(w)(x_\ell), y_1, \dots, y_s)^{(k)}. \tag{617}$$