# OpenReview forum: "Clifford Group Equivariant Neural Networks"
_NeurIPS.cc/2023/Conference — NeurIPS 2023 oral_

### Official Review · Reviewer_5NZ6 · 2023-06-22

**Soundness:** 4 excellent
**Presentation:** 4 excellent
**Contribution:** 4 excellent
**Rating:** 8
**Confidence:** 4

**Summary:**

This paper presents a novel approach for creating E(n)-equivariant models based on Clifford (geometric) algebras.

The authors identify the Clifford group and its action. By extending this action to the entire Clifford algebra, remarkable properties can be obtained enabling the construction of equivariant maps from the Clifford algebra to itself, as well as other equivariant operations, such as grade projection.
Using these, an equivariant Clifford group neural network operating on vector fields can be constructed, capable of a more accurate and nuanced interpretation of the underlying structures than the baseline scalar-field networks.

The proposed method exhibits application versatility across different tasks and dimensionalities of the space.

**Strengths:**

S1. The high quality of writing and presentation are two strong features of the paper. The authors succeeded in finding a good balance between the amount of technical detail and the narration clarity.

S2. The paper presents novel theoretical results based on Clifford algebras, enabling an original way of constructing equivariant neural networks, which is a great contribution to the field.

S3. Experimental validation is versatile and, to a large extent, convincing.

**Weaknesses:**

W1. Missing details in the presentation of the proposed method, including:

(See the questions section for more information.)

- translation equivariance (Q1)

- invariant prediction computation (Q2),

- the architecture of the proposed networks in the experiments (Q4).



W2. Incomplete comparison with other methods:

- The support of the claim that the scalar-feature methods are not able to extract some essential geometric properties from input data needs to be elaborated on (Q3).

- Discussion on the complexity of the proposed method vs the baselines is missing (Q4).

- Some training details are missing (Q5).


Addressing these weaknesses will improve the presentation and clarify the support of certain claims.

**Questions:**

Q1. Translations:

- The authors claim to present an E(n)-equivariant method.

- As discussed in Section 3.2, equivariance w.r.t. the Clifford group implies equivariance w.r.t. the orthogonal group O(n). It is, however, unclear how **translation** equivariance is attained.
In fact, all the presented experiments involve only the orthogonal group transformations: in the $n$-body experiment, the authors use mean-subtracted positions of the particles, thereby removing the effect of translation already on the input level.

- Could the authors clarify how translations can be handled using the proposed layers?



Q2. How are O(n)-**invariant** predictions obtained with the proposed method?



Q3. The claim of better underlying geometric properties capturing is meant to be supported by the *signed-volume experiment*. I would like for the authors to elaborate on the claim itself and clarify the presented experimental support.

- How do the authors define *covariance*?

- Some of the considered scalar-feature methods extract equivariant features and compute the invariant features from them to perform classification/segmentation requiring this.
For instance, in the theory of the VN method [34], there's nothing that restricts the features to only be equivariant under rotations and not reflections. In fact, the invariant features are obtained as the inner product of equivariant features, which also cancels the effect of reflections thus making the method produce O(3)-invariant predictions.
This makes the VN method unable to distinguish a tetrahedron and its reflected copy (and thus, their signed volumes will not be distinguished either) by the design of the invariant computation block only, even though the method extracts equivariant features.
This should be taken into account in the comparison in Section 5.1.

- Could the authors provide more details of the experimental setup? I want to see the number of samples, the ratio of positive/negative volumes, and the success rates of the predictions given positive/negative volumes for the proposed method and the baselines.



Q4. Could the authors provide details of their method architecture used in the experiments in Sections 5.1-2? How does the complexity of the method compare to the baselines?



Q5. Is the training setup the same for all the methods in the experiments? I.e., were the training hyperparameters optimized for the proposed method and used for others?

**Limitations:**

The limitations and broader impact are adequately addressed by the authors.

---

> ### Author Rebuttal · Authors · 2023-08-08
>
> We thank the reviewer for their highly valuable feedback. We address their concerns and questions in the following. The reviewer has already transformed weaknesses into questions (thanks!), which we answer directly.
>
> 1. *How is translation equivariance obtained?*
>
>     a. Translation invariance is obtained through the subtraction of a reference point, like the mean position of the point cloud, which is a typical approach to getting translation invariance. To obtain equivariance, one simply adds the reference point back onto the NN's prediction. Villar et al. (2021) show that this has some universality properties regarding invariant functions. While relative positions are the golden standard for translation equivariance, the Clifford algebra approach can offer an alternative. One of the reasons why we developed the theory for the fully general, potentially degenerate metric case is that we can get $\rho(w)$ to act as a Euclidean isometry (see Remark D.30). This means we can get translation equivariance without subtracting a reference point. This is what the recent developments in projective geometric algebra (PGA) study, which use the signature $(n, 0, 1)$ (Roelfs \& de Keninck, (2021)). However, there were some practical technicalities. To give an idea, we have to map the data from the usual vector space V to its dual subspace in a well-defined manner, which is where the Euclidean group acts. However, here the metric is fully degenerate and does not correspond to the metric on the original space V. These difficulties motivated us to leave this for future work.
> 2. *How are invariant predictions obtained?*
>
>     a. There are two approaches to obtaining invariance. First, the scalar subspace $\mathbb{F}=\mathrm{Cl}^{(0)}(V, q)$ is always invariant with respect to $\rho(w)$. So, one can take the grade projection of the NNs output onto the scalar subspace to obtain invariant outputs. On top of this, the algebra-extended quadratic form $\bar{q}: \mathrm{Cl}(V, q) \to \mathrm{Cl}^{(0)}(V, q)=\mathbb{F}$ is always invariant. So one can concatenate the zero-projection with the quadratic form of the covariant subspaces to get an invariant. One can use typical feed-forward neural networks to project this to an invariant of the dimension the problem requires.
>
>     **Action taken:** we now clarify this in the invariant tasks.
>
> 3. *Clarifications regarding the volume experiments.*
>
>     a. 1. We define a covariant object as a quantity that transforms in a nontrivial but predictable way under a change of coordinates. 2. The reviewer is right that scalar methods such as the default implementation of VNs produce O(3) invariant features, and thus VNs are not able to distinguish a tetrahedron and its reflected copy. We add this for clarity. Of course, we do not claim that such methods cannot be adapted to obtain access to these features. However, one of the merits of the Clifford algebra networks is that the network can readily distinguish these cases. 3. We're sorry that there is an unclarity regarding classification/regression. In Sec. 5.1, we detail that we regress the signed volume against the spatial positions. The number of training samples is then given in Figure 3. The ratio of positive vs. negative volumes is 50/50. The task is thus to predict the volume and not only its orientation.
>
>     **Action taken:** we adjusted section 5 to clarify these matters.
>
> 4. *Clarify the neural architecture against baselines for Section 5.1-2. How does the complexity of the method compare against the baselines?*
>
>     a. We use feed-forward networks based on the layers proposed in Section 3. We compare against baselines of similar numbers of parameters, where source codes were taken from original implementations. For 5.2, we took the numbers directly from the source paper, and used networks with similar numbers of parameters. For further details, please consider the attached PDF file.
>
>     **Action taken:** we now elaborate better on model and baseline complexities in the experimental section.
>
> 5. *Is the training setup the same for all experiments? Were hyperparameters optimized?*
>     a. We did no exessive hyperparameter tuning for CGENN architectures in experiments 5.1 and 5.2. We rather used design principles (number of parameters, layers from baseline architectures). We searched for learning rates for 5.3 and tried a few architectures and learning rate regimes for 5.4. Reported baseline numbers for experiments 5.2, 5.3, and 5.4 are taking from existing literature. For experiments in 5.1, baselines were optimized for similar parameter counts as CGENNs. We also ran small learning rate searches for them.
>
> We are very grateful for the reviewer's feedback. Where applicable, we have made an effort to disclose all the changes made to the manuscript. Given this and our previous discussions, we are hopeful that the reviewer is open to promoting the score. Should there be any questions or concerns, we are always ready for further dialogue.
>
> #### References
> - Villar, Soledad, et al. Scalars are universal: Equivariant machine learning, structured like classical physics. Advances in Neural Information Processing Systems 34 (2021): 28848-28863.
> - Roelfs, Martin, and Steven De Keninck. Graded symmetry groups: plane and simple. Advances in Applied Clifford Algebras 33.3 (2023): 30.

---

> > ### Comment · Reviewer_5NZ6 · 2023-08-11
> >
> > I appreciate the authors' comprehensive rebuttal addressing all of my questions and thank them for that!
> >
> > I would further like to clarify the following items.
> >
> > 1. *Translation equivariance*
> >
> >     Is the network configuration used in the $n$-body experiment translation-*equivariant*?  I.e., do the authors add the subtracted point onto the prediction?
> >
> >     I would recommend that the authors include the translation-equivariance clarification in the paper.  Otherwise, it is not entirely clear how exactly the currently presented method is suitable for constructing *E(n)*-equivariant networks (line 2).
> >
> > 2. *Neural architecture*
> >
> >       With reference to Table 1 in the rebuttal PDF and comparing the architectures for O(5)-volume (convex hulls) and O(5)-regression experiments, I wonder what causes the difference in the number of parameters.
> >
> >     In both cases, the authors use a 4-layer feedforward network. However, for the (seemingly simpler) task of O(5)-regression where the input is 2 5D points (vs., I presume, larger input for convex-hull volume prediction), the number of parameters is one order of magnitude higher.
> >
> >      -  Is there a typo or how can the authors explain this otherwise?
> >      -  Using a feed-forward neural structure in experiments 5.1-5.2, do the authors address the *permutation equi- and invariance* wrt the input points/particles? Or is this requirement relaxed?
> >     - Could the authors also clarify this for experiments 5.3-5.4?

---

> > > ### Author Response · Authors · 2023-08-13
> > >
> > > Thank you for your thoughtful queries and for giving us the opportunity to further clarify our work.
> > >
> > > **Translation equivariance**: In the $n$-body experiment, we have indeed outlined the translation equivariance by mentioning, “The input consists of the mean-subtracted particle positions (to achieve translation equivariance) and…”. To make this clearer, we will explicitly mention in the revised version that we add the subtracted point back onto the prediction. We agree that emphasizing this point will provide better clarity on how our method achieves E(n)-equivariance, and we appreciate your recommendation in this regard.
> > >
> > >
> > > **Neural architecture**: The difference in parameter count between the O(5)-volume (convex hulls) and O(5)-regression experiments is primarily due to the different numbers of hidden dimensions used. The latter uses hidden dimensions of 128 (which is arguably overparameterized) compared to the former's 16 dimensions. For the regression experiment, we intentionally matched the number of parameters to those of the baselines to ensure a fair comparison. We'll ensure that such details are more explicitly mentioned in the revised manuscript.
> > >
> > > **Permutation invariance**: In our synthetic experiments, we relax the permutation invariance. Specifically: for the convex hull experiments, our goal was to test our raw feed-forward parameterizations in their most unadorned form against the baselines, among which some were also introduced as feed-forward architectures.
> > >
> > > **Dealing with permutation invariance in experiments 5.3 and 5.4**: We employ graph neural networks (GNNs) to address the permutation invariance for these experiments. Although our manuscript was limited in detailing this due to space constraints, we briefly touched upon using graph neural networks in 5.3. Delving deeper, these are straightforward message passing networks wherein both the message and update networks utilize our proposed layers. We acknowledge the omission of some of these details, especially regarding experiment 5.4, in our pursuit to fit the space constraints. We will rectify this by adding these details back in the revised version for a clearer understanding.
> > >
> > > We hope that these explanations successfully addressed your comments. We appreciate the time you spent on this review, which has resulted in an improved revision of the manuscript. If it aligns with your assessment, we'd be grateful if you'd consider promoting our score. As always, we remain at your disposal for any further questions or clarifications.

---

> > > > ### Comment · Reviewer_5NZ6 · 2023-08-13
> > > >
> > > > Thank you for thoroughly answering my queries!
> > > >
> > > > Based on the quality and content of your rebuttal addressing my concerns about the original manuscript, I am confidently raising your score.

---

### Official Review · Reviewer_W8CU · 2023-06-29

**Soundness:** 3 good
**Presentation:** 3 good
**Contribution:** 3 good
**Rating:** 8
**Confidence:** 1

**Summary:**

I like to admit that I am not familiar with this area of work. I try my best to review the key contributions of this work. My review is more of clarification than critic.

The theory part of the paper lies mainly in section 3.1 starting with Eq.(5).
The computation part of the paper is given in section 4. The authors introduce two kinds of layers. Linear Layers and Geometric Product Layers.

Linear layer mix the input channels of the same grade (Eq.(11)).
Interaction layer takes the geometric product of the i and j grade of x1, x2 and project to the k-th grade.

Several experiments were performed using this neural network. I will discuss more about the experiments in later part of this review.

**Strengths:**

1. The paper explore a different approach to deep learning. This paper helps in adding a new dimension in the field of deep learning.
2. The authors provided a huge amount of background and foundations in the supplementary materials. This is important because I presume relatively few readers are familiar with Clifford algebra.
3. Although I cannot fully understand the theoretical part of this work. It seems to me the authors try to substantiate their work with theories.

**Weaknesses:**

1. The strength of this paper may also be "its weakness", readers without background on Clifford algebra may struggle to pick up this area of work. It will be good if the authors follow up with a good tutorial. Perhaps a series of online learning materials and provide its link in this paper. For example, make an archive tutorial paper (or online video) and those locations are cited in this paper. The supplementary materials of this paper is good. but it is insufficient to bring people outside of this field up to the level to use this paper.
2. The output of the neural network is given by Eq.(11) and Eq.(13). I understand that the output is also an element of the Clifford algebra. I struggle to understand how this output is link to the use cases in the experiments. It will be good if the authors help the readers by describing the experiments in detail. Give a step-by-step walk through of :
2a: how the data is being prepared and in what format
2b: how the data is being feed into the neural network
2c: how to link the k-th grade of x to the numerical output.
2d: perhaps walk through with a simple example, like one example using complex numbers.

I explain 2c a bit more here. For a complex number c1 = 2+3i, the zeroth grade is 3 (or the 1 element). The first grade is 2 (or i). If the author can guide the reader through on how to input c1 into the linear layer and explain what is the output, that will help. Similarly, the author may also give another example c1=1-2i and c2=4+I on how to feed c1 and c2 into Eq.(12) and Eq.(13).

**Questions:**

How to prepare data to feed into the neural network? Perhaps the authors can give a simple example(s) of processing complex numbers and/or quaternions.


**Limitations:**

Readers without a background of abstract algebra or Clifford algebra may find it hard to understand this paper.

---

> ### Author Rebuttal · Authors · 2023-08-08
>
> We thank the reviewer for the actionable feedback. We address their concerns and questions in the following.
>
> 1. *Familiarize readers with the Clifford algebra, perhaps with a tutorial.*
>
>     a. Following up with a tutorial is a great suggestion! Indeed, we are already working on a blog post series that explains the subject matter in a less demanding way. We realized, after studying related work, that the orthogonal group acting on the Clifford algebra was not studied in a most general form. Physicists, engineers, and mathematicians usually restricted themselves to, e.g., definite or nondegenerate quadratic forms. In the context of deep learning, we studied this more generally, requiring a more rigorous approach.
>
> 2. *How is the data being prepared and processed? Explain with an example.*
>
>     a. We can give you some intuitions here. 2a: The neural network tensor shapes of the Clifford algebra networks are of the form $B \times C \times 2^n$, where $n = \dim(V)$. If we have scalar-valued data, we can use the first component of that last dimension to embed those, since $\mathrm{Cl}^{(0)}(V, q)=\mathbb{F}$. For vector-valued data we can use $\mathrm{Cl}^{(1)}(V, q) = V$. The other components are left at zero at first. 2b: This tensor is then fed into the neural network, which also computes features in the other Clifford subspaces, and as such, we get densely filled multivectors. 2c: At the end of the network, we can project onto the grade $k$ subspace to get a $k$-vector. Usually, this would be the grade $0$ subspace for scalar-valued predictions or grade $1$ for vector-valued predictions. 2d: Regarding your complex numbers example, the tensor shape would be $B \times C \times 2$, where we have 2 for the real and imaginary components. You populate the last dimension using your complex-valued data. We hope this helps!
>
>     **Action taken:** we now give a bit more intuition about these procedures in the experimental section.
>
>
> We thank the reviewer for their valuable insights. We have strived to be as explicit as possible in tackling the remaining issues. If our responses meet expectations, we hope the reviewer maintains their support for our paper. We are ready for any more discussions or questions the reviewer might have.

---

> > ### Comment · Reviewer_W8CU · 2023-08-18
> >
> > I like to thank the author for answering my queries. I will work through the paper again. Really look forward to any new tutorial materials. I understand there is a deadline for rebuttal and focus of the author should be to discuss here. I look forward to the eventual tutorial materials on arxiv after the whole process of double blinded process is over.

---

### Official Review · Reviewer_jDZi · 2023-07-04

**Soundness:** 4 excellent
**Presentation:** 2 fair
**Contribution:** 3 good
**Rating:** 8
**Confidence:** 4

**Summary:**

In this submission the authors introduce an equivariant neural network architecture building on the Clifford algebra (defined by a vector space and a quadratic form).
They remind that this algebra admits an orthogonal decomposition over sub-vector spaces called grade $m$ where $m$ is the number of basis elements.
This is referred as the 'multivector grading of the Clifford algebra'.
They identify a (Clifford) subgroup and its (adjusted twisted conjugation) action which not only acts on the original vector space but also on the Clifford algebra whilst satisfy many properties including multiplicatively w.r.t. the geometric product.
They show that equivariance w.r.t. this a Clifford group acting on the Clifford algebra implies equivariance w.r.t. the orthogonal group acting on the Clifford algebra.
Then they propose several layers, including a linear layer and a geometric product layer which they show to be equivariant w.r.t. the action of the Clifford group.
Then they empirically assess their proposed approach on synthetic tasks with $O(3)$, $O(5)$ or $E(3)$ equivariance and show that it overperform scalar based methods but also steerable NNs.
Finally, they tackle the task of identifying top quarks from high-energy jets produced in particle collisions at CERN, and show that they are able to perform as well as recent specialised neural networks.


**Strengths:**

- This is a nice paper that builds on this recent research avenue of equivariant neural networks based on the Clifford/geometric algebra.
- In find particularly appealing the fact that the methodology applies to any dimension, and more generally than $O(n)$ equivariance via the choice of quadratic form $\mathfrak{q}$ as per Section 5.4 (if I understood correctly).
- Additionally, it appears that the method is empirically able to fit well a variety of functions and able to generalise.
- The paper is pretty well written given the space constraint, although I give some suggestion on improvements below.

**Weaknesses:**

- Likely due to space constraint, yet the methodology would deserve more discussion.
    - For instance, how is the data embedded into the algebra (e.g. for higher order quantities such as an inertia tensor)?
    - How is the quadratic form $\mathfrak{q}$ chosen (e.g. is it the Minkowski product for the Jet tagging experiment)?
    - I also believe that the linear and geometric layers could benefit from an illustrative diagram. Perhaps it may be useful to draw parallel with tensor product and irreducible representations for readers familiar with this language.
- What's more the main paper is currently lacking a discussion on scalability of the method, and how this compares to Clebsch-Gordan tensor product approaches, although I understand that the space is limited.
- Another points which I find slightly frustrating is the lack of ablation study, as generally it is hard to identify why this method overperform tensor product / irreps approaches. Is it more parameter efficient? Does it generalise better? Does it yield a better optimisation landscape? Is it computationally faster? Obviously fully answering this questions would deserve a full paper on its own, yet some elements of answer would be nice :)

**Questions:**

- line 62: 'our method readily generalizes to orthogonal groups regardless of the dimension'. Doesn't the method proposed in [Cesa et al. 2021] also work for any dimension $n$?
- Eq 1: What is the intuition behind the independence of the choice of orthogonal basis? Is it because by combining elements from one basis (via the geometric product) one could obtain the second basis? How come the limiting factor is really the number $m$ of such basis elements?
- line 232: Worth saying a bit more about the 'layer-wide normalization' proposed in [70]?
- line 235: 'However, we still require activation functions after linear layers'. Does this statement come from practical requirements for the method to work well, or from some universal approximation theorem with necessary non-linearity after the linear layer?
- Section 5.1: Is the 'signed volume' a pseudo-scalar?
- One of the advantage of this method that it readily works whatever the input space dimension, yet what is the scalability of the method with respect to the dimension of the space?
- Section 5.2: What are the reasons for EMLP underperforming? It is theoretically a universal approximator.
- Section 5: Do all methods have the same number of parameters? or same compute time?


A PROGRAM TO BUILD E(n)-EQUIVARIANT STEERABLE CNNS, Cesa, Gabriele and Lang, Leon and Weiler, Maurice, 2021

**Limitations:**

- 'CGENNs induce a (non-prohibitive) degree of computational overhead'. Where does this overhead come from? What specific operation would benefit from having a custom GPU kernel? More generally it would be really nice to discuss the scalability of the method. In particular vs other 'steerable' methods which rely on parametrised tensor product via Clebsch-Gordan coefficients.

---

> ### Author Rebuttal · Authors · 2023-08-08
>
> We thank the reviewer for the actionable feedback. We address their concerns and questions in the following.
>
> 1. *Data embedding?*
>
>     a. The data is embedded by an additional multivector dim in the network tensors (response to reviewer W8CU), whose size depends on the algebra, e.g., 4 or 8 for 2D or 3D algebras. Multivectors, under the geom. product, parameterize bilinear operations similar to tensors. Not all tensors can be expressed as multivectors, but many from physics can. For the inertia tensor example, see Berrondo et al.(2012).
>
>     **Action taken:** we give more intuition now in the paper.
>
> 2. *Choice of quadratic form?*
>
>     a. In 3D, the signature is $[1, 1, 1]$, i.e., $p=3, q=0, r=0$. For more specialized problems we use different signatures. E.g., for SU(2), we use $[-1, -1]$, i.e., $p=0, q=2, r=0$ (see response to w3C8). In space-time settings, we use $[1, -1, -1, -1]$, i.e., $p=1, q=3, r=0$.
>
>     **Action taken:** we explain this now in the paper.
>
> 3. *Visualization?*
>
>     a. We added visualizations of the layers to the attached PDF and to the appendix.
>
> 4. *Scalability of the method?*
>     a.  Let $\dim(V)=n$, then the complexity of a fully connected geometric product layer is roughly $O(c _ {out} \cdot c _ {in}  \cdot 2^{3n})$, where $c _ {out}$ and $c _ {in}$ are output and input channels.
>
>     The scalability is worse than scalarization methods, and similar to Clebsch-Gordan based E3NN depending on the highest order of irrep that is chosen.
>
> 5. *Performance?*
>
>     a. It is a rather general finding that any polynomial in multivectors is equivariant in any inner-product space. We constructed our layers with this in mind, ensuring that we capture the 0th, 1st, and 2nd-order terms of such a polynomial, leading to flexible parameterizations. In 3D, CGENNs are similar to E3NN based layers, i.e., existing results of E3NN based methods (e.g., TFN, SEGNN) should match the performance (see SEGNNs in the $n$-body experiment). However, depending implementation and optimization specifics, numbers can differ. Clear benefits of CGENNs are found in non-Euclidian (higher-dimensional) cases.
>
> 6. *Compare to Cesa et al. (2021)?*
>
>     a. Currently, Cesa et al. support steerable CNNs equivariant to 2D/3D isometries. Other differences: Cesa et al. design equivariant CNNs, where the group acts on the grid (e.g., an image). We operate on the particles directly. Second, Cesa et al. find equivariant bases for linear maps in such CNNs. Our steerable basis is given directly by the input data, and as such we also operate using bilinear layers, which is crucial for expressive GNNs (Finzi et al., 2021).
>
> 7. *Intuition behind the independence of the choice of basis?*
>
>     a. Great question! In general, basis independency goes back to Einstein's covariance principle, stating that one should come to the same conclusions regardless of the chosen frame of reference. More specifically, it means that grade-decomposition is well-defined and not dependent on a change of coordinates. We were unable to find in the literature a proof for this decomposition in the (non-definite, potentially degenerate) general case. As such, we regard it an original contribution of the work. Secondly, it directly allows for the equivariance of the grade projections!
>
>     **Action taken**: we explain the basis independence proof more clearly now.
>
> 8. *Layer norm.*
>
>     a. The layer norm is defined as $x \mapsto \frac{x - \mathbb{E}[x]}{\mathbb{E}[\sqrt{|q(x)|}}$, where the expectations are taken over channel dims. Intuitively, it re-centers and rescales multivectors according to their average norm.
>
> 9. *Nonlinearities?*
>
>     a. We need a sufficiently flexible, nonlinear function to learn expressive representations. We empirically find that including additional nonlinearities after the linear layers results in improved performances.
>
>     **Action taken**: we mention this in the methodology section now.
>
> 10. *Signed-volume a pseudoscalar?*
>
>     a. Yes.
>
> 11. *Scalability regarding space dimension?*
>
>     a. The dimensionality of the algebra scales as $2^n$, where $n=\dim(V)$. This looks bad at first sight, but usually the dimensionality of geometric spaces is not so large. Further, one can operate using subalgebras, which is typical in geometric algebra literature, or one can simply not consider the highest subspaces.
>
> 12. *What are the reasons for EMLP underperforming?*
>     a. For O(5) regression, we took datasets and EMLP results directly from Finzi et al. (2021). We went through the EMLP code and noted that the output irreps of each layer is set equal to the output irrep of the whole task. For the O(5) regression task, whose output irrep is one-dimensional, this means that there will be no vector-valued outputs of the bilinear layers. This is potentially less expressive. Further, it depends highly on the specifics of the implementations and parameterizations. Our polynomials form a rather general basis from which we constructed our layers.
>
> 13. *Do all methods have the same number of parameters or compute-time?*
>
>     a. We kept the number of parameters similar between models. The compute-time however can depend on which method one uses. Steerable methods that use bilinear layers are generally slower than scalarization methods. Compared to existing steerable methods, Clifford layers are not fundamentally less efficient.
>
> 14. *Computational overhead, scalability.*
>
>     a. The computations overhead originates from the bilinear geometric product layers. Please also consider our previous responses (4, 13) regarding scalability.
>
> #### Refs
> - Berrondo, M., J. Greenwald, and C. Verhaaren. Unifying the inertia and Riemann curvature tensors through geometric algebra. American Journal of Physics 80.10 (2012): 905-912.
> - Finzi, Marc, Max Welling, and Andrew Gordon Wilson. A practical method for constructing equivariant multilayer perceptrons for arbitrary matrix groups. ICML, 2021.

---

> > ### Comment · Reviewer_jDZi · 2023-08-21
> > **response**
> >
> > Thanks for taking the time to address my questions and remarks, but also for providing additional visualisations!
> > I acknowledge that it's tricky to strike a good balance between clarity and space constraint, and believe that the submission is doing a pretty good job already.

---

### Official Review · Reviewer_gQKE · 2023-07-09

**Soundness:** 3 good
**Presentation:** 2 fair
**Contribution:** 4 excellent
**Rating:** 7
**Confidence:** 4

**Summary:**

The paper introduces a new method for constructing E(n)-equivariant networks called Clifford Group Equivariant Neural Networks (CGENNs), based on an adjusted definition of the Clifford group, a subgroup of the Clifford algebra. The researchers have shown that the group’s action forms an orthogonal automorphism that extends beyond the typical vector space to the entire Clifford algebra, respecting its multivector grading and multiplicative structure. This leads to non-equivalent subrepresentations corresponding to the multivector decomposition and allows parameterizing equivariant neural network layers. CGENNs operate directly on a vector basis and can be generalized to any dimension.

Incorporating group equivariance in neural networks has proved useful in many areas, including modeling the dynamics of complex physical systems and studying or generating molecules, proteins, and crystals. This paper demonstrates that the CGENNs can handle directional information more accurately and nuancedly than scalarization methods, operating directly in a vector basis rather than alternative basis representations. They can also transform higher-order features carrying vector-valued information, and readily generalize to orthogonal groups of any dimension or metric signature. The authors successfully demonstrated these advantages on various tasks, including three-dimensional n-body experiment, a four-dimensional Lorentz-equivariant high-energy physics experiment, and a five-dimensional convex hull experiment.

**Strengths:**

**Originality**
This paper has many strengths across multiple dimensions. First, it tackles the problem of building an Equivariant network using a fresh perspective that has not been, as far as I know, used in the literature. Thus the use of the Clifford Group is an exciting direction of technological endeavor. Certainly, the mathematics behind building the Linear, Geometric Product, and Nonlinearities is nontrivial but offers what appears to be greater modeling flexibility.

**Quality**
It is abundantly clear that immense care has been taken with the writing of this paper. This is substantiated by the large and extensive appendix which in itself can prove to be a useful reference outside of the main thesis of this paper. Furthermore, it is clear that a computational approach was taken to build the equivariant layers as I appreciated the authors that the authors acknowledged the naive $(n+1)^3$ interactions needed in their geometric layers which could be reduced by first applying a linear projection.

**Clarity**
The authors should be given credit for attempting to make this paper clear and accessible. Unfortunately, it is not outside for those who are likely already familiar with geometric algebras. Although the writing and the flow of information are still very clear if one pretends to know what the Clifford group is.

**Significance**
In this reviewer's opinion, this work is significant for several reasons. First, it allows for the creation of equivariant networks in a completely different manner and one that appears to be more flexible/expressive. This is due to the Clifford group allowing for non-equivalent subrepresentations. For example,  Clifford group equivariant models can extract covariant quantities which cannot be extracted from scalarization methods. Furthermore, the early empirical results suggest that such an approach also leads to empirical benefits and beats current equivariant models which suggests that this is a ripe direction for continued investigation.

**Weaknesses:**

This paper has a few areas for improvement that I highlight below.

1.) I thank the author for their efforts to make the paper accessible, but the appendix is not a very friendly introduction. For instance, in some places, the authors acknowledge that their definitions depart from existing ones in the literature. But in other places, they claim to follow the format of other works (e.g. C.1). This makes it hard to follow and trust what the authors state.

2. Perhaps what bothers me most is the fact that the motivation for using the Clifford algebra group is lost in all the technical jargon. The importance of being able to use multi-vectors in ML-specific problems is never really motivated in a convincing way. Even at present, I find it difficult to pinpoint exact non-toy applications where these tools might prove useful. Albeit there is no doubt the tools themselves are mathematically interesting.

3. There are parts of the paper where it becomes unclear which part of the theory is an original contribution of the authors versus something that is already present in the math literature on Clifford group/algebra. As far as I can tell the design of the adjusted twisted conjugation is novel and its extension is an original contribution of the authors (and the subsequent ML-specific layers). Can the authors tell me if I missed anything?

4. The experiments are quite preliminary. Perhaps this is by design as the authors note that the computational complexity can partially be resolved with better hardware-specific implementations. I would have loved to see the power of these methods on larger-scale equivariant problems. For instance, I am certain the authors are familiar with molecular dynamics simulations at a larger scale. These would already have the desired symmetries and established baselines to compare against.

5. The authors could improve the method section by providing an empirical investigation of the runtime complexity of their actual experiments in comparison to the baselines. Right now, a reader may intuit that the Clifford group equivariant networks are more computationally expensive but, exactly how much is not immediately clear.

**Questions:**

1. One part that is not clear to me is how equivariance is preserved in the geometric layers if you use a linear map to first project down to $l$. This would appear to break equivariance. Can the authors please comment on this and how it doesn't break the desired symmetry?

2. A limitation of the steerable equivariant literature is that it is heavily reliant on the Wigner-D matrices for $O(3)$. It seems the current effort in the Clifford group also applies to the orthogonal and $E(n)$ group. Can this approach be used to model other continuous symmetries e.g. $SU(2)$ or is this not possible?

---

> ### Author Rebuttal · Authors · 2023-08-08
>
> We thank the reviewer for the actionable feedback. We address their concerns and questions in the following.
>
> 1. *Accessibility and introductions of notations.*
>
>     a. During our literature research, we quickly found out that there are at least three communities (physicists, mathematicians, computer vision engineers) working with Clifford algebra (aka geometric algebra). These are all interested in different properties and, as such, use different conventions and notations that were not compatible with each-other. For example, different group actions (e.g., equation 348), or restricting themselves to homogeneous multivectors, or restricting the quadratic form to be definite or nondegenerate. One of the contributions of this paper is the study of existing literatures, unifying and generalizing them. We study the action of the orthogonal group on the whole Clifford algebra within the context of equivariant neural networks. We derived and re-derived everything we needed, with generality and group representations for NNs in mind.
>
> 2. *Motivation of operating on multivector representations.*
>
>     a. There are multiple reasons for using multivector representations. First, our theory allows for networks that operate in any inner product space, computing geometrically meaningful interactions (through geometric products). Second, if one takes a different route and goes for more general tensor product representations, one needs either irreps decomposition, which is hard to attain for any quadratic space, or use tensor product representations directly, which will grow intractably. We argue that Clifford algebra layers balances the two approaches, by allowing for a product structure (geometric product vs tensor product), which, due to their fundamental relations, will always remain finite dimensional, and, a way to decompose the whole Clifford algebra in smaller (but not necessarily irreducible) subrepresentations, which allow to reduce computational complexity (similar to the use of smaller irreps), without violating the equivariance properties. Third, there exist several physics problems that contain objects that transform in a non-standard way. For example, Maxwell’s equation treat magnetic fields as pseudovectors which can be represented as bivectors when using Clifford algebras / Clifford equivariant networks.
>
>     **Action taken:** We elaborate on motivation for multivector representation now in more detail.
>
> 3. *Originality of theory.*
>
>     a. While there already exist literature about the Clifford algebra, different notions of the Clifford groups with different actions, non of these sources we could simply cite for our purposes, because they all use their own notations/conventions, and have different corner cases, aspects or applications in mind. So a major and original contribution was to carefully construct/adjust definitions to get all our desired results (like multiplicativity, orthogonality) out and even provide proofs for the most general (non-definite, degenerate) cases.
>
>     **Action taken**: we clarify this now in the main paper.
>
> 4. *Preliminary experiments.*
>
>     a. The reason we chose not to go with molecular dynamics simulations is that the current state-of-the-art methods are highly finetuned and engineered to squeeze out all the features from the data. We felt that the adaptation and inclusion of these techniques to the Clifford layers was out of scope of the current work, and probably deserves a paper on its own. Finally, we would argue that the top-tagging experiment is at a reasonable scale and much less of a toy setting with 1.2M training samples and fully connected graphs with over 200 particles.
>     Note that architectures, such as LorentzNet, were specifically designed for this task in previous works.
>
> 5. *Improve the method section by providing runtime complexity.*
>
>     a. We are hesitant with making claims about empirical runtime complexity at the current moment. The geometric product as a bilinear operation can be implemented as efficiently as existing methods that use bilinear (tensor) operations. Even stronger, this implementation would then work for any inner product space. However, comparing the current implementations to existing, highly optimized code-bases would be unfair and cast an undeserved shadow on equivariant Clifford layers. To give an approximate result. Please also consider our response to reviewer jDZi regarding runtime complexity.
>
> 6. *Equivariance of projection.*
>
>     a. The equivariance of the projection follows directly from the equivariance of the linear layer. I.e.,
>     $$
>     \rho(w)\left(T^{lin} _ {\phi_{c_{out}}}(x _ 1, \dots, x_\ell)^{(k)}\right) = \rho(w)\left(\sum _ {c _ {in}=1}^\ell \phi _ {c _ {out}c _ {in} k} \, x _ {c _ {in}}^{(k)},\right)=\sum _ {c _ {in}=1}^\ell \phi _ {c _ {out}c _ {in} k} \, \rho(w)\left(x _ {c _ {in}}\right)^{(k)}
>     $$
>
>     which follows from the fact that $\rho(w)$ is linear, commutes with scalar multiplication, and respects grade projections.
>
> 7. *Use the Clifford group for other continuous symmetries like $SU(2)$.*
>
>     a. Yes, and this is very much open for exploration! For your $SU(2)$ example, see e.g. Wilson (2020). Specifics of (Clifford) group representations are elusive for now, but we believe that the presented work gives a good starting point for such scientific endeavors. As such, further explorations for Clifford networks operating in (relativistic) quantum mechanics or quantum information theory is a very interesting future direction.
>
>
> We are grateful for the reviewer's feedback. We have made an effort to disclose all the changes made to the manuscript as clearly as possible. Given this and our previous discussions, we are hopeful that the reviewer continues to support the paper. Should there be any questions or concerns, we are always ready for further dialogue.
>
> #### References
> - Wilson, Robert A. Subgroups of Clifford algebras. arXiv preprint arXiv:2011.05171 (2020).

---

> > ### Comment · Reviewer_gQKE · 2023-08-11
> > **Re:Rebuttal**
> >
> > I thank the authors for their detailed rebuttal. I am mostly satisfied with their answers. I still contend that the Appendix is not an easy introduction to the subject as I might care for but given the scope and divide between the different communities it is still a welcome first step. As such, I'm upgrading my confidence to a 4 and maintaining my original score (7).

---

> > > ### Author Response · Authors · 2023-08-12
> > >
> > > We are grateful for your acknowledgment of the improvements we've made in our rebuttal and are happy to hear that our efforts have enhanced your confidence in our work. Regarding the appendix, we understand and respect your perspective. It's always a balancing act to cater to various communities with varying familiarity with the subject. We'll continue to strive for clarity and accessibility in future iterations. Your feedback has been instrumental in this process.

---

### Official Review · Reviewer_w3C8 · 2023-07-10

**Soundness:** 3 good
**Presentation:** 2 fair
**Contribution:** 3 good
**Rating:** 7
**Confidence:** 2

**Summary:**

The paper develops a new kind of neural network model based on the Clifford group, which makes the neural network E(n)-equivariant.  The model is tested on a number of datasets, showing that it is capable of recovering the desired structure from the data.


**Strengths:**

This seems like an interesting and important direction.  The results are convincing, in the sense that this approach "works."  The method is backed up by nice theory.



**Weaknesses:**

Although the intro and figure 1 at the beginning are helpful, I think the rest of the paper is hard to follow.  There are some theoretical results, but then when it comes to the methodology, section 4, I am left wondering what is actually being computed.  For example, what is involved in computing equation 11?  it says the x_i are elements of Cl(V,q), ok but what is the input to this system?  same for eqs. 12 and 13.
And what is the objective function?
For the first experiment on signed volumes, this seems like an interesting task, but it is not clear what is the form of the input data - a data cloud of x,y,z points?  What is the input to the Clifford network?
I am having a hard time picturing concretely what is going on here.  More specifics would be helpful.



**Questions:**

How would you compare to the following paper: https://proceedings.neurips.cc/paper/2021/file/f1b0775946bc0329b35b823b86eeb5f5-Paper.pdf.
This paper develops a universal family of neural networks for O(d)-equivariant functions. This seems to stand in tension with the claims in the Clifford algebra paper that scalar feature-based methods cannot capture certain kinds of information. (I'm sure that some more thought could explain what's going on, but I feel like that's work the authors of the submission should do, rather than the reader.)
Relatedly, I don't think I saw any statement about whether or not the Clifford algebra approach is universal (maybe I missed it?).
Also, the authors end Section 3 by saying "equivariance w.r.t. the Clifford group [...] implies equivariance w.r.t. the orthogonal group," but this statement, by itself, still leaves a question open: Do there exist functions equivariant under O(d) but not the "Clifford group," which their method then couldn't learn?

---

> ### Author Rebuttal · Authors · 2023-08-08
>
> We thank the reviewer for the encouraging feedback. We address their concerns and questions in the following.
>
> 1. *What is being computed in, e.g., equation 11?*
>
>     a. As input data, we consider points in a vector space $V$, for example, $\mathbb{R}^3$. These can be locations, velocities, forces, etc. Since $\mathrm{Cl}^{(1)}(V, q)$ is isomorphic to the vector space $V$ that we construct it on, we can treat the data as objects in the Clifford algebra. As such, these will be multivectors where only the vector part is nonzero. However, as we propagate through the network, the other multivector elements will get populated. As such, equation 11 computes at first linear combinations of vector coefficients expressed in a fixed basis. But later in the network, these will be multivectors. Equations 12 and 13 compute geometric products.
>
>     **Action taken**: we have added a few sentences clarifying these matters.
>
> 2. *What is the objective function*?
>
>     a. The objective functions in all experiments were mean-squared errors except for the top-tagging experiment. This is a binary classification task and uses the binary cross-entropy loss.
>
>     **Action taken**: we have updated the text to clarify these matters.
>
> 3. *In the volume experiments, what are the inputs to the Clifford network*?
>
>     a. Indeed, the input is a point cloud (tetrahedron). The positions are expressed as coefficients in a fixed basis, which get processed by the neural network.
>
>     **Action taken**: we have updated the text to clarify these matters.
>
> 4. *Compare to Villar et al., 2021.*
>
>     a. Thanks for bringing up this paper! The crucial difference is that Villar et al.  consider representations of the orthogonal group $O(n)$  acting on $(\mathbb{R}^{n})^k$ and maps from $(\mathbb{R}^{n})^k \to \mathbb{R}^n$, or, more in our notation, $V^k \to V$. This is a subset of the spaces that the orthogonal group can act on. There are also maps from other representations of O(n), e.g., tensor product representations, Clifford algebra representations, (irreducible) sub-representations, etc. The signed volume experiment considers a map into the space of scalars that are not invariant to reflections, which is an example of a map not captured by the theory of Villar et al.
>
>     **Action taken**: We have added this paper to the literature list and discuss our findings in comparison with these results.
>
> 5. *The universality of the Clifford approach. Do there exist functions equivariant under O(d) but not the "Clifford group"?*
>
>     a. We show in the paper that there are maps that Clifford layers can represent that scalarization methods cannot. Similarly, we can certainly also construct maps that Clifford methods cannot represent. In our paper, we only consider one fixed action for the Clifford algebra (and their subvector spaces), namely the adjusted twisted conjugation. In this setting, $O(d)$-equivariance is the same as Clifford group equivariance. One can consider maps between spaces outside of the Clifford algebra that the Clifford group does not cover. However, the Clifford algebra offers a closed set of nice, geometrically inspired representations that are more expressive than scalars but still tractable.
>
> We thank the reviewer again for the valuable feedback. Where applicable, we tried to be as transparent as possible about the adjustments made to the manuscript. Considering these and the above, we hope that the reviewer is open to promoting the score. If any questions or comments arise, we are happy to discuss further.

---

### Author Rebuttal · Authors · 2023-08-08

We extend our gratitude to the reviewers for their efforts in evaluating our paper. The overall positive response we received has been highly encouraging. The feedback has undoubtedly improved the quality of our work, and we sincerely appreciate the insightful comments and suggestions that were provided.

We endeavored to consider the feedback as comprehensively as possible,  leading to a revision process that significantly honed the paper. We have strived to make all the necessary changes, balancing them with given space constraints. We have addressed every point in our responses and indicated where the paper was adapted based on the feedback.

We would like to reiterate that our work on Clifford Group Equivariant Neural Networks (CGENNs) sets a distinctive path from the usual geometric deep learning approaches. Unlike typical methods, CGENNs operate directly on an input vector basis compared to, e.g., popular spherical harmonics basis, are more expressive than scalar methods, and apply to inner product spaces of any dimension.

Concluding, we thank the reviewers for recognizing the potential and novelty of our work. It is encouraging to know that our paper is viewed as a significant contribution that provides a direction in the field of geometric deep learning.

---

### Decision · Program_Chairs · 2023-09-21

**Decision:**

Accept (oral)

**Comment:**

This paper considers the Clifford Algebra and designs linear and polynomial layers that are equivariant to actions from the Clifford group. This allows designing neural network architecture suitable for different geometric settings including estimation of volume, convex hull, and $E(3)$ equivariant functions. We encourage the authors to make an effort improving the readability and accessibility of the paper to the general GDL community. Making it more self-contained and adding relevant implementation details would be appreciated.